# Fast, Accurate Manifold Denoising by
# Tunneling Riemannian Optimization

**Shiyu Wang** [1 2]   **Mariam Avagyan** [1 2]   **Yihan Shen** [3 2]   **Arnaud Lamy** [1 2]   **Tingran Wang** [1 2]   **Szabolcs Márka** [4 2 5]
**Zsuzsanna Márka** [6 2]   **John Wright** [1 7 2]

## Abstract

Learned denoisers play a fundamental role in various signal generation (e.g., diffusion models) and reconstruction (e.g., compressed sensing) architectures, whose success derives from their ability to leverage low-dimensional structure in data. Existing denoising methods, however, either rely on local approximations that require a linear scan of the entire dataset or treat denoising as generic function approximation problems, sacrificing efficiency and interpretability. We consider the problem of efficiently denoising a new noisy data point sampled from an unknown $d$-dimensional manifold $\mathcal{M} \in \mathbb{R}^D$, using only noisy samples. This work proposes a framework for test-time efficient manifold denoising, by framing the concept of "learning-to-denoise" as *"learning-to-optimize"*. We have two technical innovations: (i) *online learning* methods which learn to optimize over the manifold of clean signals using only noisy data, effectively "growing" an optimizer one sample at a time. (ii) *mixed-order* methods which guarantee that the learned optimizers achieve global optimality, ensuring both efficiency and near-optimal denoising performance. We corroborate these claims with theoretical analyses of both the complexity and denoising performance of mixed-order traversal. Our experiments on scientific manifolds demonstrate significantly improved complexity-performance tradeoffs compared to nearest neighbor search, which underpins existing provable denoising approaches based on exhaustive search.

## 1. Introduction

*Denoising* is a core task in signal and image processing. Denoisers also play a fundamental role in state-of-the-art approaches to signal *generation* and *reconstruction*. Diffusion models generate intricate images from pure noise, via a sequence of denoising steps; learned compressed sensing methods reconstruct accurate medical and scientific images from incomplete, indirect measurements, again via a sequence of denoising steps. While these iterative procedures produce high-quality results, they are computationally costly: a sophisticated learned denoiser needs to be applied repeatedly to produce a single output. The test-time cost of denoising is a major bottleneck for both high-resolution image generation and real-time image reconstruction.

*Accurate* denoising is critical, because the denoiser encodes prior knowledge about the set of images of interest (natural, medical, scientific, etc.). These images reside near low-dimensional subsets of the image space, which are often conceptualized as low-dimensional manifolds; learning to denoise is tantamount to learning these manifolds.

In this paper, we study a model *manifold denoising* problem, in which the goal is to learn to denoise data lying near a $d$-dimensonal submanifold $\mathcal{M}$ of a high-dimensional space $\mathbb{R}^D$. As we will review below, there are extensive literatures on learning to denoise, and learning manifold models from data. However, there is relatively little work on provably learning *test-time efficient* denoisers. Existing methods with provable good manifold denoising performance involve linearly scanning large datasets. As a baseline, nearest neighbor search across a minimal covering set has worst case complexity at least $O(De^{cd})$, where $e^{cd}$ is the size of the dataset required to cover the d-dimensional manifold, and $D$ is the cost of computing distance in the ambient dimension. On the other hand, more practical neural network models currently lack guarantees of performance and efficiency.

We bridge this gap, developing and analyzing a family of manifold denoisers that are (i) test-time efficient, (ii) accurate, and (iii) trainable. Our main idea is to *cast the problem of denoising a new, noisy test sample $\boldsymbol{x}$ as an optimization problem over the a-priori unknown Riemannian manifold*

---

[1]Department of Electrical Engineering, Columbia University [2]Data Science Institute, Columbia University [3]Department of Computer Science, Columbia University [4]Department of Physics [5]Institute of Advanced Studies (iASK), Chernel utca 14, Kőszeg, 9730, Hungary [6]Columbia Astrophysics Laboratory [7]Department of Applied Physics and Applied Mathematics, Columbia University. Correspondence to: Shiyu Wang <sw3601@columbia.edu>.

*Proceedings of the 42$^{nd}$ International Conference on Machine Learning*, Vancouver, Canada. PMLR 267, 2025. Copyright 2025 by the author(s).

$\mathcal{M}$. This enables us to draw on tools from Riemannian optimization, to develop methods which converge to a small neighborhood of the ground truth – a significant efficiency gain vis-a-vis exhaustive scan. The code for our framework and experiments is available at `https://github.com/shiyu-w/Manifold_Traversal`.

One challenge stems from the unknown nature of the manifold and observation of only noisy samples. To apply ideas from Riemannian Optimization, we need an accurate approximation of the manifold. Our framework addresses this by *learning* the optimizer directly from noisy data via an online algorithm. The method uses the manifold's low dimensional structure and learns local linear models to facilitate movement in the tangent direction. Another challenge is that optimization methods converge to critical points, including suboptimal local minimizers. To address this, we build tunnels (Figure 4) to allow escape from each local minimizer. When stuck at a suboptimal local minimizer, we select the tunnel which would bring us closer to the target point, an idea reminiscent of graph-based nearest neighbor search (Malkov & Yashunin, 2018).

Our method, with high probability, achieves near optimal denoising error $\|\widehat{x} - x_{\natural}\| \lesssim \kappa\sigma\sqrt{d}$ with test-time computational cost $O\left(C(\mathcal{M}, \epsilon_1, \delta)(Dd + e^{cd}) + D \times \$_{c'\tau_{\mathcal{M}}}^{\eta}(\mathcal{M})\right)$, where $\$_{r}^{\eta}(\mathcal{M})$ is a novel complexity measure which quantifies the cost of escaping local minimizers, $C(\mathcal{M}, \epsilon_1, \delta)$ depends on the diameter and curvature of $\mathcal{M}$, the step size $\delta$ and stopping tolerance $\epsilon_1$, and $c, c'$ are numerical constants.

## 2. Relationship to the Literature

Denoising has a long history in signal processing and machine learning, evolving from early statistical techniques to modern deep learning methods. Traditional denoising techniques leverage structural assumptions about the clean signal $x_{\natural}$. For smooth signals, Fourier-based methods suppress high-frequency noise (Wiener, 1949). Sparsity assumptions have given rise to wavelet shrinkage and dictionary learning techniques (Donoho, 1995; Elad & Aharon, 2006). In images with self-similarity, methods such as nonlocal means and BM3D leverage repetitive patterns to enhance denoising performance (Buades et al., 2005; Dabov et al., 2007). Low-dimensional subspace methods, e.g.,principal component analysis and the Karhunen–Loève transform, approximate signals using fewer basis components to filter out noise (Does et al., 2019; Aysin et al., 1998).

Recent developments in signal reconstruction and generation have placed denoisers in a more central role: an accurate denoiser serves as an implicit model for clean signals. The plug-and-play framework leverages denoisers as implicit priors within iterative optimization, enabling flexible and efficient reconstruction across tasks like deblurring, super-resolution, and inpainting (Venkatakrishnan et al., 2013; Zhang et al., 2021). A high-quality denoiser not only removes noise but also encodes structural information about the data, making it a powerful tool for complex inverse problems. Recently, diffusion models have demonstrated that iterative denoising can effectively model complex data distributions, emerging as a powerful generative modeling technique (Ho et al., 2020; Song & Ermon, 2019).

With the advance of deep learning, generic ML architectures such as FCNNs, CNNs (Ilesanmi & Ilesanmi, 2021), transformers (Yao et al., 2022), or the U-Net (Fan et al., 2022) have demonstrated strong denoising capabilities by learning to approximate denoising functions, a concept known as learning-to-denoise. Their effectiveness largely stems from their ability to capture underlying low-dimensional structures in data—an idea explicitly leveraged in autoencoder-based methods (Vincent et al., 2008). However, these models do not directly incorporate the low-dimensional structure of the data, leading to considerable inefficiency in computation and scalability. Furthermore, as neural networks function are black-box models, they often lack theoretical guarantees or clear interpretations.

Given that high-dimensional data often reside on or near a lower-dimensional submanifold (Tenenbaum et al., 2000; Fefferman et al., 2016), incorporating *manifold structure* into denoising tasks — so-called manifold denoising (Hein & Maier, 2006) — has emerged as a rapidly growing area of interest. In the theoretical literature on manifold estimation and denoising (Genovese et al., 2012), the predominant methods are based on local approximation (Fefferman et al., 2020; Yao et al., 2023), and while they offer near-optimal theoretical estimation and denoising guarantees, their test-time efficiency suffers — each new test sample requires a linear scan of the entire dataset. Likewise, many empirically effective methods (e.g., (Hein & Maier, 2006; Gong et al., 2010)) for manifold denoising rely on nearest neighbor searches, resulting in high test-time cost.

In this work, we develop test-time efficient denoisers using ideas from Riemannian optimization (Absil et al., 2008; Sato, 2021). Standard approaches in this area require $\mathcal{M}$ to be known a-priori. Several recent works (Sober et al., 2020; Sober & Levin, 2020; Shustin et al., 2022) develop manifold optimizers for unknown $\mathcal{M}$, using local affine approximation (via moving least squares). While this approach is inspiring, it encounters the same test-time efficiency issues as the above manifold denoisers, since these approximations are formed on-the-fly by linearly scanning a large dataset.

We develop test-time efficient denoisers by learning Riemannian optimizers over a particular geometric graph, which approximates $\mathcal{M}$. We draw inspiration from graph-based nearest neighbor search (Malkov & Yashunin, 2018), while leveraging the low dimensionality of $\mathcal{M}$ to avoid costly

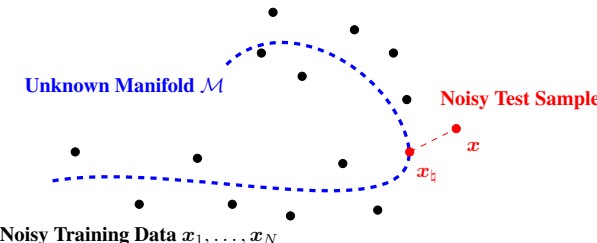

**Figure 1: Problem Setup.** $x_1 = x_{1,\natural} + z_1, \ldots, x_N = x_{N,\natural} + z_N$ are noisy traning samples from an unknown manifold $\mathcal{M} \subset \mathbb{R}^D$. $x = x_\natural + z$ is the noisy test sample.

ambient-space distance calculations.

## 3. Problem Formulation

Our goal is to *learn to denoise* data sampled from a $d$-dimensional submanifold $\mathcal{M}$ of $\mathbb{R}^D$. We observe iid training samples $x_1, \ldots, x_N \in \mathbb{R}^D$ generated as

$$\underset{\text{training sample}}{x_i} = \underset{\text{clean signal}}{x_{i,\natural}} + \underset{\text{noise}}{z_i}, \qquad (3.1)$$

with signal $x_{i,\natural} \sim \mu_\natural$, a distribution supported on $\mathcal{M}$, and noise $z_i \sim_{\text{iid}} \mathcal{N}(0, \sigma^2)$, independent of the signal. Figure 1 illustrates this setup. Our goal is to produce $f : \mathbb{R}^D \to \mathbb{R}^D$ such that for new samples $x = x_\natural + z$ from the same distribution, $f(x_\natural + z) \approx x_\natural$, i.e., $f$ *denoises* $x$. We seek $f$ satisfying the following properties:

**[D1] Provably Accurate Denoising:** near-optimal denoising performance $\mathbb{E}\|f(x) - x_\natural\|_2^2 \lesssim d\sigma^2$, with $d$ the intrinsic dimension of $\mathcal{M}$ and $\sigma$ the noise standard deviation.

**[D2] Fast Evaluation at Test Time:** $f$ can be applied to new samples with computational cost $C(\mathcal{M}) \cdot (D + e^{cd})d$, where $Dd$ is the cost of projecting from $\mathbb{R}^D$ to $\mathbb{R}^d$, and $e^{cd}d$ is the cost of searching a $d$-dimensional manifold.

**[D3] Data-Driven Learning:** $f$ can be learned using only noisy training samples.

Below, we introduce a trainable denoising method based on *manifold optimization*, which achieves **[D1]-[D3]**.

## 4. Denoising and Manifold Optimization

Since the observed signal $x$ is a noisy version of some signal $x_\natural$ on the manifold $\mathcal{M}$, one natural approach to denoising is to *project* $x$ onto $\mathcal{M}$, by solving

$$\min_{q \in \mathcal{M}} \varphi_x(q) \equiv \tfrac{1}{2}\|q - x\|_2^2. \qquad (4.1)$$

the solution $\hat{x}$ to this problem can be interpreted probabilistically as a *maximum likelihood* estimate of $x_\natural$ when the underlying manifold has a uniform distribution; it also accurately approximates the minimum mean squared error

(MMSE) denoiser when $\sigma$ is small.[1] The projection problem (4.1) can be interpreted as a *manifold optimization* problem – we seek to minimize the function $\varphi_x$ over a smooth Riemannian submanifold $\mathcal{M}$ of $\mathbb{R}^D$.

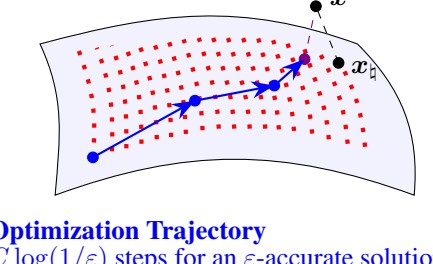

→ **Optimization Trajectory**
$C \log(1/\varepsilon)$ steps for an $\varepsilon$-accurate solution
⋯ **Exhaustive Covering**
$(C/\varepsilon)^d$ points for an $\varepsilon$-accurate solution

**Figure 2: Dimension scaling advantage** of optimization for searching Riemannian manifolds. Brute force search (the core of SOTA provable methods) requires test-time computation *exponential* in intrinsic dimension $d = \dim(\mathcal{M})$.

**Dimension Scaling Advantage of Iterative Optimization.** The optimization problem (4.1) could, in principle, be solved in a variety of ways. One simple approach is to compute $\varphi_x$ on a dense grid of samples $q_1, \ldots, q_M \in \mathcal{M}$, and select the sample $q_i$ with the smallest objective value. As illustrated in Figure 2, such exhaustive search becomes increasingly inefficient as the manifold dimension $d$ increases. While exhaustive search is not a method of choice for solving smooth optimization problems, it plays a critical role in state-of-the-art theoretical manifold denoisers (Yao et al., 2023). At their core is a local approximation of $\mathcal{M}$, formed by selecting near neighbors of $x$ by linearly scanning a dataset $x_1, \ldots, x_N$ which is large enough to densely cover $\mathcal{M}$ – a form of exhaustive search.

A more scalable alternative is to produce $\hat{x}$ by *iterative optimization* – e.g., by gradient descent. The objective function $\varphi_x$ is differentiable, with gradient $\nabla_q \varphi_x = q - x$. The *Riemannian gradient* of $\varphi_x$ at point $q \in \mathcal{M}$ is the projection of $\nabla \varphi_x$ onto the tangent space $T_q\mathcal{M}$ of $\mathcal{M}$ at $q$:

$$\text{grad}[\varphi_x](q) = \mathcal{P}_{T_q\mathcal{M}}\nabla\varphi_x(q). \qquad (4.2)$$

This is the component of the gradient along the manifold $\mathcal{M}$, and a direction of steepest ascent on $\mathcal{M}$. A *Riemannian gradient method* (Absil et al., 2008) steps along $\mathcal{M}$ in the direction of $-\text{grad}[\varphi_x]$, setting

$$q^+ = \exp_q\big(-t\,\text{grad}[\varphi_x](q)\big) \qquad (4.3)$$

---

[1] Indeed, letting $f_{\text{MMSE}}(x) = \arg\min_f \mathbb{E}_{x_\natural,z}\|f(x_\natural + z) - x_\natural\|_2^2$ denote the MMSE denoiser, we have that $\|f_{\text{MMSE}}(x) - \mathcal{P}_\mathcal{M}[x]\|_2 = O(\sigma)$, on a set of $x$ of measure $1 - O(\sigma)$. Other statistical criteria, such as *maximum a-posteriori (MAP)* also lead to manifold optimization problems: in the Bayesian setting in which we have a prior density $\rho_\natural : \mathcal{M} \to \mathbb{R}$ on clean signals, the MAP estimate minimizes $\frac{1}{2}\|q - x\|_2^2 - \lambda \log \rho_\natural(q)$ over $\mathcal{M}$.

where $t > 0$ is a step size, and $\exp_{\boldsymbol{q}} : T_{\boldsymbol{q}}\mathcal{M} \to \mathcal{M}$ is the exponential map, which takes a direction (tangent vector) $\boldsymbol{v} \in T_{\boldsymbol{q}}\mathcal{M}$ to a new point $\exp_{\boldsymbol{q}}(\boldsymbol{v})$ in $\mathcal{M}$.

As shown in Figure 2, with appropriate $t$, this method converges linearly to the global minimizer $\widehat{\boldsymbol{x}}$, provided it is initialized close enough to $\widehat{\boldsymbol{x}}$ – this means that the method requires $C \log(1/\varepsilon)$ steps to reach an $\varepsilon$-approximation of $\widehat{\boldsymbol{x}}$. Inspired by this observation, we seek test-time efficient denoisers that emulate the gradient iteration (4.3). There are two main challenges in realizing this idea: first, we do not know $\mathcal{M}$ – we only have noisy samples $\boldsymbol{x}_1, \ldots, \boldsymbol{x}_N$. Second, the optimization problem (4.1) can exhibit suboptimal local minimizes; to guarantee the performance of our denoiser, we need to ensure convergence to the global optimizer. Below, we sketch our approach to these challenges, deferring a full construction to Section 6.

**Challenge I: $\mathcal{M}$ is a-priori unknown.** Our approach is to *learn* an approximate Riemannian optimizer from data. We will approximate the manifold with a collection of landmarks $\boldsymbol{q}_1, \ldots, \boldsymbol{q}_M$ [2], which are linked by a geometric graph $G$. As illustrated in Figure 3, we will equip this graph with all of the necessary structure to enable optimization – in particular, an approximation to the tangent space to $\mathcal{M}$ at each landmark, which enables us to approximate the Riemannian gradient, and edge embeddings which enable us to traverse the graph in the negative gradient direction.

**Challenge II: Suboptimal Minimizers.** The distance function $\varphi_{\boldsymbol{x}}(\boldsymbol{q})$ may exhibit suboptimal local minimizers. For example, Figure 4 (left): the point $\boldsymbol{q}$ is a local minimizer of the function $\varphi_{\boldsymbol{x}}(\boldsymbol{q}) = \frac{1}{2}\|\boldsymbol{q} - \boldsymbol{x}\|_2^2$.

In Section 6 below, we will show how to *eliminate suboptimal minimizers* by appropriately modifying the graph $G$ – informally, by adding "tunnels" that allow local descent to escape local minimizers and obtain the global optimum.

## 5. Mixed-Order Optimization over $\mathcal{M}$

As described in the previous section, we build an approximate Riemannian optimizer for $\mathcal{M}$. Our optimizer operates over a collection of landmarks $\boldsymbol{q}_1, \ldots, \boldsymbol{q}_M$. To traverse this set of landmarks, we need to be able to (i) approximate the Riemannian gradient of our objective function $\varphi_{\boldsymbol{x}}$ at a given landmark $\boldsymbol{q}$, and (ii) to choose which landmark $\boldsymbol{q}^+$ to move to next, based on the gradient. The following definition contains the required infrastructure:

**Definition 5.1.** [Tangent Bundle Graph] A tangent bundle graph $G$ on vertices $V = (1, \ldots, M)$ consists of set of undirected first-order edges $E^1 \subseteq V \times V$, where each element is denoted as $u \overset{1}{\leftrightarrow} v$ and
**Landmarks** $Q$: $\boldsymbol{q}_i \in \mathbb{R}^D$ for each vertex $i = 1, \ldots, M$,
**Tangent spaces** $T$: $T_i = \mathrm{span}(\boldsymbol{U}_i)$ [3], with orthonormal basis $\boldsymbol{U}_i \in \mathbb{R}^{D \times d}$, at each vertex $i = 1, \ldots, M$,
**Edge embeddings** $\Xi$: $\boldsymbol{\xi}_{u \to v} = \mathcal{P}_{T_u}(\boldsymbol{q}_v - \boldsymbol{q}_u) \in T_u$ [4], for each first-order edge $u \overset{1}{\to} v \in E^1$, where $\mathcal{P}_{T_u}(\boldsymbol{q}_v - \boldsymbol{q}_u) \doteq \boldsymbol{U}_u^T(\boldsymbol{q}_v - \boldsymbol{q}_u)$.

Based on these objects, we can approximate the Riemannian gradient of $\varphi_{\boldsymbol{x}}$ as $\widehat{\mathrm{grad}}[\varphi_{\boldsymbol{x}}](\boldsymbol{q}_i) = \mathcal{P}_{T_i}(\boldsymbol{q}_i - \boldsymbol{x})$.

**First-order (Gradient) Steps over the Tangent Bundle Graph.** The edge embedding $\boldsymbol{\xi}_{u \to v}$ represents a direction in the tangent space $T_u$ which points from $u$ to $v$, and negative Riemannian gradient $-\widehat{\mathrm{grad}}[\varphi_{\boldsymbol{x}}](\boldsymbol{q}_u)$ at $\boldsymbol{q}_u$ is our desired direction for movement. A very intuitive update rule is simply to move from $u$ to the vertex $u^+$ which satisfies

$$u^+ = \arg\max_{v:u \overset{1}{\to} v} \left\langle -\widehat{\mathrm{grad}}[\varphi_{\boldsymbol{x}}](\boldsymbol{q}_u), \boldsymbol{\xi}_{u \to v} \right\rangle. \qquad (5.1)$$

The test-time cost of computing a gradient step is $O(Dd + d \cdot \deg^1(u))$. Here, $\deg^1(u)$ is the degree of the vertex $u$, i.e., its number of first-order neighbors. The $O(Dd)$ term is the cost of computing the Riemannian gradient, while the $d \cdot \deg^1(u)$ is the cost of searching for a neighbor of $u$ which maximizes the correlation in (5.1).

**Zero-order Edges and Steps.** The gradient method described above efficiently converges to the near-critical point. However, it may get trapped at local minimizers. To ensure global optimality, we propose a *mixed-order method* which takes both first-order steps, based on gradient information, and zero-order steps, based on function values.

We add an additional set of edges $E^0$, which we term *zero-order edges* to the graph $G$. We use the notation[5] $u \overset{0}{\to} v$ if $u$ and $v$ are connected by a zero-order edge. As outlined in Algorithm 1, at each step, our mixed order method first attempts a gradient step, by selecting the first-order neighbor whose edge embedding is best aligned with the negative gradient. If this step does not decrease the objective value, the algorithm then performs a zero-order step, by choosing

---

[2] The points shown in Figure 3 represent the landmarks, which serve as a discrete approximation of the unknown manifold. Together with connecting edges, they form a structured domain for optimization. All components – landmarks, edges, and related quantities – are learned directly from noisy data via our online learning algorithm described in Section 6.

[3] We use $T_i$, $T_{\boldsymbol{q}_i}$ interchangeably to denote the tangent space at landmark $\boldsymbol{q}_i$ or vertex $i$. We also use $T_{\boldsymbol{q}_i}\mathcal{M}$ when $\boldsymbol{q}_i \in \mathcal{M}$.

[4] In the language of Riemannian geometry, the $\boldsymbol{\xi}_{u \to v}$ are intended to represent the *logarithmic map* $\log_{\boldsymbol{q}_u}(\boldsymbol{q}_v)$.

[5] In the paper, we use $u \overset{0}{\to} v$ and $\boldsymbol{q}_u \overset{0}{\to} \boldsymbol{q}_v$ interchangeably to denote the zero-order edge from landmark $\boldsymbol{q}_u$ (vertex $u$) to landmark $\boldsymbol{q}_v$ (vertex $v$).

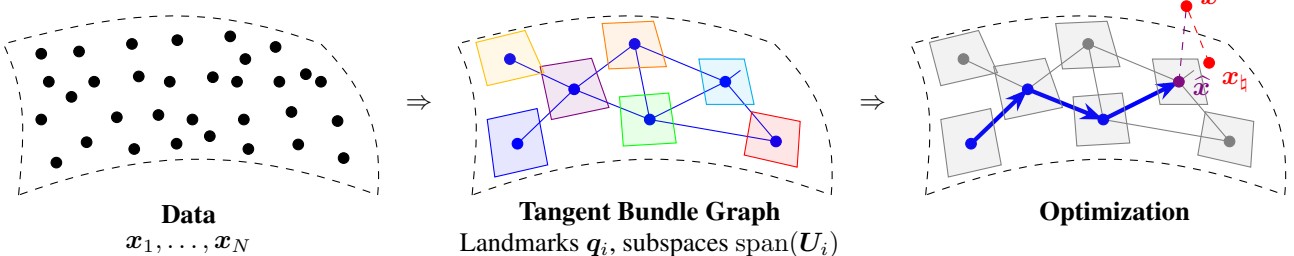

**Data**
$x_1, \ldots, x_N$

**Tangent Bundle Graph**
Landmarks $q_i$, subspaces $\mathrm{span}(U_i)$

**Optimization**

Figure 3: **Learning a Manifold Optimizer from Samples.** Given raw data samples $x_1, \ldots, x_N$ (left), we construct an approximation (center) to $\mathcal{M}$ which consists of landmarks $q_1, \ldots, q_M$, approximate tangent spaces, and a geometric graph $G$ whose vertices are the landmarks. Right. We approximately optimize over $\mathcal{M}$ by optimizing over the graph $G$.

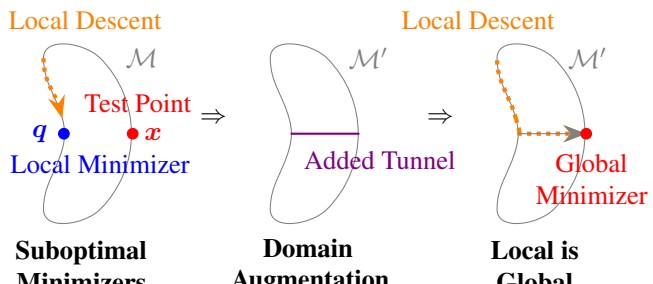

Figure 4: **Eliminating Suboptimal Minimizers by Adding Tunnels.** Consider a test point $x$, with corresponding objective function $\varphi_x(q) = \|q - x\|_2^2$. Left: point $q$ is a local minimizer of $\varphi_x(q)$ over $\mathcal{M}$. Center: We modify the domain $\mathcal{M}$ to connect $q$ and $x$ – adding a "tunnel" connecting these points. Right: local descent over the augmented domain $\mathcal{M}'$ converges to the *global minimizer $x$*.

the zero-order neighbor with smallest objective value:

$$u^+ = \arg \min_{v:u \xrightarrow{0} v} \varphi_x(v). \tag{5.2}$$

This operation requires us to compute the objective function $\varphi_x$ at each of the zero-order neighbors $v$ of $u$. Thus, the computational cost is $O(D \deg^0(u))$. When $D$ and $\deg^0(u)$ are large, the cost of a zero-order step is significantly larger than that of a first-order step – this is why our method prioritizes first-order steps. However, zero-order steps are essential to guarantee global optimality. In the next section, we will show how to construct $G$ to ensure that the mixed-order method converges to a global optimum.

## 6. Learning to Optimize over $\mathcal{M}$

The proposed mixed-order method can efficiently navigate the manifold $\mathcal{M}$. However, $\mathcal{M}$ is a-priori unknown and only noisy samples are available. We next propose an *online learning* method (Algorithm 2), that learns a mixed-order optimizer directly from noisy data.

Our online learning algorithm produces a set of landmarks $Q = \{q_i\}$, tangent space $T_i$ and edge embeddings $\Xi_i$ at

each landmark $q_i$, first-order edges $E^1$ and zero-order edges $E^0$, which have been previously described in Section 5.1. The algorithm processes incoming data sequentially. For each new noisy data point $x$, we perform mixed-order manifold traversal (Algorithm 1) using the existing traversal network $(Q, T, \Xi, E^0, E^1)$. Manifold traversal outputs a vertex $i$, which corresponds to a landmark $q_i$ which locally minimizes the squared distance $\varphi_x(q) = \frac{1}{2}\|q - x\|_2^2$. The resulting vertex $i$ is taken as an input to Algorithm 2.

Depending on $\varphi_x(q_i)$, we encounter one of three scenarios:

- *Inlier*: Landmark $q_i$ is sufficiently close to $x$ (i.e., $\|q_i - x\| \le R(i)$). The noisy point $x$ is denoised using the local model at $q_i$ by setting $\hat{x} = q_i + P_{T_{q_i}}(x - q_i)$. The point $x$ is used to update the local model, by updating the landmark $q_i$ (as a running average), tangent space $T_{q_i}$ (using incremental PCA – see Appendix D), and edge embeddings, setting $\xi_{ij} = P_{T_{q_i}}(q_j - q_i) \quad \forall i \xrightarrow{1} j \in E^1$.

- If $\|q_i - x\| > R(i)$, we perform exhaustive search, scanning all landmarks to find $q_{i_\star}$, the *global* minimizer. Based on $\|q_{i_\star} - x\|$, we distinguish between two cases:

  - $q_i$ *is a suboptimal local minimizer*: If $\|q_{i_\star} - x\| \le R(i_\star)$, i.e., $q_{i_\star}$ is close enough to $x$, we build a tunnel $i \xrightarrow{0} i_\star$ from $q_i$ to $q_{i_\star}$, use $q_{i_\star}$ to denoise $x$, and use $x$ to update the local model at $q_{i_\star}$.

  - $x$ *is an outlier*: No existing landmark is close to $x$. We make $x$ a new landmark $q_M$, and build first-order edges $M \xleftrightarrow{1} j$ when $\|q_M - q_j\| \le R_{\mathrm{nbrs}}$, and initialize a local model at $q_M$.

As more samples are grouped into this landmark, the cumulative effect of noise diminishes, gradually reducing both the landmark's deviation from the true manifold and the error in its tangent space estimation. The threshold $R(i)$ for accepting inlying data points $x$ is allowed to vary with the number of data points assigned to a given landmark $q_i$ (see Section 8 and Appendix F).

By processing one sample at a time, the online learning approach distributes the computational cost of training over

---

**Algorithm 1** `ManifoldTraversal`

---

**Input:** Network $G$, $\boldsymbol{x} \in \mathbb{R}^D$, initial vertex $i \leftarrow 1$
**while** not converged **do**
$\quad \boldsymbol{g} \leftarrow \boldsymbol{U}_i^*(\boldsymbol{x} - \boldsymbol{q}_i)$
$\quad i^\sharp \leftarrow \arg\max_{j:i \xrightarrow{1} j} \langle \boldsymbol{g}, \boldsymbol{\xi}_{ij} \rangle.$
$\quad$ **if** $\|\boldsymbol{q}_{i^\sharp} - \boldsymbol{x}\| < \|\boldsymbol{q}_i - \boldsymbol{x}\|$ **then**
$\quad\quad i \leftarrow i^\sharp$
$\quad$ **else**
$\quad\quad i \leftarrow \arg\min_{j:i \xrightarrow{0} j} \|\boldsymbol{q}_j - \boldsymbol{x}\|$
$\quad$ **end if**
**end while**
**Output:** $i$

---

time and ensures memory efficiency, enabling it to adapt to large and high-dimensional datasets.

After seeing enough samples, Algorithm 2 creates a set of landmarks $\boldsymbol{Q}$, which forms a discrete approximation of the manifold $\mathcal{M}$, along with a geometric graph that captures both the local geometry of the manifold and its global connectivity (Figure 6). This structure enables efficient and accurate navigation for a new noisy sample at test time.

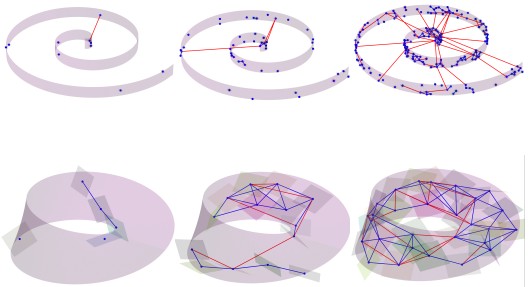

Figure 5: **Growing Traversal Networks for Synthetic Manifolds.** Growing manifold traversal networks on the Swiss roll and Möbius strip at three different times during early training. First-order edges (blue) allow navigation along $\mathcal{M}$; zero-order edges (red) ensure global optimality.

## 7. Theoretical Analysis

Our main theoretical result shows that the proposed mixed-order traversal method rapidly converges to a near-optimal denoised signal. We study the behavior of this method on a noisy input $\boldsymbol{x} = \boldsymbol{x}_\natural + \boldsymbol{z}$, with $\boldsymbol{x}_\natural$ an arbitrary element of $\mathcal{M}$, and $\boldsymbol{z} \sim_{\text{iid}} \mathcal{N}(0, \sigma^2)$. Here, the goal is to produce an output $\widehat{\boldsymbol{x}} \approx \boldsymbol{x}_\natural$ – in particular, we would like to achieve $\|\widehat{\boldsymbol{x}} - \boldsymbol{x}_\natural\| \lesssim \sigma\sqrt{d}$, which is optimal for small $\sigma$.

Our analysis assumes access to an accurate collection of landmarks $\boldsymbol{Q} = \{\boldsymbol{q}_1, \ldots, \boldsymbol{q}_M\} \subset \mathcal{M}$ and their tangent spaces $T_{\boldsymbol{q}_i}$, as well as appropriately structured first-order and zero-order edge sets $E^1$ and $E^0$ – in a nutshell, we prove that given an appropriately structured traversal network,

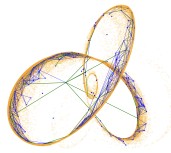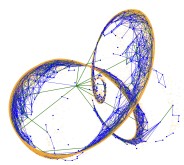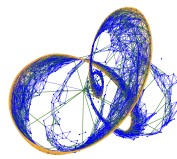

Figure 6: **Growing Traversal Networks from Scientific Data.** Visualization of 2048-dimensional gravitational waves. We show clean samples (orange), landmarks (blue dots), first-order edges (blue), and zero-order edges (green). As the online algorithm sees more points, it learns an increasingly good approximation to $\mathcal{M}$. Number of training points: Left: 1,000. Middle: 10,000. Right: 100,000.

---

**Algorithm 2** `OnlineLearningForManifoldTraversal`

---

**Input:** Current network $G$, $\boldsymbol{x} \in \mathbb{R}^D$.
$i \leftarrow$ `ManifoldTraversal`$(G, \boldsymbol{x})$
**if** $\|\boldsymbol{q}_i - \boldsymbol{x}\|_2 \leq R(i)$ **then**
$\quad$ Denoise via $\widehat{\boldsymbol{x}} \leftarrow \mathcal{P}_{\boldsymbol{q}_i + T_i} \boldsymbol{x}$
$\quad$ Update local parameters $q_i, T_i, \Xi_i$
**else**
$\quad i_\star \leftarrow \arg\min_i \|\boldsymbol{q}_i - \boldsymbol{x}\|_2$
$\quad$ **if** $\|\boldsymbol{q}_{i_\star} - \boldsymbol{x}\|_2 \leq R(i_\star)$ **then**
$\quad\quad \boldsymbol{q}_i$ is a local min, add a zero-order edge $i \xrightarrow{0} i_\star$
$\quad\quad$ Denoise $\boldsymbol{x}$, update $\boldsymbol{q}_{i_\star}, T_{i_\star}, \Xi_{i_\star}$
$\quad$ **else**
$\quad\quad$ Create a new landmark $\boldsymbol{q}_M \leftarrow \boldsymbol{x}$
$\quad\quad$ Connect it to neighboring landmarks within $R_{\text{nbrs}}$
$\quad\quad$ Initialize local parameters $T_M, \Xi_M$
$\quad$ **end if**
**end if**
**Output:** $G = (V, E)$

---

mixed-order traversal is both *accurate* and highly *efficient*, corroborating the conceptual picture in Figure 2.

We analyze a version (Algorithm 3) of the mixed-order method, which consists of three phases: a first-order Phase I, which, starting from an arbitrary initialization, produces an approximate critical point $\boldsymbol{q}_{i_I}$, a zero-order Phase II, which jumps to a point $\boldsymbol{q}_{i_{II}}$ in a $c\tau_\mathcal{M}$ neighborhood of the ground truth $\boldsymbol{x}_\natural$, followed by a first-order Phase III, produces a point $\boldsymbol{q}_{i_{III}}$ within distance $C\kappa\sigma\sqrt{d}$ of $\boldsymbol{x}_\natural$.

**Complexity of Escaping Suboptimal Minimizers.** A key element of Algorithm 3 (and more generally Algorithm 1) is the use of *zero-order edges* (or tunnels) to escape suboptimal critical points. The complexity of this step of the algorithm is dictated by the number of zero-order edges emanating from the point $\boldsymbol{q}_{i_I}$. There is a clear geometric interpretation to this number, which is illustrated in Figure 7: a point $\boldsymbol{q} \in \mathcal{M}$ is a critical point of the distance function

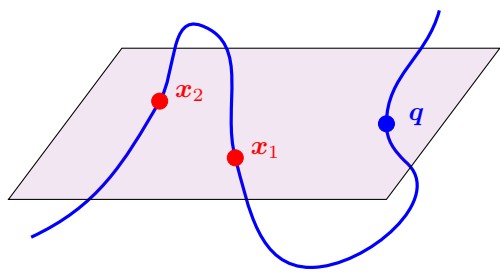

**Normal Space** $q + N_q \mathcal{M}$

$q$ is a *critical point* of $\varphi_{\boldsymbol{x}_1} = \frac{1}{2}\|\boldsymbol{q} - \boldsymbol{x}_1\|^2$ and $\varphi_{\boldsymbol{x}_2} = \frac{1}{2}\|\boldsymbol{q} - \boldsymbol{x}_2\|^2$ over $\mathcal{M}$

Figure 7: **Critical Points of the Distance Function.** The point $q$ is a critical point of the distance $\varphi_{\boldsymbol{x}}(\boldsymbol{q}) = \frac{1}{2}\|\boldsymbol{q} - \boldsymbol{x}\|_2^2$ for any point $\boldsymbol{x}$ satisfying $\boldsymbol{x} - \boldsymbol{q} \in N_q \mathcal{M}$. The *vista number* $\$(\mathcal{M})$ bounds the number of $\boldsymbol{x}$ for which this is true – i.e., the number of $\boldsymbol{x}$ for which $\boldsymbol{q}$ is a local minimizer. This in turn bounds the number of tunnels which must be added to ensure that local descent converges to a global optimizer. In the example illustrated here, $\$(\mathcal{M}) = 3$.

$\varphi_{\boldsymbol{x}}(\boldsymbol{q}) = \frac{1}{2}\|\boldsymbol{x} - \boldsymbol{q}\|_2^2$ if and only if $\boldsymbol{x} - \boldsymbol{q} \in N_q \mathcal{M}$. Hence, the number of (clean) target points $\boldsymbol{x}_\natural \in \mathcal{M}$ for which $\boldsymbol{q}$ is a critical point is given by the number of intersections of $\mathcal{M}$ with the normal space $\boldsymbol{q} + N_q \mathcal{M}$. Inspired by the geometry of this picture, we denote this quantity $\$(\mathcal{M})$:

$$\$(\mathcal{M}) = \max_{\boldsymbol{q} \in \mathcal{M}} \# \left[\mathcal{M} \cap (\boldsymbol{q} + N_q \mathcal{M})\right]. \tag{7.1}$$

Because Phase I of our algorithm produces *approximate* critical points, we work with a stable counterpart to this quantity: let $N_q^\eta \mathcal{M} = N_q \mathcal{M} + B(0, \eta)$ denote an $\eta$ dilation of the normal space at $\boldsymbol{q}$. We set

$$\$_r^\eta(\mathcal{M}) = \max_{\boldsymbol{q} \in \mathcal{M}} \mathfrak{N}\left((\boldsymbol{q} + N_q^\eta \mathcal{M}) \cap \mathcal{M}, d_{\mathcal{M}}(\cdot, \cdot), r\right) \tag{7.2}$$

where $\mathfrak{N}(S, \rho, r)$ denotes the covering number of set $S$ in metric $\rho$ with covering radius $r$. Intuitively, this counts the "number of times" the manifold intersects the dilated normal space $N^\eta$. As we will establish in Theorem 7.1, this quantity upper bounds the number of zero-order edges at each landmark (i.e. $\deg^0(\boldsymbol{q})$) required for global optimality.

**Main Result.** Our main result is as follows:

**Theorem 7.1.** *Let $\mathcal{M} \subset \mathbb{R}^D$ be a complete and connected $d$-dimensional manifold whose extrinsic geodesic curvature is bounded by $\kappa$. Assume $\kappa \operatorname{diam}(\mathcal{M}) \geq 1$ and $\sigma\sqrt{D} \leq c_1 \tau_{\mathcal{M}}$.*
*Assumptions on $Q$: The landmarks $Q = \{\boldsymbol{q}_1, \ldots, \boldsymbol{q}_M\} \subset \mathcal{M}$ are $\delta$-separated, and form a $\delta$-net for $\mathcal{M}$, under the metric $d_{\mathcal{M}}(\cdot, \cdot)$. Assume $\delta \leq \operatorname{diam}\mathcal{M}$.*
*Assumptions on $E^1$: First-order graph $E^1$ is defined such that $u \overset{1}{\leftrightarrow} v \in E^1$ when $\|\boldsymbol{q}_u - \boldsymbol{q}_v\|_2 \leq R_{nbrs}$. Assume*

$a\delta = R_{nbrs} \leq c_2 \sigma\sqrt{d}$ *for some $a \geq 40$.*
*Assumptions on $E^0$: $E^0$ is a minimal collection of edges satisfying the following covering property: for distinct $\boldsymbol{q}, \boldsymbol{q}' \in Q$, if $\boldsymbol{q}' \in \boldsymbol{q} + N_q^\eta \mathcal{M}$ with*

$$\eta \geq \epsilon_1 + c_4 \sigma\sqrt{d}\sqrt{\kappa \operatorname{diam}(\mathcal{M}) + \log(\delta^{-1}\operatorname{diam}(\mathcal{M}))},$$

*there exists a zero-order edge $\boldsymbol{q} \overset{0}{\to} \boldsymbol{q}''$ with $\boldsymbol{q}'' \in B_{\mathcal{M}}(\boldsymbol{q}', c_5 \tau_{\mathcal{M}})$.*

*With high probability in the noise $\boldsymbol{z}$, Algorithm 3 with parameters*

$$R_a = R_{nbrs} - \delta,$$
$$\epsilon_1 > R_{nbrs}\Big(c_6 \kappa \operatorname{diam}(\mathcal{M}) +$$
$$c_7 \kappa \sigma d^{1/2}\sqrt{\kappa \operatorname{diam}(\mathcal{M}) + \log(\delta^{-1}a \operatorname{diam}(\mathcal{M}))}\Big),$$
$$\epsilon_2 > c_8 \max\{\kappa, 1\}\sigma\sqrt{d},$$

*produces an output $\boldsymbol{q}_\star$ satisfying $d_{\mathcal{M}}(\boldsymbol{q}_\star, \boldsymbol{x}_\natural) \leq 2\epsilon_2$ with an overall number of arithmetic operations bounded by*

$$O\left((D + e^{c' \log(a)d})\left(\frac{\operatorname{diam}^2(\mathcal{M})}{\epsilon_1 \delta} + \frac{1}{\kappa\delta}\right)d + D \times \$_{c_6 \tau_{\mathcal{M}}}^\eta(\mathcal{M})\right).$$

Here, $\tau_{\mathcal{M}}$ is the *reach* of $\mathcal{M}$, i.e., the radius of the largest tubular neighborhood of $\mathcal{M}$ where the projection $\mathcal{P}_{\mathcal{M}}\boldsymbol{x}$ is unique (Federer, 1959). The assumption $\sigma\sqrt{D} \leq c_1 \tau_{\mathcal{M}}$ ensures that, with high probability $\|\boldsymbol{z}\| \leq \tau_{\mathcal{M}}$, the projection $P_{\mathcal{M}}(\boldsymbol{x})$ is close to $\boldsymbol{x}_\natural$ in the geodesic distance $d_{\mathcal{M}}$.

The extrinsic geodesic curvature $\kappa$ is the supremum of $\|\ddot{\gamma}\|$ over all unit speed geodesics $\gamma(t)$ on $\mathcal{M}$. This quantity measures how "curvy" geodesics in $\mathcal{M}$ are. $\operatorname{diam}(\mathcal{M})$ is the diameter of $\mathcal{M}$ in the intrinsic distance $d_{\mathcal{M}}$. $\$_{c_6 \tau_{\mathcal{M}}}^\eta(\mathcal{M})$ upper bounds the number of zero-order edges per landmark.

**Interpretation.** This result shows that given accurate landmarks, tangent spaces, and first-order and zero-order graphs, the algorithm converges to a $\max\{\kappa, 1\}\sigma\sqrt{d}$ neighborhood of $\boldsymbol{x}_\natural$, which is best achievable up to constant when $\kappa$ is bounded. The algorithm admits an upper bound on the required number of arithmetic operations. $(Dd + e^{c' \log(a)d}d)$ is the computational cost of taking one first-order step, as $Dd$ comes from projection of the $D$-dimensional gradient into the $d$-dimensional tangent space, and $de^{c' \log(a)d}$ represents the cost of comparing the $d$-dimensional dot product between the negative Riemannian gradient with all first-order edge embeddings, with number of neighbors bounded by $e^{c' \log(a)d}$.
$\left(\frac{\operatorname{diam}^2(\mathcal{M})}{\epsilon_1 \delta} + \frac{1}{\kappa\delta}\right)$ represent the total number of first order steps taken. $\frac{\operatorname{diam}^2(\mathcal{M})}{\epsilon_1 \delta}$ represents the number of steps on the path from $\boldsymbol{q}_{i^0}$ (initialization) to $\boldsymbol{q}_{i_1}$. Since the initialization could be arbitrarily bad, in this phase we can only guarantee decrease in the value of $\varphi_{\boldsymbol{x}}(\boldsymbol{q})$, so naturally $\operatorname{diam}^2(\mathcal{M})$

**Algorithm 3** `101Traversal`

---

**Input:** Network $G, \boldsymbol{x} \in \mathbb{R}^D, R_a, \epsilon_1, \epsilon_2$
Initialization $i$ {Phase I}
**while** $\|\mathcal{P}_{T_i}(\boldsymbol{x} - \boldsymbol{q}_i)\|_2 > \epsilon_1$ **do**
   $i \leftarrow \underset{j:i \overset{1}{\to} j}{\arg\min} \left\|\mathcal{P}_{B(0,R_a)}\mathcal{P}_{T_i}(\boldsymbol{x} - \boldsymbol{q}_i) - \mathcal{P}_{T_i}(\boldsymbol{q}_j - \boldsymbol{q}_i)\right\|_2$
**end while**
$i_{\mathrm{I}} \leftarrow i$
$i_{\mathrm{II}} \leftarrow \underset{j:i_{\mathrm{I}} \overset{0}{\to} j}{\arg\min} \|\boldsymbol{q}_j - \boldsymbol{x}\|_2$ {Phase II}
$i \leftarrow i_{\mathrm{II}}$ {Phase III}
**while** $\|P_{T_i}(\boldsymbol{x} - \boldsymbol{q}_i)\|_2 > \epsilon_2$ **do**
   $i \leftarrow \underset{j:i \overset{1}{\to} j}{\arg\min} \left\|\mathcal{P}_{B(0,R_a)}\mathcal{P}_{T_i}(\boldsymbol{x} - \boldsymbol{q}_i) - \mathcal{P}_{T_i}(\boldsymbol{q}_j - \boldsymbol{q}_i)\right\|_2$
**end while**
$i_{\mathrm{III}} \leftarrow i$
**Output:** $\boldsymbol{q}_{i_{\mathrm{III}}}$

---

captures the worst case initialization. The $\epsilon_1 \delta$ represents the minimal decrease in each step: since the landmarks $Q$ forms a $\delta$-net of the manifold, each gradient direction is approximately covered, and each first order step have gradient norm at least $\epsilon_1$, by definition of the stopping criterion. On the other hand, $\frac{1}{\kappa\delta}$ represents the number of operations from $\boldsymbol{q}_{i_{\mathrm{II}}}$ to $\boldsymbol{q}_{i_{\mathrm{III}}}$. By construction $\boldsymbol{q}_{i_{\mathrm{II}}} \in B_{\mathcal{M}}(\boldsymbol{x}_\natural, c_5 \tau_{\mathcal{M}})$, so when we consider $d_{\mathcal{M}}(\boldsymbol{x}_\natural, \cdot)$ as the new objective function, the worst case initialization is $\frac{1}{\kappa}$ and on this scale each gradient step is guaranteed to walk along the manifold, giving us a $\delta$-decrease in the intrinsic distance to $\boldsymbol{x}_\natural$.

Finally, the $D \times \$_{c_6 \tau_{\mathcal{M}}}^{\eta}(\mathcal{M})$ represents the cost of the zero order step from $\boldsymbol{q}_{i_1}$, the $\epsilon_1$-approximate critical point for $\varphi_{\boldsymbol{x}}$, to $\boldsymbol{q}_{i_{\mathrm{II}}}$, the point that lies intrinsically close to $\boldsymbol{x}_\natural$. $\$_{\tau_{\mathcal{M}}}^{\eta}(\mathcal{M})$ is a new geometric quantity that we've defined, and it captures how much $\mathcal{M}$ intersects its own dilated normal space. Intuitively larger $\$_{\tau_{\mathcal{M}}}^{\eta}(\mathcal{M})$ means one would expect more local minimizes while performing first order descent. Notably $\$_{\tau_{\mathcal{M}}}^{\eta}(\mathcal{M})$ can be exponential in $d$ in the worst case manifold, in which case our algorithm behaves similarly to nearest neighbor search. Intuitively, the parameter $\epsilon_1$ (and corresponding requirement $\eta$ on the zero order edges) cuts out a tradeoff between the complexity of the first order phases and the complexity of the zero order phase.

# 8. Simulations and Experiments

In this section, we visualize the traversal networks constructed using Algorithm 2 across synthetic manifolds and high-dimensional scientific data. Our experiments show that denoising performance of Algorithm 2 improves with increased number of training data, indicating that the algorithm effectively learns to denoise. Based on experiments with gravitational wave signals, we further demonstrate that Algorithm 2 achieves a better test-time complexity and accu-

racy tradeoff compared to Nearest Neighbor over the same set of landmarks. Additional experiments on image datasets and comparisons with autoencoders for gravitational wave denoising are detailed in Appendix G.

**Traversal Network Construction for Various Manifolds.** Algorithm 2 grows the manifold traversal network, processing one sample at a time and *learning tangent spaces and landmarks* in the process. Figure 5 reveals the graph construction process during early training for the Swiss roll and Mobius strip (intrinsic $d = 2$, ambient $D = 3$). The figure provides visual confirmation of the fact that first-order edges (blue) capture local information (clearly visible in the Möbius strip), and zero-order edges (red) create tunnels that help escape local minimizers (clearly visible in the Swiss roll).

In addition to synthetic manifolds in $\mathbb{R}^3$, we also grow traversal networks on high-dimensional real-world data. We learn a denoiser on a dataset of 100,000 noisy gravitational waves (Abramovici et al., 1992; Aasi et al., 2015) using the *online method* as described in Algorithm 2. Data are $D = 2048$-dimensional, with intrinsic dimension $d = 2$, depicted in Figure: 6. We refer the reader to the Appendix B.1 for data generation details.

**Improvement of Denoising Performance with Streaming Data.** We measure the performance of our learned denoiser on a dataset of 100,000 noisy gravitational waves. Figure 8 shows the training error of the learned denoiser. The training error across the first $n$ data points is given by

$$\mathrm{MSE} = \frac{1}{n}\sum_{i=1}^{n} \|\widehat{\boldsymbol{x}}_i - \boldsymbol{x}_{i,\natural}\|_2^2 \qquad (8.1)$$

where $\widehat{\boldsymbol{x}}_i$ is the denoised point, and $\boldsymbol{x}_{i,\natural}$ is the ground truth. We plot the theoretical lower bound as $\sigma^2 d$ and see that the denoiser error decreases, showing potential to converge to the optimal theoretical lower bound.

**Tradeoff: Test-time efficiency vs. Denoising Performance.** We investigate the tradeoffs between performance and complexity for our mixed order method, and baselines such as nearest neighbor (NN) search. After obtaining a traversal network $(Q, T, \Xi, E^0, E^1)$, we compare using the following experimental setup. For NN and our method, we search over the same set of landmarks $Q$. We measure accuracy in mean squared error (same metric as in (8.1)) and complexity in number of multiplications. An important parameter in Algorithm 2 is the denoising radius $R(i)$ which controls the complexity by determining the number of landmarks created.

Conceptually, $R(i)$ measures distance between a noisy point $\boldsymbol{x}$ and the the landmark that best describes it. Throughout

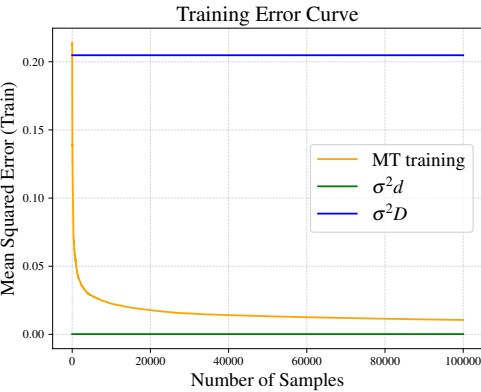

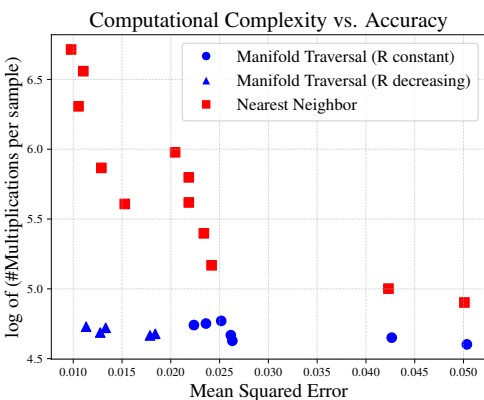

Figure 8: **Training Error:** Training error decreases with the number of training points. We train a denoiser with 100,000 noisy samples with parameters $R(i), R_{\text{nbrs}}$. The error curve shows the potential to approach the theoretical optimal $\sigma^2 d$.

Algorithm 2, the landmark error decreases, and $R(i)$ should be reduced accordingly. Hence, we define a general formula for $R(i)$ as follows:

$$R(i) = \sqrt{\sigma^2 D + \sigma^2 D/N_i + \sigma^2 d} \qquad (8.2)$$

where the first error term $\sigma^2 D$ comes from noisy points $x_i$, and the second term $\sigma^2 D/N_i$ comes from the fact that there is distance between landmarks and the true manifold. Initially, a landmark $q_i$ is created using one noisy point $x$. As more and more points are used to update landmark $q_i$ and other local parameters at vertex $i$, local approximation gets more and more accurate, and the distance between the landmark and true manifold should decrease. This is why we divide the error $\sigma^2 D$ by $N_i$, the number of points used to update landmark $q_i$ and other local parameters, making $R(i)$ smaller. Lastly, $\sigma^2 d$ term comes from the error $\|P_{\mathcal{M}} x - x_\natural\|_2^2$ across the manifold $\mathcal{M}$ – see Appendix F.

Figure 9 summarizes test-time accuracy versus complexity of the proposed mixed-order method, comparing it to nearest neighbor search. We do this comparison based on a test set of $20,000$ noisy points. By varying $R(i)$ and $R_{\text{nbrs}}$ (see Table 1 in the Appendix for details), we obtain twelve different networks $\{(Q_j, T_j, \Xi_j, E_j^0, E_j^1)\}_{j=1}^{12}$. We compare our method of network $j$ with nearest neighbor search over the same set of landmarks $Q_j$. The details can be found in Appendix G. As shown in Figure 9, manifold traversal achieves significantly better complexity-accuracy trade-offs compared to nearest neighbor search. Figure 9 illustrates that decreasing $R(i)$, as the number of samples $N_i$ assigned to landmark $q_i$ increases, leads to better accuracy compared to maintaining a constant radius.

## 9. Conclusions

We introduced a novel framework for test-time efficient and accurate manifold denoising when the manifold is un-

Figure 9: **Test-Time Complexity-Accuracy Tradeoff of Mixed-order Method versus Nearest Neighbor.** Over a test set of $20,000$ noisy points, our proposed mixed-order method achieves better tradeoffs compared to nearest neighbor search over the same set of landmarks.

known and only noisy samples are given. The framework incorporates an *online learning* method to construct an augmented graph for optimizing on the approximated manifold, and a *mixed-order* method that ensures both efficiency and global optimality. Our experiments on scientific manifolds demonstrate that the proposed methods achieve a superior complexity-accuracy tradeoff vis. nearest neighbor search, which is the core of many existing provable denoising approaches. Furthermore, our analyses show that the mixed-order method attains near-optimal denoising performance, assuming the online learning method produces an ideal graph, and we provide complexity analyses for the mixed-order method under this assumption.

A promising future direction is to establish theoretical guarantees for the accuracy of the landmarks generated by the online learning method, as they play a crucial role in denoising performance. The current learning method dynamically builds edges in the graph as needed. Another potential avenue for future research is to develop a sparser network using pruning techniques while maintaining global optimality, which could further improve test-time efficiency. More broadly, we aim to leverage this designed method to study the traversal properties of natural datasets across a wide and diverse range of datasets. Additionally, integrating this method as a denoiser block within signal generation and reconstruction architectures could be a valuable direction, potentially accelerating the entire process.

## Acknowledgements

The authors gratefully acknowledge support from the National Science Foundation, through the grant NSF 2112085. Shiyu Wang also gratefully acknowledges support from the Chiang Chen Industrial fellowship. The authors are grateful for the generous support of Columbia University in the City of New York. The authors would like to highlight the inspirational and community creating role of NSF's CCF-1740391 grant. The authors thank Davit Shadunts for his contributions to refactoring and organizing the codebase used in the final version of the paper.

## Impact Statement

This paper presents work whose goal is to advance the field of Machine Learning. There are many potential societal consequences of our work, none which we feel must be specifically highlighted here.

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

## Appendix

## A. Main Claims

**Theorem A.1.** *Let $\mathcal{M} \subset \mathbb{R}^D$ be a complete and connected $d$-dimensional manifold whose extrinsic geodesic curvature is bounded by $\kappa$.*

- **Assumptions on $\mathcal{M}$** *Assume $\kappa \operatorname{diam}(\mathcal{M}) \geq 1$ and $\sigma\sqrt{D} \leq \frac{1}{640}\tau_{\mathcal{M}}$.*

- **Assumptions on $Q$** *Suppose that the landmarks $Q = \{\boldsymbol{q}_1, \ldots, \boldsymbol{q}_M\} \subset \mathcal{M}$ are $\delta$-separated, and form a $\delta$-net for $\mathcal{M}$, under the metric $d_{\mathcal{M}}(\cdot, \cdot)$. Assume $\delta \leq \operatorname{diam}(\mathcal{M})$.*

- **Assumptions on $E^1$** *First-order graph $E^1$ is defined such that $u \xrightarrow{1} v \in E^1$ when $\|\boldsymbol{q}_u - \boldsymbol{q}_v\|_2 \leq R_{\text{nbrs}}$. Assume $a\delta = R_{\text{nbrs}} \leq \frac{\sqrt{2}}{64}\sigma\sqrt{d}$ for some $a \geq 40$.*

- **Assumptions on $E^0$** *$E^0$ is a minimal collection of edges satisfying the following covering property: for distinct $\boldsymbol{q}, \boldsymbol{q}' \in Q$, if $\boldsymbol{q}' \in \boldsymbol{q} + N_{\boldsymbol{q}}^\eta \mathcal{M}$ with*

$$\eta \geq \epsilon_1 + \delta + 14\sigma\sqrt{d}\sqrt{\kappa\operatorname{diam}(\mathcal{M}) + \log(\operatorname{diam}(\mathcal{M})) - \log(\delta) + \log(7)}, \tag{A.1}$$

*there exists a ZOE $\boldsymbol{q} \xrightarrow{0} \boldsymbol{q}''$ with $\boldsymbol{q}'' \in B_{\mathcal{M}}(\boldsymbol{q}', \frac{1}{80}\tau_{\mathcal{M}})$.*

*With probability at least $1 - 4e^{-\frac{d}{16}} - \left(\frac{e}{2}\right)^{-\frac{D}{2}}$ in the noise $\boldsymbol{z}$, the Algorithm 3 with parameters*

$$R_a = R_{\text{nbrs}} - \delta \tag{A.2}$$

$$\epsilon_1 > \frac{2R_{nbrs}}{0.55}\left(\frac{2}{3} + \frac{8}{3}\kappa\operatorname{diam}(\mathcal{M})+\right. \tag{A.3}$$

$$\left. +16\kappa\sigma\sqrt{d}\sqrt{\kappa\operatorname{diam}(\mathcal{M}) + \log(a) - \log(\delta) + \log(\operatorname{diam}(\mathcal{M})) + \log(100)}\right) \tag{A.4}$$

$$\epsilon_2 > C\max\{\kappa, 1\}\sigma\sqrt{d} \tag{A.5}$$

*produces an output $\boldsymbol{q}_\star$ satisfying*

$$d_{\mathcal{M}}(\boldsymbol{q}_\star, \boldsymbol{x}_\natural) \leq 2\epsilon_2 \tag{A.6}$$

*with an overall number of arithmetic operations bounded by*

$$\left(Dd + d\left(1 + 4\sqrt{2}ae\right)^d\right)\left(\frac{10\operatorname{diam}^2(\mathcal{M}) + 20\sigma^2 D}{\epsilon_1\delta} + \frac{8}{\kappa\delta}\right) + D \times \$_{\frac{1}{160}\tau_{\mathcal{M}}}^\eta(\mathcal{M}) \tag{A.7}$$

*where $\$_r^\eta(\mathcal{M})$ follows the definition in Equation (7.2).*

**Proof.** With the above assumptions and the combination of Proposition A.3, Proposition A.4, and Proposition A.5, it's easy to show that with probability at least $1 - 4e^{-\frac{d}{16}} - \left(\frac{e}{2}\right)^{-\frac{D}{2}}$ in the noise $\boldsymbol{z}$, the output of the algorithm $\boldsymbol{q}_\star$ satisfies

$$d_{\mathcal{M}}(\boldsymbol{q}_\star, \boldsymbol{x}_\natural) \leq 2\epsilon_2, \tag{A.8}$$

and phase I and phase III use at most $\frac{\operatorname{diam}^2(\mathcal{M}) + 2\sigma^2 D}{0.1375\epsilon_1\delta} + \frac{8}{\kappa\delta}$ steps, and phase II takes at most $D \times \$_{\frac{1}{160}\tau_{\mathcal{M}}}^\eta(\mathcal{M})$ operations.

Suppose $S_{\boldsymbol{q},\eta} \subset \mathcal{M}$ is the minimal set that $\frac{1}{160}\tau_{\mathcal{M}}$-covers the $(\boldsymbol{q} + N_{\boldsymbol{q}}^\eta) \cap \mathcal{M}$ under the metric $d_{\mathcal{M}}(\cdot, \cdot)$, with $\eta \geq \epsilon_1 + \delta + \sigma\left(4\sqrt{\log|\mathcal{Q}|} + 10\sqrt{d}\right)$. From the definition that $\$_r^\eta(\mathcal{M}) = \max_{\boldsymbol{q}\in\mathcal{M}}\mathfrak{N}\left((\boldsymbol{q} + N_{\boldsymbol{q}}^\eta\mathcal{M}) \cap \mathcal{M}, d_{\mathcal{M}}(\cdot, \cdot), r\right)$, where $\mathfrak{N}(S, \rho, r)$ denotes the covering number of set $S$ in metric $\rho$ with covering radius $r$, we have $|S_{\boldsymbol{q},\eta}| = \mathfrak{N}\left((\boldsymbol{q} + N_{\boldsymbol{q}}^\eta) \cap \mathcal{M}), d_{\mathcal{M}}(\cdot, \cdot), \frac{1}{160}\tau_{\mathcal{M}}\right) \leq \$_{\frac{1}{160}\tau_{\mathcal{M}}}^\eta$.

For any $\boldsymbol{q} \in \mathcal{Q}$, as the landmark $\mathcal{Q}$ $\delta$-covers the manifold, it $\delta$-covers $S_{\boldsymbol{q},\eta}$. We denote the minimal set of landmarks that $\delta$-cover $S_{\boldsymbol{q},\eta}$ as $\mathcal{Q}_S$. Then for any $\boldsymbol{q}' \in \mathcal{Q}, \boldsymbol{q}' \in \boldsymbol{q} + N_{\boldsymbol{q}}^\eta$, there exists a point $\boldsymbol{s} \in S_{\boldsymbol{q},\eta}$ such that $d_{\mathcal{M}}(\boldsymbol{s}, \boldsymbol{q}') \leq \frac{1}{160}\tau_{\mathcal{M}}$. Since $\boldsymbol{s} \in S_{\boldsymbol{q},\eta}$, there exists a landmark $\boldsymbol{q}'' \in \mathcal{Q}_S$ such that $d_{\mathcal{M}}(\boldsymbol{s}, \boldsymbol{q}'') \leq \delta$.

Therefore, for any $\boldsymbol{q}' \in \boldsymbol{q} + N_{\boldsymbol{q}}^\eta$, there exists $\boldsymbol{q}'' \in \mathcal{Q}_S$ such that

$$d_{\mathcal{M}}(\boldsymbol{q}', \boldsymbol{q}'') \leq d_{\mathcal{M}}(\boldsymbol{q}', \boldsymbol{s}) + d_{\mathcal{M}}(\boldsymbol{s}, \boldsymbol{q}'') \leq \frac{1}{160}\tau_{\mathcal{M}} + \delta \leq \frac{1}{160}\tau_{\mathcal{M}} + \frac{\sqrt{2}}{40*64*640}\tau_{\mathcal{M}} \leq \frac{1}{80}\tau_{\mathcal{M}}, \quad \text{(A.9)}$$

where we used $\delta = \frac{R_{\text{nbrs}}}{a}$, $a \geq 40$, $R_{\text{nbrs}} \leq \frac{\sqrt{2}}{64}\sigma\sqrt{d}$, $d \leq D$, and $\sigma\sqrt{D} \leq \frac{1}{640}\tau_{\mathcal{M}}$ assumed in Theorem A.1 from the third-to-last step to the second-to-last step.

As a result, the set $\left\{\boldsymbol{q} \xrightarrow{0} \boldsymbol{q}'' \mid \forall \boldsymbol{q}'' \in \mathcal{Q}_S\right\}$ satisfies the covering property in the assumption on $E^0$. Since $E^0$ is the minimum collection that satisfies this covering property, the number of zero-order edges at $\boldsymbol{q}$, i.e., $\deg^0(\boldsymbol{q})$ satisfies

$$\deg^0(\boldsymbol{q}) \leq \left|\left\{\boldsymbol{q} \xrightarrow{0} \boldsymbol{q}'' \mid \forall \boldsymbol{q}'' \in \mathcal{Q}_S\right\}\right| \leq |S_{\boldsymbol{q},\eta}| \leq \$_{\frac{1}{160}\tau_{\mathcal{M}}}^\eta. \quad \text{(A.10)}$$

And we note that projection of the gradient onto the tangent space takes the cost of $D * d$ number of operations, and choosing the first-order neighbor takes the cost of $d * \max_{\boldsymbol{q}_u \in \mathcal{Q}}\left|E_u^1\right|$. The number of arithmetic operations of zero-order step is $D * \max_{\boldsymbol{q} \in Q}\deg^0(\boldsymbol{q})$, where $D$ comes from the Euclidean distance calculation, and $\max_{\boldsymbol{q} \in Q}\deg^0(\boldsymbol{q})$ is bounded by the defined geometric quantity $\$_{\frac{1}{160}\tau_{\mathcal{M}}}^\eta(\mathcal{M})$. Combined all of terms above, we end up with the bound on the number of operations performed by our algorithm. ∎

**Fact A.2.** For a manifold $\mathcal{M}$ with reach $\tau_{\mathcal{M}}$ and extrinsic curvature bounded by $\kappa$, we have

$$\tau_{\mathcal{M}} \leq 1/\kappa. \quad \text{(A.11)}$$

See Proposition 2.3 in (Aamari et al., 2019). We used this fact directly in the following proofs.

**Proposition A.3.** *Let $\mathcal{M} \subset \mathbb{R}^D$ be a complete and connected d-dimensional manifold whose extrinsic geodesic curvature is bounded by $\kappa$. Suppose the landmarks $\{\boldsymbol{q}_u\}$ are $\delta$-separated, and form a $\delta$-net for $\mathcal{M}$, and that the first order graph $E^1$ satisfies $u \xrightarrow{1} v \in E^1$ when $\|\boldsymbol{q}_u - \boldsymbol{q}_v\|_2 \leq R_{nbrs}$, and that $40\delta \leq R_{nbrs} \leq \tau_{\mathcal{M}}$. Assume $\delta \leq \text{diam}(\mathcal{M})$ and $\kappa\text{diam}(\mathcal{M}) \geq 1$. Then with probability at least*

$$1 - e^{-\frac{9d}{2}} - \left(\frac{e}{2}\right)^{-\frac{D}{2}} \quad \text{(A.12)}$$

*in the noise $\boldsymbol{z}$, the first phase of first-order optimization in Algorithm 3 with parameters*

$$R_a = R_{nbrs} - \delta, \quad \text{(A.13)}$$

$$\epsilon_1 > \frac{2R_{nbrs}}{0.55}\left(\frac{2}{3} + \frac{8}{3}\kappa\text{diam}(\mathcal{M}) + \right. \quad \text{(A.14)}$$

$$\left. + 16\kappa\sigma\sqrt{d}\sqrt{\kappa\text{diam}(\mathcal{M}) + \log(a) - \log(\delta) + \log(\text{diam}(\mathcal{M})) + \log(100)}\right), \quad \text{(A.15)}$$

$$\text{(A.16)}$$

*produces $\boldsymbol{q}_{i_1}$ satisfying*

$$\text{grad}[\varphi_{\boldsymbol{x}}](\boldsymbol{q}_{i_1}) = \|\mathcal{P}_{T_{i_1}}(\boldsymbol{x} - \boldsymbol{q}_{i_1})\|_2 \leq \epsilon_1 \quad \text{(A.17)}$$

*using at most*

$$\frac{\text{diam}^2(\mathcal{M}) + 2\sigma^2 D}{0.1375\epsilon_1\delta} \quad \text{(A.18)}$$

*steps.*

**Proof.**

We can bound the total number of steps by dividing maximum initial distance by the minimal distance decrease over each first-order step.

We note that

$$
\begin{aligned}
\varphi_{\boldsymbol{x}}(\boldsymbol{q}_{i^0}) &= \frac{1}{2}\|\boldsymbol{q}_{i^0} - \boldsymbol{x}\|_2^2 \\
&\leq \|\boldsymbol{q}_{i^0} - \boldsymbol{x}_\natural\|_2^2 + \|\boldsymbol{z}\|_2^2 \\
&\leq \mathrm{diam}^2(\mathcal{M}) + \|\boldsymbol{z}\|_2^2
\end{aligned}
\tag{A.19}
$$

We let $\bar{\boldsymbol{z}}$ be the standard unit variance Gaussian variable, for any $0 < t < 1/2$, we have

$$
\begin{aligned}
\mathbb{P}[\|\boldsymbol{z}\|_2^2 \geq 2\sigma^2 D] = \mathbb{P}[\|\bar{\boldsymbol{z}}\|_2^2 \geq 2D] &= \mathbb{P}[e^{t\|\bar{\boldsymbol{z}}\|_2^2} \geq e^{2tD}] \\
&\leq \frac{\mathbb{E}[e^{t\|\bar{\boldsymbol{z}}\|_2^2}]}{e^{2tD}} \\
&= \frac{e^{-2tD}}{(1-2t)^{\frac{D}{2}}},
\end{aligned}
\tag{A.20}
$$

where we used Markov's inequality from the second line to the third line. In particular, we can pick $t = \frac{1}{4}$, so with probability at least $1 - (\frac{e}{2})^{-\frac{D}{2}}$ in the noise $\boldsymbol{z}$, we have

$$
\varphi_{\boldsymbol{x}}(\boldsymbol{q}_{i^0}) \leq \mathrm{diam}^2(\mathcal{M}) + 2\sigma^2 D.
\tag{A.21}
$$

Now we will analyze the decrease over each first-order step from $u$ to $u^+$. We have

$$
\begin{aligned}
\varphi_{\boldsymbol{x}}(\boldsymbol{q}_{u^+}) - \varphi_{\boldsymbol{x}}(\boldsymbol{q}_u) &= \frac{1}{2}(\|\boldsymbol{x} - \boldsymbol{q}_{u^+}\|_2^2 - \|\boldsymbol{x} - \boldsymbol{q}_u\|_2^2) \\
&= \frac{1}{2}\|\boldsymbol{q}_{u^+} - \boldsymbol{q}_u\|_2^2 - \langle \mathcal{P}_{T_u}(\boldsymbol{x} - \boldsymbol{q}_u), \mathcal{P}_{T_u}(\boldsymbol{q}_{u^+} - \boldsymbol{q}_u)\rangle - \langle \mathcal{P}_{N_u}(\boldsymbol{x} - \boldsymbol{q}_u), \mathcal{P}_{N_u}(\boldsymbol{q}_{u^+} - \boldsymbol{q}_u)\rangle.
\end{aligned}
\tag{A.22}
$$

We apply Lemma B.1 to bound the first term and Lemma B.8 to get a high probability bound on the last term, setting $t = 3\sigma\sqrt{d}$.

Next, we apply Lemma B.3 and Lemma B.4 to bound the second term, which dominates the decrease in the values of the objective function after taking a first-order step. As a reminder, the step rule in Algorithm 3 is

$$
u^+ = \arg\min_{u \xrightarrow{1} v} \left\|\mathcal{P}_{B(0,R_a)}\mathcal{P}_{T_u}(\boldsymbol{x} - \boldsymbol{q}_u) - \mathcal{P}_{T_u}(\boldsymbol{q}_v - \boldsymbol{q}_u)\right\|_2.
\tag{A.23}
$$

We observe that before phase I in Algorithm 3 terminates, $\|\mathcal{P}_{T_u}(\boldsymbol{x} - \boldsymbol{q}_u)\| \geq \epsilon_1 > R_a$ holds. Then the step rule is equivalent to

$$
u^+ = \arg\min_{u \xrightarrow{1} v} \left\|\frac{\mathcal{P}_{T_u}(\boldsymbol{x} - \boldsymbol{q}_u)}{\|\mathcal{P}_{T_u}(\boldsymbol{x} - \boldsymbol{q}_u)\|_2} R_a - \mathcal{P}_{T_u}(\boldsymbol{q}_v - \boldsymbol{q}_u)\right\|_2,
\tag{A.24}
$$

in which we have $\left\|\frac{\mathcal{P}_{T_u}(\boldsymbol{x}-\boldsymbol{q}_u)}{\|\mathcal{P}_{T_u}(\boldsymbol{x}-\boldsymbol{q}_u)\|_2} R_a\right\|_2 = R_a$. Applying Lemma B.3, there exists $u \xrightarrow{1} v \in E^1$, such that $\left\|\frac{\mathcal{P}_{T_u}(\boldsymbol{x}-\boldsymbol{q}_u)}{\|\mathcal{P}_{T_u}(\boldsymbol{x}-\boldsymbol{q}_u)\|_2} R_a - \mathcal{P}_{T_u}(\boldsymbol{q}_v - \boldsymbol{q}_u)\right\|_2 \leq \delta + \frac{1}{2}\kappa R_a^2$. From the construction of this step rule, we have $\left\|\frac{\mathcal{P}_{T_u}(\boldsymbol{x}-\boldsymbol{q}_u)}{\|\mathcal{P}_{T_u}(\boldsymbol{x}-\boldsymbol{q}_u)\|_2} R_a - \mathcal{P}_{T_u}(\boldsymbol{q}_{u^+} - \boldsymbol{q}_u)\right\|_2 \leq \left\|\frac{\mathcal{P}_{T_u}(\boldsymbol{x}-\boldsymbol{q}_u)}{\|\mathcal{P}_{T_u}(\boldsymbol{x}-\boldsymbol{q}_u)\|_2} R_a - \mathcal{P}_{T_u}(\boldsymbol{q}_v - \boldsymbol{q}_u)\right\|_2 \leq \delta + \frac{1}{2}\kappa R_a^2$. Thus when $\boldsymbol{x} \neq \boldsymbol{q}_u$, applying Lemma B.4, we have $\langle \mathcal{P}_{T_u}(\boldsymbol{x} - \boldsymbol{q}_u), \mathcal{P}_{T_u}(\boldsymbol{q}_v - \boldsymbol{q}_u)\rangle \geq 0.55\|\mathcal{P}_{T_u}(\boldsymbol{x} - \boldsymbol{q}_u)\|_2 \|\mathcal{P}_{T_u}(\boldsymbol{q}_v - \boldsymbol{q}_u)\|_2$.

Combining all the above results, we conclude that with probability at least $1 - e^{-\frac{9d}{2}}$ in the noise $\boldsymbol{z}$, we have

$$
\begin{aligned}
\varphi_{\boldsymbol{x}}(\boldsymbol{q}_{u^+}) - \varphi_{\boldsymbol{x}}(\boldsymbol{q}_u) \leq &\frac{2}{3}\|\mathcal{P}_{T_u}(\boldsymbol{q}_{u^+} - \boldsymbol{q}_u)\|_2^2 - 0.55\|\mathcal{P}_{T_u}(\boldsymbol{x} - \boldsymbol{q}_u)\|_2 \|\mathcal{P}_{T_u}(\boldsymbol{q}_{u^+} - \boldsymbol{q}_u)\|_2 \\
&+ \frac{8}{3}\kappa(\mathrm{diam}(\mathcal{M}) + \sqrt{2}\sigma\sqrt{\log|E^1|} + 3\sigma\sqrt{d})\|\mathcal{P}_{T_u}(\boldsymbol{q}_{u^+} - \boldsymbol{q}_u)\|_2^2.
\end{aligned}
\tag{A.25}
$$

Given our assumption on $\epsilon_1$ and applying the bound of $\left|E^1\right|$ in Lemma A.3, together with the assumption that $\delta \leq \operatorname{diam}(\mathcal{M})$ and $\kappa \operatorname{diam}(\mathcal{M}) > 1$, it follows that before Phase I ends, we have

$$
\begin{aligned}
0.55 \|\mathcal{P}_{T_u}(\boldsymbol{x} - \boldsymbol{q}_u)\|_2 > {}& 0.55\epsilon_1 \\
> {}& 2R_{\mathrm{nbrs}} \left( \frac{2}{3} + \frac{8}{3}\kappa \operatorname{diam}(\mathcal{M}) + \right. \\
& \left. + 16\kappa\sigma\sqrt{d}\sqrt{\kappa \operatorname{diam}(\mathcal{M}) + \log(a) - \log(\delta) + \log(\operatorname{diam}(\mathcal{M})) + \log(100)} \right) \\
> {}& 2R_{\mathrm{nbrs}} \left( \frac{2}{3} + \frac{8}{3}\kappa \operatorname{diam}(\mathcal{M}) \right. \\
& \left. + 8\kappa\sigma\sqrt{d} + 8\kappa\sigma\sqrt{d}\sqrt{\kappa \operatorname{diam}(\mathcal{M}) + \log(a) - \log(\delta) + \log(\operatorname{diam}(\mathcal{M})) + \log(100)} \right) \\
> {}& 2R_{\mathrm{nbrs}} \left( \frac{2}{3} + \frac{8}{3}\kappa \operatorname{diam}(\mathcal{M}) + 8\kappa\sigma\sqrt{d} + 8\kappa\sigma\sqrt{\log|E^1|} \right) \\
> {}& 2R_{\mathrm{nbrs}} \left( \frac{2}{3} + \frac{8}{3}\kappa \left( \operatorname{diam}(\mathcal{M}) + \sigma\sqrt{2\log|E^1|} + 3\sigma\sqrt{d} \right) \right),
\end{aligned}
$$
(A.26)

together with $\|\mathcal{P}_{T_u}(\boldsymbol{q}_{u^+} - \boldsymbol{q}_u)\|_2 \leq \|\boldsymbol{q}_{u^+} - \boldsymbol{q}_u\|_2 \leq R_{\mathrm{nbrs}}$, then we have

$$
\varphi_{\boldsymbol{x}}(\boldsymbol{q}_{u^+}) - \varphi_{\boldsymbol{x}}(\boldsymbol{q}_u) \leq -\frac{0.55}{2}\|\mathcal{P}_{T_u}(\boldsymbol{q}_{u^+} - \boldsymbol{q}_u)\|_2 \|\mathcal{P}_{T_u}(\boldsymbol{x} - \boldsymbol{q}_u)\|_2.
$$
(A.27)

In the following, we develop a lower bound for $\|\mathcal{P}_{T_u}(\boldsymbol{q}_{u^+} - \boldsymbol{q}_u)\|_2$. Let $\boldsymbol{\gamma} : [0, 1] \to \mathcal{M}$ be a minimum length geodesic joining $\boldsymbol{q}_u$ and $\boldsymbol{q}_{u^+}$ with constant speed $d_{\mathcal{M}}(\boldsymbol{q}_u, \boldsymbol{q}_{u^+})$, where $\boldsymbol{\gamma}(0) = \boldsymbol{q}_u, \boldsymbol{\gamma}(1) = \boldsymbol{q}_{u^+}$. Then we have

$$
\begin{aligned}
\|P_{T_u}(\boldsymbol{q}_{u^+} - \boldsymbol{q}_u)\|_2 &= \|P_{T_u}(\boldsymbol{\gamma}(1) - \boldsymbol{\gamma}(0))\|_2 \\
&= \left\| P_{T_u} \left( \int_{a=0}^1 \dot{\boldsymbol{\gamma}}(a)da \right) \right\|_2 \\
&= \left\| \dot{\boldsymbol{\gamma}}(0) + \int_{a=0}^1 \int_{b=0}^a P_{T_u}\ddot{\boldsymbol{\gamma}}(b)dbda \right\|_2 \\
&\geq \|\dot{\boldsymbol{\gamma}}(0)\|_2 - \left\| \int_{a=0}^1 \int_{b=0}^a P_{T_u}\ddot{\boldsymbol{\gamma}}(b)dbda \right\|_2 \\
&\geq \|\dot{\boldsymbol{\gamma}}(0)\|_2 - \int_{a=0}^1 \int_{b=0}^a \|P_{T_u}\ddot{\boldsymbol{\gamma}}(b)\|_2 dbda \\
&\geq \|\dot{\boldsymbol{\gamma}}(0)\|_2 - \int_{a=0}^1 \int_{b=0}^a \kappa \|\dot{\boldsymbol{\gamma}}(0)\|_2^2 dbda \\
&= d_{\mathcal{M}}(\boldsymbol{q}_{u^+}, \boldsymbol{q}_u) - \frac{1}{2}\kappa d_{\mathcal{M}}^2(\boldsymbol{q}_{u^+}, \boldsymbol{q}_u) \\
&\geq \frac{1}{2}d_{\mathcal{M}}(\boldsymbol{q}_{u^+}, \boldsymbol{q}_u) \\
&\geq \frac{1}{2}\delta,
\end{aligned}
$$
(A.28)

where in the third line we've used the fact that $\dot{\boldsymbol{\gamma}}(0)$ lies in the tangent space of $q_u$, in the forth and fifth lines we applied triangle inequality, and in the final lines we've used our assumption that $d_{\mathcal{M}}(\boldsymbol{q}_{u^+}, \boldsymbol{q}_u) \leq \tau_{\mathcal{M}} \leq \frac{1}{\kappa}$ and the landmarks are $\delta$-separated.
Plugging this back into Equation (A.27), we have

$$
\varphi_{\boldsymbol{x}}(\boldsymbol{q}_{u^+}) - \varphi_{\boldsymbol{x}}(\boldsymbol{q}_u) \leq \frac{\delta}{2} \left( -\frac{0.55}{2}\|\mathcal{P}_{T_u}(\boldsymbol{x} - \boldsymbol{q}_u)\|_2 \right) \leq \frac{-0.55}{4}\delta\epsilon_1.
$$
(A.29)

Lastly, combining this result with Equation (A.21) yields the desired upper bound on the number of iterations.

■

In the following proposition, we let

$$N_{\boldsymbol{q}}^{\eta}\mathcal{M} \tag{A.30}$$

denote the $\eta$-dilated normal space:

$$N_{\boldsymbol{q}}^{\eta}\mathcal{M} = \left\{ \boldsymbol{v} \in \mathbb{R}^D \mid \|\mathcal{P}_{T_{\boldsymbol{q}}\mathcal{M}}\boldsymbol{v}\|_2 \le \eta \right\}. \tag{A.31}$$

**Proposition A.4.** *Assume $\delta \le \operatorname{diam}(\mathcal{M})$, $\kappa\operatorname{diam}(\mathcal{M}) \ge 1$ and $\sigma\sqrt{D} \le \frac{1}{640}\tau_{\mathcal{M}}$. Suppose for every $\boldsymbol{q} \in Q$ and every distinct $\boldsymbol{q}' \in \boldsymbol{q} + N_{\boldsymbol{q}}^{\eta}\mathcal{M}$ with*

$$\eta \ge \epsilon_1 + \delta + 14\sigma\sqrt{d}\sqrt{\kappa\operatorname{diam}(\mathcal{M}) + \log(\operatorname{diam}(\mathcal{M})) - \log(\delta) + \log(7)}, \tag{A.32}$$

*there exists a ZOE $\boldsymbol{q} \xrightarrow{0} \boldsymbol{q}''$ with $\boldsymbol{q}'' \in B_{\mathcal{M}}(\boldsymbol{q}', \frac{1}{80}\tau_{\mathcal{M}})$ Then, with probability at least $1 - 2e^{-\frac{9d}{2}}$ in the noise $\boldsymbol{z}$, whenever $\boldsymbol{q}_{i_{\mathrm{I}}}$ satisfies*

$$\|\mathcal{P}_{T_{\boldsymbol{q}_{i_{\mathrm{I}}}}\mathcal{M}}(\boldsymbol{x} - \boldsymbol{q}_{i_{\mathrm{I}}})\| \le \epsilon_1, \tag{A.33}$$

*the second phase (zero-order step) produces $\boldsymbol{q}_{i_{\mathrm{II}}}$ satisfying*

$$d_{\mathcal{M}}(\boldsymbol{q}_{i_{\mathrm{II}}}, \boldsymbol{x}_{\natural}) \le 0.1/\kappa. \tag{A.34}$$

**Proof.** From Lemma A.2, we have

$$|\mathcal{Q}| \le \left( 1 + 4\sqrt{2}\delta^{-1}\operatorname{diam}(\mathcal{M})e^{\kappa\operatorname{diam}(\mathcal{M})} \right)^d, \tag{A.35}$$

which gives us that

$$\begin{aligned}
\log|\mathcal{Q}| &\le d\log\left( 1 + 4\sqrt{2}\delta^{-1}\operatorname{diam}(\mathcal{M})e^{\kappa\operatorname{diam}(\mathcal{M})} \right) \\
&\le d\log\left( \left(\frac{1}{e} + 4\sqrt{2}\right)\delta^{-1}\operatorname{diam}(\mathcal{M})e^{\kappa\operatorname{diam}(\mathcal{M})} \right) \\
&\le d\left( \log\left( 7\delta^{-1}\operatorname{diam}(\mathcal{M}) \right) + \kappa\operatorname{diam}(\mathcal{M}) \right),
\end{aligned} \tag{A.36}$$

where we used the assumption that $\delta \le \operatorname{diam}(\mathcal{M})$ and $\kappa\operatorname{diam}(\mathcal{M}) \ge 1$ from the first line to the second line. Together with the assumption on $\eta$, we have

$$\eta \ge \epsilon_1 + \delta + \sigma\left( 4\sqrt{\log|\mathcal{Q}|} + 10\sqrt{d} \right). \tag{A.37}$$

Under these conditions and the assumption that $\sigma\sqrt{D} \le \frac{1}{640}\tau_{\mathcal{M}}$, applying Lemma C.3, we conclude that with probability at least $1 - 2e^{\frac{-9d}{2}}$ in the noise, $\boldsymbol{q}_{i_{\mathrm{II}}} = \arg\min_{\boldsymbol{q}':\boldsymbol{q}_{i_{\mathrm{I}}}\xrightarrow{0}\boldsymbol{q}'} \|\boldsymbol{q}' - \boldsymbol{x}\|_2$ satisfies

$$d_{\mathcal{M}}(\boldsymbol{x}_{\natural}, \boldsymbol{q}_{i_{\mathrm{II}}}) \le \frac{1}{20}\tau_{\mathcal{M}} + 2\delta \le \frac{1}{20}\tau_{\mathcal{M}} + \frac{1}{20}R_{\mathrm{nbrs}} \le \frac{1}{20}\tau_{\mathcal{M}} + \frac{1}{20}\tau_{\mathcal{M}} = \frac{1}{10}\tau_{\mathcal{M}}. \tag{A.38}$$

■

**Proposition A.5.** *Assume $\sigma\sqrt{D} \le \frac{1}{640}\tau_{\mathcal{M}}$. Suppose the landmarks $\{\boldsymbol{q}_u\}$ are $\delta$-separated, and form a $\delta$-net for $\mathcal{M}$, and that the first order graph $E^1$ satisfies $u \xrightarrow{1} v \in E^1$ when $\|\boldsymbol{q}_u - \boldsymbol{q}_v\|_2 \le R_{\mathrm{nbrs}}$, and that*

$$40\delta \le R_{nbrs} \le \frac{\sqrt{2}}{64}\sigma\sqrt{d}. \tag{A.39}$$

*Initializing at $\boldsymbol{q}_{i_{\mathrm{II}}} \in B_{\mathcal{M}}(\boldsymbol{x}_{\natural}, 0.1/\kappa)$, with probability at least*

$$1 - e^{-\frac{9}{2}d} - e^{-\frac{1}{16}d}, \tag{A.40}$$

*the third phase of first-order optimization in Algorithm 3 with parameters*

$$R_a = R_{\mathrm{nbrs}} - \delta, \tag{A.41}$$

$$\epsilon_2 > C \max\{\kappa, 1\} \sigma \sqrt{d}, \tag{A.42}$$

*for some $C > 0$, produces $\boldsymbol{q}_{i_{\mathrm{III}}}$ which satisfies*

$$d_{\mathcal{M}}(\boldsymbol{q}_{i_{\mathrm{III}}}, \boldsymbol{x}_{\natural}) \le 2\epsilon_2, \tag{A.43}$$

*using at most*

$$\frac{8}{\kappa\delta} \tag{A.44}$$

*steps.*

**Proof.** Let

$$\boldsymbol{q}_{i_{\mathrm{II}}} = \boldsymbol{q}^0, \boldsymbol{q}^1, \boldsymbol{q}^2, \dots \tag{A.45}$$

denote the sequence of landmarks $\boldsymbol{q}^j$ produced by first order optimization in Phase III. Let $j_\star \in \mathbb{Z} \cup \{\infty\}$ denote the number of iterations taken by this phase of optimization. If the algorithm terminates at some finite $j_\star$, we have

$$\|P_{T_{\boldsymbol{q}^{j_\star}}}(\boldsymbol{x} - \boldsymbol{q}^{j_\star})\| \le \epsilon_2. \tag{A.46}$$

Below, we will prove that indeed the algorithm terminates, and bound the number of steps $j_\star$ required. To this end, we will prove that on an event of probability at least

$$1 - e^{-\frac{9}{2}d} - e^{-\frac{1}{16}d}, \tag{A.47}$$

the iterates $\boldsymbol{q}^j$ $(j = 0, 1, \dots, j_\star)$ satisfy the following property:

$$d_{\mathcal{M}}(\boldsymbol{q}^j, \boldsymbol{x}_{\natural}) \le \max\left\{ \frac{1}{10\kappa} - c_1 j\delta, C_1 \max\{\kappa, 1\}\sigma\sqrt{d} \right\}. \tag{A.48}$$

This immediately implies that there exists $C_2$ such that for all

$$j \ge \frac{C_2}{\kappa\delta}, \tag{A.49}$$

the iterates $\boldsymbol{q}^j$ satisfies

$$d_{\mathcal{M}}(\boldsymbol{q}^j, \boldsymbol{x}_{\natural}) \le C_1 \max\{\kappa, 1\}\sigma\sqrt{d}. \tag{A.50}$$

**Step Sizes.** Our next task is to verify (A.48). We begin by noting some bounds on the step size $d_{\mathcal{M}}(\boldsymbol{q}^j, \boldsymbol{q}^{j-1})$. By construction, for each $j$,

$$\|\boldsymbol{q}^{j+1} - \boldsymbol{q}^j\|_2 \le R_{\mathrm{nbrs}} \le \frac{\sqrt{2}}{64}\sigma\sqrt{d}. \tag{A.51}$$

Since we know $\frac{1}{\kappa} \ge \tau_{\mathcal{M}}$, $D \ge d$, and we've assumed $\tau_{\mathcal{M}} \ge 640\sigma\sqrt{D}$, we have

$$R_{\mathrm{nbrs}} \le \min\left\{ \frac{\sqrt{2}}{64}\sigma\sqrt{d}, \tau_{\mathcal{M}}, \frac{1}{200\kappa} \right\}. \tag{A.52}$$

Since $\boldsymbol{q}^j, \boldsymbol{q}^{j+1} \in \mathcal{M}$ and $\|\boldsymbol{q}^{j+1} - \boldsymbol{q}^j\|_2 \le \tau_{\mathcal{M}}$, applying Lemma B.5, we have

$$d_{\mathcal{M}}(\boldsymbol{q}^{j+1}, \boldsymbol{q}^j) \le 2\|\boldsymbol{q}^{j+1} - \boldsymbol{q}^j\|_2 \le \min\left\{ \frac{\sqrt{2}}{32}\sigma\sqrt{d}, \frac{1}{100\kappa} \right\}. \tag{A.53}$$

From Lemma A.5, take $t = \frac{\sqrt{2}}{4}\sigma\sqrt{d}$, with probability at least $1 - e^{-\frac{d}{16}}$, the random variable

$$T_{\max} = \sup_{\boldsymbol{y} \in B_{\mathcal{M}}(\boldsymbol{x}_{\natural}, 1/\kappa), \boldsymbol{v} \in T_{\boldsymbol{y}}\mathcal{M}, \|\boldsymbol{v}\|_2 = 1} \langle \boldsymbol{v}, \boldsymbol{z} \rangle, \tag{A.54}$$

satisfies

$$T_{\max} \ge \frac{\sqrt{2}}{4}\sigma\sqrt{d}, \tag{A.55}$$

which gives us $\frac{\sqrt{2}}{32}\sigma\sqrt{d} \le \frac{1}{8}T_{\max}$. Hence on an event of probability at least $1 - e^{-\frac{d}{16}}$, for every $j$,

$$d_{\mathcal{M}}(\boldsymbol{q}^{j+1}, \boldsymbol{q}^j) \le \min\left\{ \frac{\sqrt{2}}{32}\sigma\sqrt{d}, \frac{1}{100\kappa} \right\} \le \min\left\{ \frac{T_{\max}}{8}, \frac{1}{100\kappa} \right\}. \tag{A.56}$$

In particular, this implies that the condition on the step size in Lemma D.7 is satisfied.

**Radius of Decrease.** From Lemma A.4, with probability at least $1 - e^{-\frac{9}{2}d}$, $T_{\max}$ satisfies

$$T_{\max} \leq C_4 \max\{\kappa, 1\} \sigma \sqrt{d}, \tag{A.57}$$

and hence the distance condition in Lemma D.7 is satisfied whenever

$$C_5 \max\{\kappa, 1\} \sigma \sqrt{d} \leq d_{\mathcal{M}}(\boldsymbol{q}^j, \boldsymbol{x}_\natural) \leq \frac{1}{10\kappa}. \tag{A.58}$$

**Proof of Equation** (A.48). We proceed by induction on $j$. For $j = 0$, $\boldsymbol{q}^0 = \boldsymbol{q}_{i_{\mathrm{II}}} \in B_{\mathcal{M}}(\boldsymbol{x}_\natural, \frac{1}{10\kappa})$, and so (A.48) holds. Now, suppose that (A.48) condition holds for iterates $0, 1, \ldots, j-1$. By Lemma D.7, *if*

$$C_5 \max\{\kappa, 1\} \sigma \sqrt{d} \leq d_{\mathcal{M}}(\boldsymbol{q}^j, \boldsymbol{x}_\natural) \leq \frac{1}{10\kappa}. \tag{A.59}$$

then

$$
\begin{aligned}
d_{\mathcal{M}}(\boldsymbol{q}^j, \boldsymbol{x}_\natural) &\leq d_{\mathcal{M}}(\boldsymbol{q}^{j-1}, \boldsymbol{x}_\natural) - c_2 d_{\mathcal{M}}(\boldsymbol{q}^j, \boldsymbol{q}^{j-1}) & \text{(A.60)}\\
&\leq d_{\mathcal{M}}(\boldsymbol{q}^{j-1}, \boldsymbol{x}_\natural) - c_2 \delta & \text{(A.61)}\\
&\leq \frac{1}{10\kappa} - c_2(j-1)\delta - c_2 j\delta & \text{(A.62)}\\
&\leq \frac{1}{10\kappa} - c_2 j\delta. & \text{(A.63)}
\end{aligned}
$$

On the other hand, if $d_{\mathcal{M}}(\boldsymbol{q}^{j-1}, \boldsymbol{x}_\natural) < C_5 \max\{\kappa, 1\} \sigma \sqrt{d}$, we have

$$
\begin{aligned}
d_{\mathcal{M}}(\boldsymbol{q}^j, \boldsymbol{x}_\natural) &\leq d_{\mathcal{M}}(\boldsymbol{q}^{j-1}, \boldsymbol{x}_\natural) + d_{\mathcal{M}}(\boldsymbol{q}^j, \boldsymbol{q}^{j-1}) & \text{(A.64)}\\
&\leq C_5 \max\{\kappa, 1\} \sigma \sqrt{d} + \frac{\sqrt{2}}{32} \sigma \sqrt{d} & \text{(A.65)}\\
&\leq C_6 \max\{\kappa, 1\} \sigma \sqrt{d}. & \text{(A.66)}
\end{aligned}
$$

Combining, we have

$$d_{\mathcal{M}}(\boldsymbol{q}^j, \boldsymbol{x}_\natural) \leq \max\left\{\frac{1}{10\kappa} - c_2 j\delta, C_6 \max\{\kappa, 1\} \sigma \sqrt{d}\right\}, \tag{A.67}$$

and (A.48) is verified.

**Proof of Termination.** We next verify that under our assumptions on $\epsilon_2$, the algorithm terminates. For an arbitrary point $\boldsymbol{q} \in B_{\mathcal{M}}(\boldsymbol{x}_\natural, 1/\kappa)$, let $\phi(t)$ denote a geodesic with $\phi(0) = \boldsymbol{x}_\natural$, $\phi(1) = \boldsymbol{q}$ and constant speed $d_{\mathcal{M}}(\boldsymbol{x}_\natural, \boldsymbol{q})$. Then

$$
\begin{aligned}
\left\|P_{T_{\boldsymbol{q}}\mathcal{M}}(\boldsymbol{x}_\natural - \boldsymbol{q})\right\|_2 &= \left\|P_{T_{\boldsymbol{q}}\mathcal{M}}\dot{\phi}(0) + \int_{a=0}^1 \int_{b=0}^a P_{T_{\boldsymbol{q}}\mathcal{M}}\ddot{\phi}(b)dbda\right\|_2 \\
&= \left\|\dot{\phi}(0) + \int_{a=0}^1 \int_{b=0}^a P_{T_{\boldsymbol{q}}\mathcal{M}}\ddot{\phi}(b)dbda\right\|_2 \\
&\leq \left\|\dot{\phi}(0)\right\|_2 + \left\|\int_{a=0}^1 \int_{b=0}^a P_{T_{\boldsymbol{q}}\mathcal{M}}\ddot{\phi}(b)dbda\right\|_2 \\
&\leq d_{\mathcal{M}}(\boldsymbol{x}_\natural, \boldsymbol{q}) + \int_{a=0}^1 \int_{b=0}^a \left\|P_{T_{\boldsymbol{q}}\mathcal{M}}\ddot{\phi}(b)\right\|_2 dbda \\
&\leq d_{\mathcal{M}}(\boldsymbol{x}_\natural, \boldsymbol{q}) + \int_{a=0}^1 \int_{b=0}^a \kappa d_{\mathcal{M}}^2(\boldsymbol{x}_\natural, \boldsymbol{q})dbda \\
&\leq d_{\mathcal{M}}(\boldsymbol{x}_\natural, \boldsymbol{q}) + \frac{1}{2}\kappa d_{\mathcal{M}}^2(\boldsymbol{x}_\natural, \boldsymbol{q}) \\
&\leq \frac{3}{2}d_{\mathcal{M}}(\boldsymbol{x}_\natural, \boldsymbol{q}).
\end{aligned} \tag{A.68}
$$

Comparing this lower bound to (A.48), we obtain that for any $\epsilon_2 > \frac{3}{2}C_1 \max\{\kappa, 1\}\sigma\sqrt{d}$, after at most

$$\widehat{j} > \frac{1}{10c_2\kappa\delta} \tag{A.69}$$

steps, the algorithm produces a point $\boldsymbol{q}^{j\star}$ satisfying

$$\|P_{T_{\boldsymbol{q}^{j\star}}\mathcal{M}}(\boldsymbol{x} - \boldsymbol{q}^{j\star})\| \leq \epsilon_2, \tag{A.70}$$

and hence the algorithm terminates after at most $\frac{1}{10c_2\kappa\delta}$ steps. To be more specific, Lemma D.7 gives $c_2 = \frac{1}{80}$, so the algorithm stops after at most $\frac{8}{\kappa\delta}$ steps.

**Quality of Terminal Point.** Finally, we bound the distance of $\boldsymbol{q}_{i_{\mathrm{III}}} = \boldsymbol{q}^{j\star}$ to $\boldsymbol{x}_\natural$. By reasoning analogous to (A.68), we obtain that for $\boldsymbol{q} \in B_{\mathcal{M}}(\boldsymbol{x}_\natural, 1/\kappa)$

$$\left\|P_{T_{\boldsymbol{q}}\mathcal{M}}(\boldsymbol{x}_\natural - \boldsymbol{q})\right\|_2 \geq \frac{1}{2}d_{\mathcal{M}}(\boldsymbol{x}_\natural, \boldsymbol{q}). \tag{A.71}$$

This immediately implies that

$$d_{\mathcal{M}}(\boldsymbol{q}^{j\star}, \boldsymbol{x}_\natural) \leq 2\epsilon_2, \tag{A.72}$$

as claimed. ∎

# B. Phase I Analysis

**Lemma B.1.** *With the assumption $R_{nbrs} \leq \tau_{\mathcal{M}}$, for any $\boldsymbol{q}_u, \boldsymbol{q}_v \in \mathcal{M}$ and $u \xrightarrow{1} v \in E^1$, we have*

$$\|\boldsymbol{q}_u - \boldsymbol{q}_v\|_2^2 \leq \frac{4}{3}\|\mathcal{P}_{T_u}(\boldsymbol{q}_u - \boldsymbol{q}_v)\|_2^2, \tag{B.1}$$

*where $T_u$ is the tangent space at $\boldsymbol{q}_u \in \mathcal{M}$, and $\mathcal{P}_{T_u}$ is the orthogonal projection onto $T_u$.*

**Proof.** For the self edge $u \xrightarrow{1} u \in E^1$, it's easy to show that $\|\boldsymbol{q}_u - \boldsymbol{q}_u\|_2^2 = \frac{4}{3}\|\mathcal{P}_{T_u}(\boldsymbol{q}_u - \boldsymbol{q}_u)\|_2^2 = 0$.
Now we consider the situation when $u \xrightarrow{1} v \in E^1 (u \neq v)$. By the theorem 2.2 in (Aamari et al., 2019) and theorem 4.18 in (Federer, 1959), we know the reach $\tau_{\mathcal{M}}$ of the manifold $\mathcal{M}$ satisfies

$$\tau_{\mathcal{M}} = \inf_{\boldsymbol{q}_u \neq \boldsymbol{q}_v \in \mathcal{M}} \frac{\|\boldsymbol{q}_u - \boldsymbol{q}_v\|_2^2}{2\|\mathcal{P}_{N_u}(\boldsymbol{q}_u - \boldsymbol{q}_v)\|_2}, \tag{B.2}$$

where $N_u$ is the normal space at $\boldsymbol{q}_u \in \mathcal{M}$, and $\mathcal{P}_{N_u}$ is the orthogonal projection onto the $N_u$.
Thus, for any $u \xrightarrow{1} v \in E^1 (u \neq v)$, we have

$$\tau_{\mathcal{M}} \leq \frac{\|\boldsymbol{q}_u - \boldsymbol{q}_v\|_2^2}{2\|\mathcal{P}_{N_u}(\boldsymbol{q}_u - \boldsymbol{q}_v)\|_2}. \tag{B.3}$$

By the rule of connecting first-order edges, for any edge $u \xrightarrow{1} v \in E^1$, we have $\|\boldsymbol{q}_u - \boldsymbol{q}_v\|_2 \leq R_{nbrs}$.
By assumption, $R_{nbrs} \leq \tau_{\mathcal{M}}$, thus for first-order edges $u \xrightarrow{1} v (u \neq v)$, we have:

$$\begin{aligned}
\|\boldsymbol{q}_u - \boldsymbol{q}_v\| &\leq R_{nbrs} \\
&\leq \tau_{\mathcal{M}} \\
&\leq \frac{\|\boldsymbol{q}_u - \boldsymbol{q}_v\|_2^2}{2\|\mathcal{P}_{N_u}(\boldsymbol{q}_u - \boldsymbol{q}_v)\|_2}.
\end{aligned} \tag{B.4}$$

Square both sides, we obtain:

$$\|\boldsymbol{q}_u - \boldsymbol{q}_v\|_2^2 \geq 4\|\mathcal{P}_{N_u}(\boldsymbol{q}_u - \boldsymbol{q}_v)\|_2^2. \tag{B.5}$$

Since the tangent and normal components are orthogonal, we have $\|\boldsymbol{q}_u - \boldsymbol{q}_v\|_2^2 = \|\mathcal{P}_{T_u}(\boldsymbol{q}_u - \boldsymbol{q}_v)\|_2^2 + \|\mathcal{P}_{N_u}(\boldsymbol{q}_u - \boldsymbol{q}_v)\|_2^2$.
Therefore, we have:

$$\|P_{T_u}(\boldsymbol{q}_u - \boldsymbol{q}_v)\|_2^2 \geq 3\|P_{N_u}(\boldsymbol{q}_u - \boldsymbol{q}_v)\|_2^2. \tag{B.6}$$

Consequently, we have

$$
\begin{aligned}
\|\boldsymbol{q}_u - \boldsymbol{q}_v\|_2^2 &= \|\mathcal{P}_{T_u}(\boldsymbol{q}_u - \boldsymbol{q}_v)\|_2^2 + \|\mathcal{P}_{N_u}(\boldsymbol{q}_u - \boldsymbol{q}_v)\|_2^2 \\
&\leq \|\mathcal{P}_{T_u}(\boldsymbol{q}_u - \boldsymbol{q}_v)\|_2^2 + \frac{1}{3}\|\mathcal{P}_{T_u}(\boldsymbol{q}_u - \boldsymbol{q}_v)\|_2^2 \\
&= \frac{4}{3}\|\mathcal{P}_{T_u}(\boldsymbol{q}_u - \boldsymbol{q}_v)\|_2^2.
\end{aligned}
\tag{B.7}
$$

This completes the proof. ∎

**Lemma B.2.** *For any $\boldsymbol{w}_\natural \in B(\boldsymbol{q}_u, R_a) \cap \mathcal{M}$, there exists $v : u \xrightarrow{1} v \in E^1$, such that*

$$
\|\boldsymbol{w}_\natural - \boldsymbol{q}_v\|_2 \leq \delta.
\tag{B.8}
$$

**Proof.** Since the landmarks form a $\delta$- cover of $\mathcal{M}$, and $R_{\text{nbrs}} = a \cdot \delta (a \geq 40)$, there exists $v \in V$ such that $\|\boldsymbol{q}_v - \boldsymbol{w}_\natural\|_2 \leq \delta = \frac{1}{a} R_{\text{nbrs}}$.
From the triangle inequality, we have

$$
\begin{aligned}
\|\boldsymbol{q}_v - \boldsymbol{q}_u\|_2 &\leq \|\boldsymbol{q}_v - \boldsymbol{w}_\natural\|_2 + \|\boldsymbol{w}_\natural - \boldsymbol{q}_u\|_2 \\
&\leq \frac{1}{a} R_{\text{nbrs}} + R_a \\
&= \frac{1}{a} R_{\text{nbrs}} + \frac{a-1}{a} R_{\text{nbrs}} \\
&= R_{\text{nbrs}}.
\end{aligned}
\tag{B.9}
$$

From the rule of connecting first-order edges, we know $u \xrightarrow{1} v \in E^1$. This completes the proof. ∎

**Lemma B.3.** *For any $\boldsymbol{w} \in T_u$, with $\|\boldsymbol{w}\|_2 \leq R_a$, there exists $v : u \xrightarrow{1} v \in E^1$, such that*

$$
\|\boldsymbol{w} - \mathcal{P}_{T_u}(\boldsymbol{q}_v - \boldsymbol{q}_u)\|_2 \leq \delta + \frac{1}{2}\kappa R_a^2.
\tag{B.10}
$$

**Proof.** Consider a constant-speed geodesic $\boldsymbol{\gamma} : [0,1] \to \mathcal{M}$, $\boldsymbol{\gamma}(0) = \boldsymbol{q}_u, \dot{\boldsymbol{\gamma}}(0) = \boldsymbol{w}$ with $\|\boldsymbol{w}\|_2 \leq R_a$. Let $\boldsymbol{\gamma}(1) = \boldsymbol{w}_\natural$.
From the fundamental theorem of calculus, we have

$$
\boldsymbol{\gamma}(1) = \boldsymbol{\gamma}(0) + \int_{t=0}^{1} \dot{\boldsymbol{\gamma}}(t)dt.
\tag{B.11}
$$

Therefore,

$$
\begin{aligned}
\|\boldsymbol{w}_\natural - \boldsymbol{q}_u\|_2 &= \left\| \int_{t=0}^{1} \dot{\boldsymbol{\gamma}}(t)dt \right\|_2 \\
&\leq \int_{t=0}^{1} \|\dot{\boldsymbol{\gamma}}(t)\| \, dt \\
&= \|\boldsymbol{w}\|_2 \\
&\leq R_a.
\end{aligned}
\tag{B.12}
$$

This means $\boldsymbol{w}_\natural \in B(\boldsymbol{q}_u, R_a)$. By lemma B.2, there exists $v : u \xrightarrow{1} v \in E^1$, such that $\|\boldsymbol{w}_\natural - \boldsymbol{q}_v\|_2 \leq \delta$.
By fundamental theorem of calculus, we also know

$$
\begin{aligned}
\boldsymbol{\gamma}(1) &= \boldsymbol{\gamma}(0) + \int_{t=0}^{1} \dot{\boldsymbol{\gamma}}(t)dt \\
&= \boldsymbol{\gamma}(0) + \int_{t=0}^{1} \left( \dot{\boldsymbol{\gamma}}(0) + \int_{s=0}^{t} \ddot{\boldsymbol{\gamma}}(s)ds \right) dt \\
&= \boldsymbol{\gamma}(0) + \dot{\boldsymbol{\gamma}}(0) + \int_{t=0}^{1} \int_{s=0}^{t} \ddot{\boldsymbol{\gamma}}(s)dsdt.
\end{aligned}
\tag{B.13}
$$

Then we have

$$\boldsymbol{w} = \boldsymbol{w}_\natural - \boldsymbol{q}_u - \int_{t=0}^{1} \int_{s=0}^{t} \ddot{\boldsymbol{\gamma}}(s) ds dt. \tag{B.14}$$

Since $\boldsymbol{w} \in T_u$, we have

$$\boldsymbol{w} = \mathcal{P}_{T_u}(\boldsymbol{w}) = \mathcal{P}_{T_u}(\boldsymbol{w}_\natural - \boldsymbol{q}_u) - \mathcal{P}_{T_u}\left(\int_{t=0}^{1} \int_{s=0}^{t} \ddot{\boldsymbol{\gamma}}(s) ds dt\right). \tag{B.15}$$

Hence we know

$$
\begin{aligned}
\|\boldsymbol{w} - \mathcal{P}_{T_u}(\boldsymbol{q}_v - \boldsymbol{q}_u)\|_2 &= \left\| \mathcal{P}_{T_u}(\boldsymbol{w}_\natural - \boldsymbol{q}_u) - \mathcal{P}_{T_u}\left(\int_{t=0}^{1}\int_{s=0}^{t} \ddot{\boldsymbol{\gamma}}(s) ds dt\right) - \mathcal{P}_{T_u}(\boldsymbol{q}_v - \boldsymbol{q}_u) \right\|_2 \\
&= \left\| \mathcal{P}_{T_u}(\boldsymbol{w}_\natural - \boldsymbol{q}_v) - \mathcal{P}_{T_u}\left(\int_{t=0}^{1}\int_{s=0}^{t} \ddot{\boldsymbol{\gamma}}(s) ds dt\right) \right\|_2 \\
&\leq \|\boldsymbol{w}_\natural - \boldsymbol{q}_v\|_2 + \left\| \int_{t=0}^{1}\int_{s=0}^{t} \ddot{\boldsymbol{\gamma}}(s) ds dt \right\|_2 \\
&\leq \delta + \int_{t=0}^{1}\int_{s=0}^{t} \|\ddot{\boldsymbol{\gamma}}(s)\|_2 ds dt \\
&\leq \delta + \frac{1}{2}\kappa \|\boldsymbol{w}\|_2^2 \\
&\leq \delta + \frac{1}{2}\kappa R_a^2.
\end{aligned}
\tag{B.16}
$$

$\blacksquare$

**Lemma B.4.** *For any $\boldsymbol{q}_u \in \mathcal{M}$ and $\boldsymbol{x} \neq \boldsymbol{q}_u$, let $v : u \xrightarrow{1} v \in E^1$ satisfies:*

$$\left\| \frac{\mathcal{P}_{T_u}(\boldsymbol{x} - \boldsymbol{q}_u)}{\|\mathcal{P}_{T_u}(\boldsymbol{x} - \boldsymbol{q}_u)\|_2} R_a - \mathcal{P}_{T_u}(\boldsymbol{q}_v - \boldsymbol{q}_u) \right\|_2 \leq \delta + \frac{1}{2}\kappa R_a^2. \tag{B.17}$$

*Then we have*

$$\langle \mathcal{P}_{T_u}(\boldsymbol{x} - \boldsymbol{q}_u), \mathcal{P}_{T_u}(\boldsymbol{q}_v - \boldsymbol{q}_u) \rangle \geq 0.55 \|\mathcal{P}_{T_u}(\boldsymbol{x} - \boldsymbol{q}_u)\|_2 \|\mathcal{P}_{T_u}(\boldsymbol{q}_v - \boldsymbol{q}_u)\|_2. \tag{B.18}$$

**Proof.** Let $\delta_T = \delta + \frac{1}{2}\kappa R_a^2$. Given our global constraints that $\delta = \frac{1}{a} R_{\text{nbrs}}$, $a \geq 40$ and $R_{\text{nbrs}} \leq \tau_{\mathcal{M}} \leq 1/\kappa$,

$$
\begin{aligned}
\frac{\delta_T}{R_a} &= \frac{\frac{1}{a} R_{\text{nbrs}} + \frac{1}{2}\kappa(1 - \frac{1}{a})^2 R_{\text{nbrs}}^2}{(1 - \frac{1}{a}) R_{\text{nbrs}}} \\
&= \frac{1}{a(1 - \frac{1}{a})} + \frac{1}{2}\kappa(1 - \frac{1}{a}) R_{\text{nbrs}} \\
&\leq \frac{1}{a - 1} + \frac{1}{2}(1 - \frac{1}{a}) \\
&\leq \frac{5}{6}.
\end{aligned}
\tag{B.19}
$$

From the assumption we have

$$\left\| \frac{\mathcal{P}_{T_u}(\boldsymbol{x} - \boldsymbol{q}_u)}{\|\mathcal{P}_{T_u}(\boldsymbol{x} - \boldsymbol{q}_u)\|_2} R_a - \mathcal{P}_{T_u}(\boldsymbol{q}_v - \boldsymbol{q}_u) \right\|_2 \leq \delta_T. \tag{B.20}$$

Square both sides of the equation (B.20), we have

$$R_a^2 + \|\mathcal{P}_{T_u}(\boldsymbol{q}_v - \boldsymbol{q}_u)\|_2^2 - 2\left\langle \frac{\mathcal{P}_{T_u}(\boldsymbol{x} - \boldsymbol{q}_u)}{\|\mathcal{P}_{T_u}(\boldsymbol{x} - \boldsymbol{q}_u)\|_2} R_a, \mathcal{P}_{T_u}(\boldsymbol{q}_v - \boldsymbol{q}_u) \right\rangle \leq \delta_T^2. \tag{B.21}$$

Then we have

$$\left\langle \frac{\mathcal{P}_{T_u}\left(\boldsymbol{x} - \boldsymbol{q}_u\right)}{\|\mathcal{P}_{T_u}\left(\boldsymbol{x} - \boldsymbol{q}_u\right)\|_2} R_a, \mathcal{P}_{T_u}\left(\boldsymbol{q}_v - \boldsymbol{q}_u\right) \right\rangle \geq \frac{R_a^2 - \delta_T^2 + \|\mathcal{P}_{T_u}\left(\boldsymbol{q}_v - \boldsymbol{q}_u\right)\|_2^2}{2}$$
$$\geq \sqrt{R_a^2 - \delta_T^2} \|\mathcal{P}_{T_u}\left(\boldsymbol{q}_v - \boldsymbol{q}_u\right)\|_2. \tag{B.22}$$

Therefore, we have

$$\langle \mathcal{P}_{T_u}\left(\boldsymbol{x} - \boldsymbol{q}_u\right), \mathcal{P}_{T_u}\left(\boldsymbol{q}_v - \boldsymbol{q}_u\right) \rangle \geq \sqrt{1 - \frac{\delta_T^2}{R_a^2}} \|\mathcal{P}_{T_u}\left(\boldsymbol{x} - \boldsymbol{q}_u\right)\|_2 \|\mathcal{P}_{T_u}\left(\boldsymbol{q}_v - \boldsymbol{q}_u\right)\|_2. \tag{B.23}$$

Substituting our result from eq(B.19) to eq (B.23), we have the following inequality:

$$\langle \mathcal{P}_{T_u}\left(\boldsymbol{x} - \boldsymbol{q}_u\right), \mathcal{P}_{T_u}\left(\boldsymbol{q}_v - \boldsymbol{q}_u\right) \rangle \geq 0.55 \|\mathcal{P}_{T_u}\left(\boldsymbol{x} - \boldsymbol{q}_u\right)\|_2 \|\mathcal{P}_{T_u}\left(\boldsymbol{q}_v - \boldsymbol{q}_u\right)\|_2. \tag{B.24}$$

∎

**Lemma B.5.** *If $\boldsymbol{q}_u, \boldsymbol{q}_v \in \mathcal{M}$ and $\|\boldsymbol{q}_u - \boldsymbol{q}_v\|_2 \leq \tau_{\mathcal{M}}$, we have*

$$d_{\mathcal{M}}(\boldsymbol{q}_u, \boldsymbol{q}_v) \leq 2\|\boldsymbol{q}_u - \boldsymbol{q}_v\|_2, \tag{B.25}$$

*where $d_{\mathcal{M}}(\cdot, \cdot)$ is the intrinsic distance along the manifold.*

**Proof.** Let $\boldsymbol{q}_t = \boldsymbol{q}_u + t(\boldsymbol{q}_v - \boldsymbol{q}_u), t \in [0, 1]$. From Theorem C in (Leobacher & Steinicke, 2021), we know

$$\frac{d}{dt}\mathcal{P}_{\mathcal{M}}[\boldsymbol{q}_t] = (\boldsymbol{I} - \mathbb{I}^*[\boldsymbol{q}_t - \mathcal{P}_{\mathcal{M}}[\boldsymbol{q}_t]])^{-1} \mathcal{P}_{T_{\mathcal{P}_{\mathcal{M}}[\boldsymbol{q}_t]}\mathcal{M}}(\boldsymbol{q}_v - \boldsymbol{q}_u). \tag{B.26}$$

Therefore, we have

$$\|\frac{d}{dt}\mathcal{P}_{\mathcal{M}}[\boldsymbol{q}_t]\|_2 \leq \|\left(\boldsymbol{I} - \mathbb{I}^*[\boldsymbol{q}_t - \mathcal{P}_{\mathcal{M}}[\boldsymbol{q}_t]]\right)^{-1}\|_{\text{op}} \|\mathcal{P}_{T_{\boldsymbol{q}_t}\mathcal{M}}(\boldsymbol{q}_v - \boldsymbol{q}_u)\|_2$$
$$\leq \frac{\|\boldsymbol{q}_v - \boldsymbol{q}_u\|_2}{1 - \kappa\|\boldsymbol{q}_t - \mathcal{P}_{\mathcal{M}}[\boldsymbol{q}_t]\|_2}$$
$$= \frac{\|\boldsymbol{q}_v - \boldsymbol{q}_u\|_2}{1 - \kappa d(\boldsymbol{q}_t, \mathcal{M})}$$
$$\leq \frac{\|\boldsymbol{q}_v - \boldsymbol{q}_u\|_2}{1 - \kappa \min\{\|\boldsymbol{q}_t - \boldsymbol{q}_u\|_2, \|\boldsymbol{q}_t - \boldsymbol{q}_v\|_2\}} \tag{B.27}$$
$$\leq \frac{\|\boldsymbol{q}_v - \boldsymbol{q}_u\|_2}{1 - \frac{1}{2}\kappa\|\boldsymbol{q}_v - \boldsymbol{q}_u\|_2}$$
$$\leq \frac{\|\boldsymbol{q}_v - \boldsymbol{q}_u\|_2}{1 - \frac{1}{2}\kappa\tau_{\mathcal{M}}}$$
$$\leq 2\|\boldsymbol{q}_v - \boldsymbol{q}_u\|_2,$$

where in the second line we've used the property that $\forall \boldsymbol{\eta}, \|\mathbb{I}^*[\boldsymbol{\eta}]\| \leq \kappa\|\boldsymbol{\eta}\|_2$, and in the last line we've applied the fact that $\tau_{\mathcal{M}} \leq 1/\kappa$. Then the intrinsic distance along the manifold is

$$d_{\mathcal{M}}(\boldsymbol{q}_u, \boldsymbol{q}_v) \leq \int_{t=0}^1 \|\frac{d}{dt}\mathcal{P}_{\mathcal{M}}[\boldsymbol{q}_t]\|_2 dt$$
$$\leq \int_{t=0}^1 2\|\boldsymbol{q}_v - \boldsymbol{q}_u\|_2 dt \tag{B.28}$$
$$= 2\|\boldsymbol{q}_v - \boldsymbol{q}_u\|_2.$$

This completes the proof. ∎

**Lemma B.6.** *For any $\boldsymbol{q}_u, \boldsymbol{q}_v \in \mathcal{M}$, we have*

$$\|\mathcal{P}_{N_u}\left(\boldsymbol{q}_v - \boldsymbol{q}_u\right)\|_2 \leq \frac{1}{2}\kappa d_{\mathcal{M}}^2(\boldsymbol{q}_u, \boldsymbol{q}_v). \tag{B.29}$$

**Proof.** Consider the geodesic $\gamma : [0,1] \to M \subset \mathbb{R}^D$ with constant speed $\|v\|_2$, where $\gamma(0) = q_u, \gamma(1) = q_v$. Then the intrinsic distance between the $q_u$ and $q_v$ is:

$$d_{\mathcal{M}}(q_u, q_v) = \int_{t=0}^{1} \|v\|_2 dt = \|v\|_2. \tag{B.30}$$

From equation (B.13) and the fact $\dot{\gamma}(0) \in T_u$, we have

$$\|\mathcal{P}_{N_u}(\gamma(1) - \gamma(0))\|_2 = \left\| \int_{t=0}^{1} \int_{s=0}^{t} \ddot{\gamma}(s) ds dt \right\|_2 \tag{B.31}$$

$$
\begin{aligned}
\|\mathcal{P}_{N_u}(q_v - q_u)\|_2 &= \|\mathcal{P}_{N_u}(\gamma(1) - \gamma(0))\|_2 \\
&= \left\| \mathcal{P}_{N_u} \int_{t=0}^{1} \int_{s=0}^{t} \ddot{\gamma}(s) ds dt \right\|_2 \\
&\leq \left\| \int_{t=0}^{1} \int_{s=0}^{t} \ddot{\gamma}(s) ds dt \right\|_2 \\
&\leq \int_{t=0}^{1} \int_{s=0}^{t} \|\ddot{\gamma}(s)\|_2 ds dt \\
&\leq \int_{t=0}^{1} \int_{s=0}^{t} \kappa \|v\|_2 ds dt \\
&= \frac{1}{2} \kappa \|v\|_2^2 \\
&= \frac{1}{2} \kappa d_{\mathcal{M}}^2(q_u, q_v).
\end{aligned}
\tag{B.32}
$$

This completes the proof. ∎

**Lemma B.7.** *If $q_u, q_v \in \mathcal{M}$ and $\|q_u - q_v\|_2 \leq \tau_{\mathcal{M}}$, we have*

$$\|\mathcal{P}_{N_u}(q_v - q_u)\|_2 \leq \frac{8}{3} \kappa \|\mathcal{P}_{T_u}(q_v - q_u)\|_2^2. \tag{B.33}$$

**Proof.** Given $q_u, q_v \in \mathcal{M}$ and $\|q_u - q_v\|_2 \leq \tau_{\mathcal{M}}$, from lemma B.5, we know $d_{\mathcal{M}}(q_u, q_v) \leq 2\|q_u - q_v\|_2$. From lemma B.6, we have $\|\mathcal{P}_{N_u}(q_v - q_u)\|_2 \leq \frac{1}{2} \kappa d_{\mathcal{M}}^2(q_u, q_v)$. By lemma B.1, we have $\|q_u - q_v\|_2^2 \leq \frac{4}{3}\|\mathcal{P}_{T_u}(q_u - q_v)\|_2^2$. Then we have

$$
\begin{aligned}
\|\mathcal{P}_{N_u}(q_v - q_u)\|_2 &\leq \frac{1}{2} \kappa d_{\mathcal{M}}^2(q_u, q_v) \\
&\leq 2\kappa \|q_u - q_v\|_2^2 \\
&= \frac{8}{3} \kappa \|\mathcal{P}_{T_u}(q_u - q_v)\|_2^2.
\end{aligned}
\tag{B.34}
$$

This completes the proof. ∎

**Lemma B.8.** *For any $t > 0$, with probability at least $1 - e^{-\frac{t^2}{2\sigma^2}}$ in the noise, for any $q_u, q_v \in \mathcal{M}$ and $u \xrightarrow{1} v \in E^1$,*

$$-\langle \mathcal{P}_{N_u}(x - q_u), \mathcal{P}_{N_u}(q_v - q_u)\rangle \leq \frac{8}{3} \kappa (\mathrm{diam}(\mathcal{M}) + \sqrt{2}\sigma\sqrt{\log|E^1|} + t)\|\mathcal{P}_{T_u}(q_v - q_u)\|_2^2. \tag{B.35}$$

**Proof.** As $u \xrightarrow{1} v \in E^1$, we have $\|q_u - q_v\| \leq R_{\mathrm{nbrs}} \leq \tau_{\mathcal{M}}$ from the construction of $E^1$. We decompose the left hand side in equation (B.35) as follows:

$$-\langle \mathcal{P}_{N_u}(x - q_u), \mathcal{P}_{N_u}(q_v - q_u)\rangle = -\langle \mathcal{P}_{N_u}(x_\natural - q_u), \mathcal{P}_{N_u}(q_v - q_u)\rangle - \langle z, \mathcal{P}_{N_u}(q_v - q_u)\rangle. \tag{B.36}$$

From lemma B.7, we have $\|\mathcal{P}_{N_u}(\boldsymbol{q}_v - \boldsymbol{q}_u)\|_2 \leq \frac{8}{3}\kappa\|\mathcal{P}_{T_u}(\boldsymbol{q}_v - \boldsymbol{q}_u)\|_2^2$. Then the first component on the right hand side can be rewritten as:

$$-\langle \mathcal{P}_{N_u}(\boldsymbol{x}_\natural - \boldsymbol{q}_u), \mathcal{P}_{N_u}(\boldsymbol{q}_v - \boldsymbol{q}_u)\rangle \leq \|\boldsymbol{x}_\natural - \boldsymbol{q}_u\|_2\|\mathcal{P}_{N_u}(\boldsymbol{q}_v - \boldsymbol{q}_u)\|_2$$
$$\leq \frac{8}{3}\kappa\operatorname{diam}\mathcal{M}\|\mathcal{P}_{T_u}(\boldsymbol{q}_v - \boldsymbol{q}_u)\|_2^2. \tag{B.37}$$

Now consider a set of random variables

$$X_{\boldsymbol{y}} = -\langle \boldsymbol{z}, \boldsymbol{y}\rangle \quad \boldsymbol{y} \in \mathcal{Y}, \tag{B.38}$$

with

$$\mathcal{Y} = \left\{ \frac{\mathcal{P}_{N_u}(\boldsymbol{q}_v - \boldsymbol{q}_u)}{\|\mathcal{P}_{N_u}(\boldsymbol{q}_v - \boldsymbol{q}_u)\|_2} \,\bigg|\, \forall u \xrightarrow{1} v \in E^1, u \neq v, \forall u, v \in \mathcal{V} \right\}.$$

Because $\boldsymbol{z} \sim \mathcal{N}(\boldsymbol{0}, \sigma^2\boldsymbol{I})$, $X_{\boldsymbol{y}}$ follows the distribution $X_{\boldsymbol{y}} \sim \mathcal{N}(0, \sigma^2)$, and $(X_{\boldsymbol{y}})_{\boldsymbol{y}\in\mathcal{Y}}$ is a Gaussian process. We first bound the expectation of the supremum of this Gaussian process $\mathbb{E}[\sup_{\boldsymbol{y}\in\mathcal{Y}} X_{\boldsymbol{y}}]$ by starting from

$$e^{t'\mathbb{E}[\sup_{\boldsymbol{y}\in\mathcal{Y}} X_{\boldsymbol{y}}]} = e^{\mathbb{E}[t'\sup_{\boldsymbol{y}\in\mathcal{Y}} X_{\boldsymbol{y}}]}$$
$$\leq \mathbb{E}[e^{t'\sup_{\boldsymbol{y}\in\mathcal{Y}} X_{\boldsymbol{y}}}]$$
$$\leq \sum_{i=1}^{|\mathcal{Y}|} \mathbb{E}[e^{t'X_{\boldsymbol{y}}}] \tag{B.39}$$
$$= |Y|e^{\frac{1}{2}\sigma^2 t'^2}, \quad \forall t' > 0.$$

Take logarithm of both sides, we have

$$\mathbb{E}[\sup_{\boldsymbol{y}\in\mathcal{Y}} X_{\boldsymbol{y}}] \leq \frac{\log|\mathcal{Y}|}{t} + \frac{\sigma^2 t'}{2}. \tag{B.40}$$

When $t' = \frac{\sqrt{2\log|Y|}}{\sigma}$, the right hand side of equation (B.40) is minimized, and we obtain

$$\mathbb{E}[\sup_{\boldsymbol{y}\in\mathcal{Y}} X_{\boldsymbol{y}}] \leq \sqrt{2}\sigma\sqrt{\log|Y|}. \tag{B.41}$$

By using Borell–TIS inequality (Adler & Taylor, 2007), for any $t > 0$ we have

$$\mathbb{P}(\sup_{\boldsymbol{y}\in\mathcal{Y}} X_{\boldsymbol{y}} - \mathbb{E}[\sup_{\boldsymbol{y}\in\mathcal{Y}} X_{\boldsymbol{y}}] > t) \leq e^{\frac{-t^2}{2\sigma^2}}. \tag{B.42}$$

This implies

$$\mathbb{P}\left(-\left\langle \boldsymbol{z}, \frac{\mathcal{P}_{N_u}(\boldsymbol{q}_v - \boldsymbol{q}_u)}{\|\mathcal{P}_{N_u}(\boldsymbol{q}_v - \boldsymbol{q}_u)\|_2}\right\rangle - \mathbb{E}[\sup_{\boldsymbol{y}\in\mathcal{Y}} X_{\boldsymbol{y}}] \leq t\right) > 1 - e^{\frac{-t^2}{2\sigma^2}}. \tag{B.43}$$

Therefore, with probability at least $1 - e^{-\frac{t^2}{2\sigma^2}}$, we have

$$-\langle \boldsymbol{z}, \mathcal{P}_{N_u}(\boldsymbol{q}_v - \boldsymbol{q}_u)\rangle \leq \|\mathcal{P}_{N_u}(\boldsymbol{q}_v - \boldsymbol{q}_u)\|_2(\mathbb{E}[\sup_{\boldsymbol{y}\in\mathcal{Y}} X_{\boldsymbol{y}}] + t)$$
$$\leq \|\mathcal{P}_{N_u}(\boldsymbol{q}_v - \boldsymbol{q}_u)\|_2(\sqrt{2}\sigma\sqrt{\log|Y|} + t) \tag{B.44}$$
$$\leq \frac{8}{3}\kappa\|\mathcal{P}_{T_u}(\boldsymbol{q}_v - \boldsymbol{q}_u)\|_2^2(\sqrt{2}\sigma\sqrt{\log|Y|} + t). \quad \text{by lemma B.7}$$

Plug equation (B.37) and equation (B.44), with probability at least $1 - e^{-\frac{t^2}{2\sigma^2}}$, we have

$$-\langle \mathcal{P}_{N_u}(\boldsymbol{x}_\natural - \boldsymbol{q}_u), \mathcal{P}_{N_u}(\boldsymbol{q}_v - \boldsymbol{q}_u)\rangle - \langle \boldsymbol{z}, \mathcal{P}_{N_u}(\boldsymbol{q}_v - \boldsymbol{q}_u)\rangle$$
$$\leq -\langle \mathcal{P}_{N_u}(\boldsymbol{x}_\natural - \boldsymbol{q}_u), \mathcal{P}_{N_u}(\boldsymbol{q}_v - \boldsymbol{q}_u)\rangle + \sup_{u\xrightarrow{1}v} -\langle \boldsymbol{z}, \mathcal{P}_{N_u}(\boldsymbol{q}_v - \boldsymbol{q}_u)\rangle \tag{B.45}$$
$$\leq \frac{8}{3}\kappa(\operatorname{diam}(\mathcal{M}) + \sqrt{2}\sigma\sqrt{\log|Y|} + t)\|\mathcal{P}_{T_u}(\boldsymbol{q}_v - \boldsymbol{q}_u)\|_2^2.$$

We conclude the proof by noting that $|Y| < |E^1|$. $\blacksquare$

# C. Phase II Analysis

**Lemma C.1.** *Given noise vector $z \sim_{\text{iid}} \mathcal{N}(0, \sigma^2)$, let $H = \max_{q \in Q} \left\| P_{T_q \mathcal{M}} z \right\|$. Then, with probability at least $1 - e^{-\frac{9d}{2}}$, we have*

$$H \leq \sigma \left( 4\sqrt{\log |Q|} + 10\sqrt{d} \right). \tag{C.1}$$

**Proof.** For any landmark $q$, we define set $S_q = \{v \in T_q M : \|v\| = 1\}$. From Corrollary 4.2.13 in (Vershynin, 2018), we know that for any $\delta$, $S_q$ can be $\delta$-covered by at most $\left( \frac{1+\delta/2}{\delta/2} \right)^d = \left( 1 + \frac{2}{\delta} \right)^d$ points. In particular, we can choose $\delta = \frac{1}{2}$ and let $C_q$ to be a 1/2 cover of $S_q$ such that $|C_q| \leq 5^d$. We let $S_Q = \cup_{q \in Q} S_q$ and $C_Q = \cup_{q \in Q} C_q$. Then, we have

$$H = \max_{u \in S_Q} \langle u, z \rangle \tag{C.2}$$

$$\leq \max_{v \in C_Q} \langle v, z \rangle + \frac{1}{2} H, \tag{C.3}$$

$$\tag{C.4}$$

this implies

$$H \leq 2 \max_{v \in C_Q} \langle v, z \rangle. \tag{C.5}$$

Let $h(z) = \max_{v \in C_Q} \langle v, z \rangle$, and since it's a 1-lipshitz function in $z$, by applying gaussian concentration inequality, we have

$$\mathbb{P} \left( h(z) - \mathbb{E}[h(z)] \geq s \right) \leq e^{-\frac{s^2}{2\sigma^2}}. \tag{C.6}$$

We then bound its expectation. For any $t > 0$,

$$\begin{aligned}
e^{t\mathbb{E}[h(z)]} &\leq \mathbb{E}[e^{t[h(z)]}] \\
&= \mathbb{E}[e^{t \max_{v \in C_Q} \langle v, z \rangle}] \\
&\leq \sum_{v \in C_Q} \mathbb{E}[e^{t \langle v, z \rangle}] \\
&= |C_Q| e^{\frac{1}{2} \sigma^2 t^2}.
\end{aligned} \tag{C.7}$$

Take logarithm of both sides, we have

$$\mathbb{E}[h(z)] \leq \frac{\log |C_Q|}{t} + \frac{\sigma^2 t}{2}. \tag{C.8}$$

When $t = \frac{\sqrt{2 \log |C_Q|}}{\sigma}$, the right hand side of equation (C.8) is minimized, and we obtain

$$\mathbb{E}[h(z)] \leq \sqrt{2}\sigma \sqrt{\log |C_Q|} = \sqrt{2}\sigma \sqrt{d \log 5 + \log |Q|}. \tag{C.9}$$

Plugging this result back to C.6, we have

$$\mathbb{P} \left( h(z) - \sqrt{2}\sigma \sqrt{d \log 5 + \log |Q|} \geq s \right) \leq e^{-\frac{s^2}{2\sigma^2}}. \tag{C.10}$$

Picking $s = 3\sigma\sqrt{d}$, this implies with probability at least $1 - e^{-\frac{9d}{2}}$, we have

$$\max_{v \in C_Q} \langle v, z \rangle = h(z) \leq \sigma \left( \sqrt{(2 \log 5)d + 2 \log |Q|} + 3\sqrt{d} \right) \leq \sigma \left( \sqrt{2 \log |Q|} + 5\sqrt{d} \right). \tag{C.11}$$

Plugging this result back to C.5, we have with probability $\geq 1 - e^{-\frac{9d}{2}}$,

$$H \leq \sigma \left( 2\sqrt{2 \log |Q|} + 10\sqrt{d} \right). \tag{C.12}$$

∎

**Lemma C.2.** *Suppose for any landmark $q'' \in q + N_q^\eta$ [6], with*

$$\eta \geq \epsilon + \delta + \sigma \left( 4\sqrt{\log |Q|} + 10\sqrt{d} \right), \tag{C.13}$$

*there exists a zero-order edge $q \xrightarrow{0} q'$, such that*

$$q' \in B_\mathcal{M}(q'', c_4 \tau_\mathcal{M}), \tag{C.14}$$

*where $\delta$ is the covering radius for $Q$, $\epsilon > 0$, and $c_4 \leq \frac{1}{40}$.*
*Then, for $x = x_\natural + z$, with probability at least $1 - e^{-\frac{9d}{2}}$ in the noise $z$, the following property obtains: for every point $q$ such that $x \in q + N_q^\epsilon$, there exists a zero-order edge $q \xrightarrow{0} q'''$ for some $q''' \in B_\mathcal{M}(x_\natural, c_4 \tau_\mathcal{M} + \delta)$.*

**Proof.** For $x = x_\natural + z \in q + N_q^\epsilon \mathcal{M}$, we have

$$\left\| P_{T_q}(x - q) \right\|_2 \leq \epsilon. \tag{C.15}$$

By the covering property, there exists $q'' \in Q$, such that

$$d_\mathcal{M}(x_\natural, q'') \leq \delta. \tag{C.16}$$

Since

$$\left\| P_{T_q}(x - q) \right\|_2 \geq \left\| P_{T_q}(q'' - q) \right\|_2 - \left\| P_{T_q}(x_\natural - q'') \right\|_2 - \left\| P_{T_q} z \right\|_2, \tag{C.17}$$

we have

$$\left\| P_{T_q}(q'' - q) \right\|_2 - \left\| P_{T_q}(x_\natural - q'') \right\|_2 - \left\| P_{T_q} z \right\|_2 \leq \epsilon. \tag{C.18}$$

Together with $\left\| P_{T_q}(x_\natural - q'') \right\|_2 \leq \|x_\natural - q''\|_2 \leq \delta$, and Lemma C.1, then with probability at least $1 - e^{-\frac{9d}{2}}$, we have

$$\begin{aligned} \left\| P_{T_q}(q'' - q) \right\|_2 &\leq \epsilon + \delta + H \\ &\leq \eta. \end{aligned} \tag{C.19}$$

By assumption, there exists a zero-order edge $q \xrightarrow{0} q'$, such that $q' \in B_\mathcal{M}(q'', c_4 \tau_\mathcal{M})$. Then

$$\begin{aligned} d_\mathcal{M}(q', x_\natural) &\leq d_\mathcal{M}(q', q'') + d_\mathcal{M}(q'', x_\natural) \\ &\leq c_4 \tau_\mathcal{M} + \delta. \end{aligned} \tag{C.20}$$

■

**Lemma C.3.** *Assume $\sigma\sqrt{D} \leq c_1 \tau_\mathcal{M}$ for some $c_1 \leq \frac{1}{640}$ and for any landmark $q'' \in q + N_q^\eta$, with $\eta \geq \epsilon + \delta + \sigma \left( 4\sqrt{\log |Q|} + 10\sqrt{d} \right)$, where $\delta$ is the covering radius of $Q$ and $\epsilon > 0$, there exists a zero-order edge $q \xrightarrow{0} q'$, such that $q' \in B_\mathcal{M}(q'', c_4 \tau_\mathcal{M})$, where $c_4 \leq \frac{1}{40}$. Then, for any noisy $x = x_\natural + z$ where $x \in q + N_q^\epsilon$, with probability at least $1 - 2e^{-\frac{9d}{2}}$ in the noise, $q^* = \arg\min_{q': q \xrightarrow{0} q'} \|q' - x\|_2$ satisfies landmark $q^* \in B_\mathcal{M}(x_\natural, (2c_4 + 16c_1)\tau_\mathcal{M} + 2\delta)$*

**Proof.** Given the assumption, lemma C.2 guarantees that with probability at least $1 - e^{\frac{-9d}{2}}$, there exists a landmark $q'''$ and a zero-order edge $q \xrightarrow{0} q'''$, such that $q''' \in B_\mathcal{M}(x_\natural, c_4 \tau_\mathcal{M} + \delta)$. Since $q^* = \arg\min_{q': q \xrightarrow{0} q'} \|q' - x\|_2$, we know that

$$\|q^* - x\|_2 \leq \|q''' - x\|_2. \tag{C.21}$$

which is the same as

$$\|q^* - x_\natural - z\|_2 \leq \|q''' - x_\natural - z\|_2. \tag{C.22}$$

From triangular inequality, we have

$$\|q^* - x_\natural\|_2 \leq \|q''' - x_\natural\|_2 + 2\|z\|_2. \tag{C.23}$$

---

[6] $N_q^\eta \mathcal{M} = \left\{ v \in \mathbb{R}^D \mid \|\mathcal{P}_{T_q \mathcal{M}} v\|_2 \leq \eta \right\}$

Since $g(z) = \|z\|_2$ is 1-lipschitz function in $z$ and $\mathbb{E}[\|z\|_2] \leq \sigma\sqrt{D}$, then with probability at least $1 - e^{-\frac{t^2}{2\sigma^2}}$, we have

$$\|z\|_2 \leq \sigma\sqrt{D} + t. \tag{C.24}$$

Take $t = 3\sigma\sqrt{D}$, then with probability at least $1 - e^{-\frac{9D}{2}}$, we have

$$\|z\|_2 \leq 4\sigma\sqrt{D} \leq 4c_1\tau_{\mathcal{M}}, \tag{C.25}$$

which implies with probability at least $1 - e^{-\frac{9D}{2}}$, we have

$$\|q^* - x_\natural\|_2 \leq \|q''' - x_\natural\|_2 + 8c_1\tau_{\mathcal{M}} \leq (c_4 + 8c_1)\tau_{\mathcal{M}} + \delta \leq \tau_{\mathcal{M}}. \tag{C.26}$$

The last inequality follows from the conditions $c_4 \leq 1/40, c_1 \leq 1/640, \delta = R_{\mathrm{nbrs}}/a \leq \tau_{\mathcal{M}}/40$. Together with the fact that on this scale $d_{\mathcal{M}}(x_\natural, q^*) \leq 2\|q^* - x_\natural\|_2$, we have

$$d_{\mathcal{M}}(x_\natural, q^*) \leq (2c_4 + 16c_1)\tau_{\mathcal{M}} + 2\delta. \tag{C.27}$$

■

## D. Phase III Analysis

**Lemma D.1.** *Consider the first-order edge $u \xrightarrow{1} v$. Let $\xi_{uv} = P_{T_u}(q_v - q_u)$ denote the edge embedding, where $P_{T_u}$ is the projection operator onto the tangent space $T_u$ of the manifold $\mathcal{M}$ at landmark $q_u$. Let $\gamma : [0, 1] \to \mathcal{M}$ be a geodesic joining $q_u$ and $q_v$ with constant speed $\|v\|_2$, where $\gamma(0) = q_u, \gamma(1) = q_v$. Then, for all $t \in [0, 1]$, we have*

$$\|\dot{\gamma}(t) - \xi_{uv}\|_2 \leq \frac{3}{2}\kappa d_{\mathcal{M}}^2(q_u, q_v). \tag{D.1}$$

**Proof.** From Taylor expansion, we have

$$q_v = q_u + \int_{a=0}^{1} \dot{\gamma}(a)da. \tag{D.2}$$

Therefore $\xi_{uv} = P_{T_u} \int_{a=0}^{1} \dot{\gamma}(a)da$. Then

$$\|\dot{\boldsymbol{\gamma}}(t) - \boldsymbol{\xi}_{uv}\|_2 = \left\|\dot{\boldsymbol{\gamma}}(t) - P_{T_u} \int_{a=0}^1 \dot{\boldsymbol{\gamma}}(a)da\right\|_2$$

$$= \left\|\dot{\boldsymbol{\gamma}}(0) + \int_{b=0}^t \ddot{\boldsymbol{\gamma}}(b)db - P_{T_u}\int_{a=0}^1\left(\dot{\boldsymbol{\gamma}}(0) + \int_{b=0}^a \ddot{\boldsymbol{\gamma}}(b)db\right)da\right\|_2$$

$$= \left\|\int_{b=0}^t \ddot{\boldsymbol{\gamma}}(b)db - \int_{a=0}^1\int_{b=0}^a P_{T_u}\ddot{\boldsymbol{\gamma}}(b)dbda\right\|_2$$

$$= \left\|\int_{a=0}^1\int_{b=0}^t \ddot{\boldsymbol{\gamma}}(b)dbda - \int_{a=0}^1\left(\int_{b=0}^t P_{T_u}\ddot{\boldsymbol{\gamma}}(b)db\right) + \left(\int_{b=t}^a P_{T_u}\ddot{\boldsymbol{\gamma}}(b)db\right)da\right\|_2$$

$$= \left\|\int_{a=0}^1\left(\int_{b=0}^t P_{N_u}\ddot{\boldsymbol{\gamma}}(b)db - \int_{b=t}^a P_{T_u}\ddot{\boldsymbol{\gamma}}(b)db\right)da\right\|_2$$

$$\leq \int_{a=0}^1\left\|\int_{b=0}^t P_{N_u}\ddot{\boldsymbol{\gamma}}(b)db\right\|_2 + \left\|\int_{b=t}^a P_{T_u}\ddot{\boldsymbol{\gamma}}(b)db\right\|_2 da \tag{D.3}$$

$$\leq \int_{a=0}^1 (t + |a - t|)\kappa\|\boldsymbol{v}\|_2^2\, da$$

$$= \left(\int_{a=0}^t [2t - a]\, da + \int_{a=t}^1 a\, da\right)\kappa\|\boldsymbol{v}\|_2^2$$

$$= \left(2t^2 - \tfrac{1}{2}t^2 + \tfrac{1}{2} - \tfrac{1}{2}t^2\right)\kappa\|\boldsymbol{v}\|_2^2$$

$$= \left(\frac{1}{2} + t^2\right)\kappa\|\boldsymbol{v}\|_2^2$$

$$\leq \frac{3}{2}\kappa\|\boldsymbol{v}\|_2^2$$

$$= \frac{3}{2}\kappa d_{\mathcal{M}}^2(\boldsymbol{q}_u, \boldsymbol{q}_v).$$

∎

**Lemma D.2.** *For any $\boldsymbol{z}$ and $\boldsymbol{x}_\natural, \boldsymbol{y} \in \mathcal{M}$, let*

$$T_{\max} = \sup_{\boldsymbol{y}\in B_{\mathcal{M}}(\boldsymbol{x}_\natural, 1/\kappa), \boldsymbol{v}\in T_{\boldsymbol{y}}\mathcal{M}, \|\boldsymbol{v}\|_2 = 1} \langle \boldsymbol{v}, \boldsymbol{z}\rangle, \tag{D.4}$$

*Then if $d_{\mathcal{M}}(\boldsymbol{y}, \boldsymbol{x}_\natural) \leq 1/\kappa$, we have*

$$\left\|-\log_{\boldsymbol{y}}\boldsymbol{x}_\natural - \mathrm{grad}[\varphi_{\boldsymbol{x}}](\boldsymbol{y})\right\|_2 \leq \frac{1}{2}\kappa d_{\mathcal{M}}^2(\boldsymbol{y}, \boldsymbol{x}_\natural) + T_{\max}. \tag{D.5}$$

**Proof.** We first decompose the left hand side into

$$\left\|-\log_{\boldsymbol{y}}\boldsymbol{x}_\natural - \mathrm{grad}[\varphi_{\boldsymbol{x}}](\boldsymbol{y})\right\|_2 = \left\|-\log_{\boldsymbol{y}}\boldsymbol{x}_\natural + P_{T_{\boldsymbol{y}}\mathcal{M}}(\boldsymbol{x} - \boldsymbol{y})\right\|_2$$
$$\leq \left\|-\log_{\boldsymbol{y}}\boldsymbol{x}_\natural + P_{T_{\boldsymbol{y}}\mathcal{M}}(\boldsymbol{x}_\natural - \boldsymbol{y})\right\|_2 + \left\|P_{T_{\boldsymbol{y}}\mathcal{M}}\boldsymbol{z}\right\|_2. \tag{D.6}$$

Consider a geodesic $\boldsymbol{\eta} : [0,1] \to \mathcal{M}$ with constant speed $\|\boldsymbol{v}_{\boldsymbol{\eta}}\|_2$, where $\boldsymbol{\eta}(0) = \boldsymbol{y}, \boldsymbol{\eta}(1) = \boldsymbol{x}_\natural$, then we bound the signal

term (first term) of the above bound:

$$\left\|-\log_{\boldsymbol{y}}\boldsymbol{x}_\natural + P_{T_{\boldsymbol{y}}\mathcal{M}}(\boldsymbol{x}_\natural - \boldsymbol{y})\right\|_2 = \left\|-\dot{\boldsymbol{\eta}}(0) + P_{T_{\boldsymbol{y}}\mathcal{M}}\int_{a=0}^1 \dot{\boldsymbol{\eta}}(a)da\right\|_2$$

$$= \left\|-\dot{\boldsymbol{\eta}}(0) + P_{T_{\boldsymbol{\eta}(0)}\mathcal{M}}\int_{a=0}^1\left(\dot{\boldsymbol{\eta}}(0) + \int_{b=0}^a \ddot{\boldsymbol{\eta}}(b)db\right)da\right\|_2$$

$$= \left\|P_{T_{\boldsymbol{\eta}(0)}\mathcal{M}}\int_{a=0}^1\int_{b=0}^a \ddot{\boldsymbol{\eta}}(b)dbda\right\|_2 \qquad (D.7)$$

$$\leq \int_{a=0}^1\int_{b=0}^a \left\|\ddot{\boldsymbol{\eta}}(b)\right\|_2 dbda$$

$$\leq \int_{a=0}^1\int_{b=0}^a \kappa\|\boldsymbol{v}_{\boldsymbol{\eta}}\|_2^2 dbda$$

$$= \frac{1}{2}\kappa d_{\mathcal{M}}^2(\boldsymbol{y}, \boldsymbol{x}_\natural).$$

Since $\boldsymbol{y} \in B_{\mathcal{M}}(\boldsymbol{x}_\natural, 1/\kappa)$, we have

$$\left\|P_{T_{\boldsymbol{y}}\mathcal{M}}\boldsymbol{z}\right\|_2 \leq T_{\max}. \qquad (D.8)$$

Combining the above, we end up with the result. ∎

**Lemma D.3.** *For any point $\boldsymbol{z}$ and $\boldsymbol{x}_\natural, \boldsymbol{y} \in \mathcal{M}$, let*

$$T_{\max} = \sup_{\boldsymbol{y}\in B_{\mathcal{M}}(\boldsymbol{x}_\natural, 1/\kappa), \boldsymbol{v}\in T_{\boldsymbol{y}}\mathcal{M}, \|\boldsymbol{v}\|_2=1} \langle \boldsymbol{v}, \boldsymbol{z}\rangle, \qquad (D.9)$$

*if $d_{\mathcal{M}}(\boldsymbol{y}, \boldsymbol{x}_\natural) \leq 1/\kappa$, then we have*

$$\left\|\mathrm{grad}[\varphi_{\boldsymbol{x}}](\boldsymbol{y})\right\|_2 \leq d_{\mathcal{M}}(\boldsymbol{y}, \boldsymbol{x}_\natural) + T_{\max}. \qquad (D.10)$$

**Proof.**

$$\left\|\mathrm{grad}[\varphi_{\boldsymbol{x}}](\boldsymbol{y})\right\|_2 = \left\|P_{T_{\boldsymbol{y}}\mathcal{M}}(\boldsymbol{y} - \boldsymbol{x})\right\|_2$$

$$\leq \left\|P_{T_{\boldsymbol{y}}\mathcal{M}}(\boldsymbol{y} - \boldsymbol{x}_\natural)\right\|_2 + \left\|P_{T_{\boldsymbol{y}}\mathcal{M}}\boldsymbol{z}\right\|_2. \qquad (D.11)$$

It's easy to show that $\left\|P_{T_{\boldsymbol{y}}\mathcal{M}}(\boldsymbol{y} - \boldsymbol{x}_\natural)\right\|_2 \leq d_{\mathcal{M}}(\boldsymbol{y}, \boldsymbol{x}_\natural)$. Let

$$T_{\max} = \sup_{\boldsymbol{y}\in B_{\mathcal{M}}(\boldsymbol{x}_\natural, 1/\kappa), \boldsymbol{v}\in T_{\boldsymbol{y}}\mathcal{M}, \|\boldsymbol{v}\|_2=1} \langle \boldsymbol{v}, \boldsymbol{z}\rangle, \qquad (D.12)$$

then we have

$$\left\|P_{T_{\boldsymbol{y}}\mathcal{M}}\boldsymbol{z}\right\|_2 \leq T_{\max}. \qquad (D.13)$$

Combining the above two bounds, we end up with the desired result. ∎

**Lemma D.4.** *Let $\zeta : [0,1] \to \mathcal{M}$ be a geodesic in a Riemannian manifold $\mathcal{M}$. Take any initial vector $\boldsymbol{v}_0 \in T_{\zeta(0)}\mathcal{M}$, and let $\boldsymbol{v}_t$ be its parallel transport along $\zeta$ up to time $t$. Then*

$$\|\boldsymbol{v}_t - \boldsymbol{v}_0\|_2 \leq 3\kappa t\|\boldsymbol{v}_0\|_2\|\dot{\zeta}\|_2. \qquad (D.14)$$

**Proof.** When paralleling transport a $\boldsymbol{v}_0 \in T_{\zeta(0)}\mathcal{M}$ along the geodesic $\zeta$, we have

$$\frac{d}{dt}\boldsymbol{v}_t = \Pi(\boldsymbol{v}_t, \dot{\zeta}(t)). \qquad (D.15)$$

From fundamental theorem of calculus, we have

$$\boldsymbol{v}_t = \boldsymbol{v}_0 + \int_{a=0}^t \Pi(\boldsymbol{v}_a, \dot{\zeta}(a))da. \qquad (D.16)$$

From above and applying Lemma 8 in (Yan* et al., 2023), we have

$$\begin{aligned}
\|\boldsymbol{v}_t - \boldsymbol{v}_0\|_2 &= \left\|\int_{a=0}^t \Pi(\boldsymbol{v}_a, \dot{\boldsymbol{\zeta}}(a)) da\right\|_2 \\
&\leq \int_{a=0}^t \left\|\Pi(\boldsymbol{v}_a, \dot{\boldsymbol{\zeta}}(a))\right\|_2 da \\
&\leq \int_{a=0}^t 3\kappa \|\boldsymbol{v}_0\|_2 \|\dot{\boldsymbol{\zeta}}\|_2 da \\
&= 3\kappa t \|\boldsymbol{v}_0\|_2 \|\dot{\boldsymbol{\zeta}}\|_2.
\end{aligned} \tag{D.17}$$

∎

**Lemma D.5.** *Let $\boldsymbol{\gamma} : [0,1] \to \mathcal{M}$ be a geodesic joining $\boldsymbol{q}_u$ and $\boldsymbol{q}_v$ with constant speed, where $\boldsymbol{\gamma}(0) = \boldsymbol{q}_u, \boldsymbol{\gamma}(1) = \boldsymbol{q}_v$. Then, for all $t \in [0,1]$, we have*

$$\left\|P_{T_{\boldsymbol{\gamma}(t)}\mathcal{M}} - P_{T_{\boldsymbol{\gamma}(0)}\mathcal{M}}\right\|_{op} \leq 3\sqrt{2}\kappa t \|\dot{\boldsymbol{\gamma}}\|_2. \tag{D.18}$$

**Proof.**

$$\begin{aligned}
\left\|P_{T_{\boldsymbol{\gamma}(t)}\mathcal{M}} - P_{T_{\boldsymbol{\gamma}(0)}\mathcal{M}}\right\|_{op} &= \sup_{\|\boldsymbol{w}\|_2=1} \left\|\left(P_{T_{\boldsymbol{\gamma}(t)}\mathcal{M}} - P_{T_{\boldsymbol{\gamma}(0)}\mathcal{M}}\right)\boldsymbol{w}\right\|_2 \\
&= \sup_{\|\boldsymbol{w}\|_2=1} \left\|\left(P_{T_{\boldsymbol{\gamma}(t)}\mathcal{M}} - P_{T_{\boldsymbol{\gamma}(0)}\mathcal{M}}\right)(\boldsymbol{w}_{||} + \boldsymbol{w}_{\perp})\right\|_2 \\
&\leq \sup_{\|\boldsymbol{w}\|_2=1} \left\|\left(P_{T_{\boldsymbol{\gamma}(t)}\mathcal{M}} - P_{T_{\boldsymbol{\gamma}(0)}\mathcal{M}}\right)\boldsymbol{w}_{||}\right\|_2 + \left\|\left(P_{T_{\boldsymbol{\gamma}(t)}\mathcal{M}} - P_{T_{\boldsymbol{\gamma}(0)}\mathcal{M}}\right)\boldsymbol{w}_{\perp}\right\|_2 \\
&= \sup_{\|\boldsymbol{w}\|_2=1} \left\|P_{T_{\boldsymbol{\gamma}(t)}\mathcal{M}}\boldsymbol{w}_{||} - \boldsymbol{w}_{||}\right\|_2 + \left\|P_{T_{\boldsymbol{\gamma}(t)}\mathcal{M}}\boldsymbol{w}_{\perp}\right\|_2,
\end{aligned} \tag{D.19}$$

where $\boldsymbol{w}_{||} = P_{T_{\boldsymbol{\gamma}(0)}\mathcal{M}}\boldsymbol{w}, \boldsymbol{w}_{\perp} = P_{T_{\boldsymbol{\gamma}(0)}^{\perp}\mathcal{M}}\boldsymbol{w}$.

Together with Lemma D.4, we bound the first term $\left\|\left(P_{T_{\boldsymbol{\gamma}(t)}\mathcal{M}} - P_{T_{\boldsymbol{\gamma}(0)}\mathcal{M}}\right)\boldsymbol{w}_{||}\right\|_2$.

$$\begin{aligned}
\left\|P_{T_{\boldsymbol{\gamma}(t)}\mathcal{M}}\boldsymbol{w}_{||} - \boldsymbol{w}_{||}\right\|_2 &= \min_{\boldsymbol{h} \in T_{\boldsymbol{\gamma}(t)}\mathcal{M}} \|\boldsymbol{h} - \boldsymbol{w}_{||}\|_2 \\
&\leq \left\|\Pi_{0 \to t}\boldsymbol{w}_{||} - \boldsymbol{w}_{||}\right\|_2 \\
&\leq 3\kappa t \|\boldsymbol{w}_{||}\|_2 \|\dot{\boldsymbol{\gamma}}\|_2.
\end{aligned} \tag{D.20}$$

Here $\Pi_{0 \to t}$ is a transport operator which transports a vector in $T_{\boldsymbol{\gamma}(0)}\mathcal{M}$ to the $T_{\boldsymbol{\gamma}(t)}\mathcal{M}$ along the geodesic $\boldsymbol{\gamma}$.

We next bound the second term $\left\|P_{T_{\boldsymbol{\gamma}(t)}\mathcal{M}}\boldsymbol{w}_{\perp}\right\|_2$.

$$\begin{aligned}
\left\|P_{T_{\boldsymbol{\gamma}(t)}\mathcal{M}}\boldsymbol{w}_{\perp}\right\|_2 &= \left\|P_{T_{\boldsymbol{\gamma}(t)}\mathcal{M}}P_{T_{\boldsymbol{\gamma}(0)}^{\perp}\mathcal{M}}\boldsymbol{w}_{\perp}\right\|_2 \\
&\leq \|\boldsymbol{w}_{\perp}\|_2 \left\|P_{T_{\boldsymbol{\gamma}(t)}\mathcal{M}}P_{T_{\boldsymbol{\gamma}(0)}^{\perp}\mathcal{M}}\right\|_{op} \\
&= \|\boldsymbol{w}_{\perp}\|_2 \left\|P_{T_{\boldsymbol{\gamma}(0)}^{\perp}\mathcal{M}}P_{T_{\boldsymbol{\gamma}(t)}\mathcal{M}}\right\|_{op} \\
&= \|\boldsymbol{w}_{\perp}\|_2 \sup_{\|\boldsymbol{u}\|_2=1} \left\|P_{T_{\boldsymbol{\gamma}(0)}^{\perp}\mathcal{M}}P_{T_{\boldsymbol{\gamma}(t)}\mathcal{M}}\boldsymbol{u}\right\|_2 \\
&= \|\boldsymbol{w}_{\perp}\|_2 \sup_{\|\boldsymbol{u}\|_2=1, \boldsymbol{u} \in T_{\boldsymbol{\gamma}(t)}\mathcal{M}} \left\|P_{T_{\boldsymbol{\gamma}(0)}^{\perp}\mathcal{M}}\boldsymbol{u}\right\|_2 \\
&= \|\boldsymbol{w}_{\perp}\|_2 \sup_{\|\boldsymbol{u}\|_2=1, \boldsymbol{u} \in T_{\boldsymbol{\gamma}(t)}\mathcal{M}} \left\|\left(\boldsymbol{I} - P_{T_{\boldsymbol{\gamma}(0)}\mathcal{M}}\right)\boldsymbol{u}\right\|_2 \\
&= \|\boldsymbol{w}_{\perp}\|_2 \sup_{\|\boldsymbol{u}\|_2=1, \boldsymbol{u} \in T_{\boldsymbol{\gamma}(t)}\mathcal{M}} \min_{\boldsymbol{h} \in T_{\boldsymbol{\gamma}(0)}\mathcal{M}} \|\boldsymbol{h} - \boldsymbol{u}\|_2 \\
&\leq \|\boldsymbol{w}_{\perp}\|_2 \sup_{\|\boldsymbol{u}\|_2=1, \boldsymbol{u} \in T_{\boldsymbol{\gamma}(t)}\mathcal{M}} \|\Pi_{t \to 0}\boldsymbol{u} - \boldsymbol{u}\|_2 \\
&\leq 3\kappa t \|\boldsymbol{w}_{\perp}\|_2 \|\dot{\boldsymbol{\gamma}}\|_2.
\end{aligned} \tag{D.21}$$

Therefore,

$$
\sup_{\|\boldsymbol{w}\|_2=1} \left\|P_{T_{\boldsymbol{\gamma}(t)}\mathcal{M}}\boldsymbol{w}_{||} - \boldsymbol{w}_{||}\right\|_2 + \left\|P_{T_{\boldsymbol{\gamma}(t)}\mathcal{M}}\boldsymbol{w}_{\perp}\right\|_2 \leq \sup_{\|\boldsymbol{w}\|_2=1} 3\kappa t\|\dot{\boldsymbol{\gamma}}\|_2(\|\boldsymbol{w}_{||}\|_2 + \|\boldsymbol{w}_{\perp}\|_2)
$$

$$
\leq \sup_{\|\boldsymbol{w}\|_2=1} 3\sqrt{2}\kappa t\|\dot{\boldsymbol{\gamma}}\|_2\|\boldsymbol{w}\|_2 \tag{D.22}
$$

$$
= 3\sqrt{2}\kappa t\|\dot{\boldsymbol{\gamma}}\|_2.
$$

Hence

$$
\left\|P_{T_{\boldsymbol{\gamma}(t)}\mathcal{M}} - P_{T_{\boldsymbol{\gamma}(0)}\mathcal{M}}\right\|_{op} \leq 3\sqrt{2}\kappa t\|\dot{\boldsymbol{\gamma}}\|_2. \tag{D.23}
$$

∎

**Lemma D.6.** *Let $\boldsymbol{\gamma} : [0,1] \to \mathcal{M}$ be a minimum length geodesic joining $\boldsymbol{q}_u$ and $\boldsymbol{q}_{u^+}$ with constant speed $d_{\mathcal{M}}(\boldsymbol{q}_u, \boldsymbol{q}_{u^+})$, where $\boldsymbol{\gamma}(0) = \boldsymbol{q}_u, \boldsymbol{\gamma}(1) = \boldsymbol{q}_{u^+}$. Suppose for any $t \in [0,1]$, $d_{\mathcal{M}}(\boldsymbol{\gamma}(t), \boldsymbol{x}_\natural) \leq 1/\kappa$, then, we have*

$$
\langle \mathrm{grad}[\varphi_{\boldsymbol{x}}](\boldsymbol{\gamma}(t)), \boldsymbol{\xi}_{uu^+} \rangle \leq -\frac{0.55}{2}\left(\frac{1}{2}d_{\mathcal{M}}(\boldsymbol{x}_\natural, \boldsymbol{\gamma}(0)) - T_{\max}\right)d_{\mathcal{M}}(\boldsymbol{q}_u, \boldsymbol{q}_{u^+})
$$

$$
+ \left(3\sqrt{2}\kappa t d_{\mathcal{M}}(\boldsymbol{q}_u, \boldsymbol{q}_{u^+})d_{\mathcal{M}}(\boldsymbol{\gamma}(t), \boldsymbol{x}_\natural) + 2T_{\max} + t d_{\mathcal{M}}(\boldsymbol{q}_u, \boldsymbol{q}_{u^+}) + \frac{1}{2}\kappa t^2 d_{\mathcal{M}}^2(\boldsymbol{q}_u, \boldsymbol{q}_{u^+})\right)d_{\mathcal{M}}(\boldsymbol{q}_u, \boldsymbol{q}_{u^+}). \tag{D.24}
$$

**Proof.** We note that

$$
\langle \mathrm{grad}[\varphi_{\boldsymbol{x}}](\boldsymbol{\gamma}(t)), \boldsymbol{\xi}_{uu^+} \rangle = \langle \mathrm{grad}[\varphi_{\boldsymbol{x}}](\boldsymbol{\gamma}(0)), \boldsymbol{\xi}_{uu^+} \rangle + \langle \mathrm{grad}[\varphi_{\boldsymbol{x}}](\boldsymbol{\gamma}(t)) - \mathrm{grad}[\varphi_{\boldsymbol{x}}](\boldsymbol{\gamma}(0)), \boldsymbol{\xi}_{uu^+} \rangle
$$
$$
\leq \langle \mathrm{grad}[\varphi_{\boldsymbol{x}}](\boldsymbol{\gamma}(0)), \boldsymbol{\xi}_{uu^+} \rangle + \|\mathrm{grad}[\varphi_{\boldsymbol{x}}](\boldsymbol{\gamma}(t)) - \mathrm{grad}[\varphi_{\boldsymbol{x}}](\boldsymbol{\gamma}(0))\|_2\, d_{\mathcal{M}}(\boldsymbol{q}_u, \boldsymbol{q}_{u^+}). \tag{D.25}
$$

For any $\boldsymbol{x} \neq \boldsymbol{q}_u$, our first-order step rule is equivalent of choosing

$$
u^+ = \arg\min_{v:u\xrightarrow{1}v\in E^1}\left\|R_a\frac{\mathcal{P}_{T_u}(\boldsymbol{x} - \boldsymbol{q}_u)}{\|\mathcal{P}_{T_u}(\boldsymbol{x} - \boldsymbol{q}_u)\|_2} - \mathcal{P}_{T_u}(\boldsymbol{q}_v - \boldsymbol{q}_u)\right\|_2.
$$

Then, lemma B.3 guarantees that there exists a $v$ satisfies Equation (B.17). From our step rule, we know that $q_{u^+}$ also satisfies Equation (B.17). Together with Lemma B.4 and $d_{\mathcal{M}}(\boldsymbol{\gamma}(0), \boldsymbol{x}_\natural) \leq 1/\kappa$, we have

$$
\langle \mathrm{grad}[\varphi_{\boldsymbol{x}}](\boldsymbol{\gamma}(0)), \boldsymbol{\xi}_{uu^+} \rangle \leq -0.55\,\|\mathrm{grad}[\varphi_{\boldsymbol{x}}](\boldsymbol{\gamma}(0))\|_2\,\|\boldsymbol{\xi}_{uu^+}\|_2
$$

$$
\leq -0.55\left(\left\|P_{T_{\boldsymbol{\gamma}(0)}\mathcal{M}}(\boldsymbol{x}_\natural - \boldsymbol{\gamma}(0))\right\|_2 - \left\|P_{T_{\boldsymbol{\gamma}(0)}\mathcal{M}}\boldsymbol{z}\right\|_2\right)\|\boldsymbol{\xi}_{uu^+}\|_2
$$

$$
\leq -0.55\left(d_{\mathcal{M}}(\boldsymbol{\gamma}(0), \boldsymbol{x}_\natural)\left(1 - \frac{1}{2}\kappa d_{\mathcal{M}}(\boldsymbol{\gamma}(0), \boldsymbol{x}_\natural)\right) - T_{\max}\right)\frac{1}{2}d_{\mathcal{M}}(\boldsymbol{q}_u, \boldsymbol{q}_{u^+}) \tag{D.26}
$$

$$
\leq \frac{-0.55}{2}\left(\frac{1}{2}d_{\mathcal{M}}(\boldsymbol{\gamma}(0), \boldsymbol{x}_\natural) - T_{\max}\right)d_{\mathcal{M}}(\boldsymbol{q}_u, \boldsymbol{q}_{u^+}).
$$

where from the second to the third line we've used the same argument from equation (A.28) to lower bound $\left\|P_{T_{\boldsymbol{\gamma}(0)}\mathcal{M}}(\boldsymbol{x}_\natural - \boldsymbol{\gamma}(0))\right\|_2$ by $d_{\mathcal{M}}(\boldsymbol{\gamma}(0), \boldsymbol{x}_\natural)\left(1 - \frac{1}{2}\kappa d_{\mathcal{M}}(\boldsymbol{\gamma}(0), \boldsymbol{x}_\natural)\right)$ and $\|\boldsymbol{\xi}_{uu^+}\|_2$ by $\frac{1}{2}d_{\mathcal{M}}(\boldsymbol{q}_u, \boldsymbol{q}_{u^+})$ And we can bound $\|\mathrm{grad}[\varphi_{\boldsymbol{x}}](\boldsymbol{\gamma}(t)) - \mathrm{grad}[\varphi_{\boldsymbol{x}}](\boldsymbol{\gamma}(0))\|_2$:

$$
\|\mathrm{grad}[\varphi_{\boldsymbol{x}}](\boldsymbol{\gamma}(t)) - \mathrm{grad}[\varphi_{\boldsymbol{x}}](\boldsymbol{\gamma}(0))\|_2 \leq \left\|\left(P_{T_{\boldsymbol{\gamma}(t)}\mathcal{M}} - P_{T_{\boldsymbol{\gamma}(0)}\mathcal{M}}\right)(\boldsymbol{\gamma}(t) - \boldsymbol{x}_\natural)\right\|_2
$$

$$
+ \left\|\left(P_{T_{\boldsymbol{\gamma}(t)}\mathcal{M}} - P_{T_{\boldsymbol{\gamma}(0)}\mathcal{M}}\right)\boldsymbol{z}\right\|_2
$$

$$
+ \left\|P_{T_{\boldsymbol{\gamma}(0)}\mathcal{M}}(\boldsymbol{\gamma}(t) - \boldsymbol{\gamma}(0))\right\|_2
$$

$$
\leq \left\|P_{T_{\boldsymbol{\gamma}(t)}\mathcal{M}} - P_{T_{\boldsymbol{\gamma}(0)}\mathcal{M}}\right\|_{op} d_{\mathcal{M}}(\boldsymbol{\gamma}(t), \boldsymbol{x}_\natural) \tag{D.27}
$$

$$
+ \left\|P_{T_{\boldsymbol{\gamma}(t)}\mathcal{M}}\boldsymbol{z}\right\|_2 + \left\|P_{T_{\boldsymbol{\gamma}(0)}\mathcal{M}}\boldsymbol{z}\right\|_2
$$

$$
+ \left\|P_{T_{\boldsymbol{\gamma}(0)}\mathcal{M}}(\boldsymbol{\gamma}(t) - \boldsymbol{\gamma}(0))\right\|_2.
$$

Together with Lemma D.5, we have

$$\left\|P_{T_{\gamma(t)}\mathcal{M}} - P_{T_{\gamma(0)}\mathcal{M}}\right\|_{op} d_{\mathcal{M}}(\gamma(t), \boldsymbol{x}_{\natural}) \le 3\sqrt{2}\kappa t d_{\mathcal{M}}(\boldsymbol{q}_u, \boldsymbol{q}_{u^+})d_{\mathcal{M}}(\gamma(t), \boldsymbol{x}_{\natural}). \tag{D.28}$$

Since $\gamma(t) \in B_{\mathcal{M}}(\boldsymbol{x}_{\natural}, 1/\kappa) \quad \forall t \in [0, 1]$, we have

$$\left\|P_{T_{\gamma(t)}\mathcal{M}}\boldsymbol{z}\right\|_2 \le T_{\max}, \tag{D.29}$$

hence

$$\left\|P_{T_{\gamma(t)}\mathcal{M}}\boldsymbol{z}\right\|_2 + \left\|P_{T_{\gamma(0)}\mathcal{M}}\boldsymbol{z}\right\|_2 \le 2T_{\max}. \tag{D.30}$$

From Taylor expansion on the geodesic $\gamma$, we have

$$\begin{aligned}
\left\|P_{T_{\gamma(0)}\mathcal{M}}(\gamma(t) - \gamma(0))\right\|_2 &= \left\|P_{T_{\gamma(0)}\mathcal{M}}\int_{a=0}^{t}\dot{\gamma}(a)da\right\|_2 \\
&= \left\|P_{T_{\gamma(0)}\mathcal{M}}\int_{a=0}^{t}\left(\dot{\gamma}(0) + \int_{b=0}^{a}\ddot{\gamma}(b)db\right)da\right\|_2 \\
&= \left\|\dot{\gamma}(0)t + \int_{a=0}^{t}\int_{b=0}^{a}P_{T_{\gamma(0)}\mathcal{M}}\ddot{\gamma}(b)dbda\right\|_2 \\
&\le td_{\mathcal{M}}(\boldsymbol{q}_u, \boldsymbol{q}_{u^+}) + \frac{1}{2}\kappa t^2 d_{\mathcal{M}}^2(\boldsymbol{q}_u, \boldsymbol{q}_{u^+}).
\end{aligned} \tag{D.31}$$

Therefore,

$$\|\text{grad}[\varphi_{\boldsymbol{x}}](\gamma(t)) - \text{grad}[\varphi_{\boldsymbol{x}}](\gamma(0))\|_2 \le 3\sqrt{2}\kappa t d_{\mathcal{M}}(\boldsymbol{q}_u, \boldsymbol{q}_{u^+})d_{\mathcal{M}}(\gamma(t), \boldsymbol{x}_{\natural}) + 2T_{\max} + td_{\mathcal{M}}(\boldsymbol{q}_u, \boldsymbol{q}_{u^+}) + \frac{1}{2}\kappa t^2 d_{\mathcal{M}}^2(\boldsymbol{q}_u, \boldsymbol{q}_{u^+}). \tag{D.32}$$

Combining all of things above, we have

$$\begin{aligned}
\langle\text{grad}[\varphi_{\boldsymbol{x}}](\gamma(t)), \boldsymbol{\xi}_{uu^+}\rangle &\le \frac{-0.55}{2}\left(\frac{1}{2}d_{\mathcal{M}}(\boldsymbol{x}_{\natural}, \gamma(0)) - T_{\max}\right)d_{\mathcal{M}}(\boldsymbol{q}_u, \boldsymbol{q}_{u^+}) \\
&\quad + \left(3\sqrt{2}\kappa t d_{\mathcal{M}}(\boldsymbol{q}_u, \boldsymbol{q}_{u^+})d_{\mathcal{M}}(\gamma(t), \boldsymbol{x}_{\natural}) + 2T_{\max} + td_{\mathcal{M}}(\boldsymbol{q}_u, \boldsymbol{q}_{u^+}) + \frac{1}{2}\kappa t^2 d_{\mathcal{M}}^2(\boldsymbol{q}_u, \boldsymbol{q}_{u^+})\right)d_{\mathcal{M}}(\boldsymbol{q}_u, \boldsymbol{q}_{u^+}).
\end{aligned} \tag{D.33}$$

∎

**Lemma D.7.** *Suppose*

$$\boldsymbol{q}_u \in B_{\mathcal{M}}(\boldsymbol{x}_{\natural}, c/\kappa) \setminus B_{\mathcal{M}}(\boldsymbol{x}_{\natural}, CT_{\max}), \tag{D.34}$$

*for some constant $c \le 0.1$, $C \ge 321$, with*

$$T_{\max} = \sup_{\boldsymbol{y} \in B_{\mathcal{M}}(\boldsymbol{x}_{\natural}, 1/\kappa), \boldsymbol{v} \in T_{\boldsymbol{y}}\mathcal{M}, \|\boldsymbol{v}\|_2=1} \langle\boldsymbol{v}, \boldsymbol{z}\rangle, \tag{D.35}$$

*and*

$$d_{\mathcal{M}}(\boldsymbol{q}_u, \boldsymbol{q}_{u^+}) \le \min\left\{\frac{1}{100\kappa}, \frac{1}{8}T_{\max}\right\}, \tag{D.36}$$

*then*

$$d_{\mathcal{M}}(\boldsymbol{q}_{u^+}, \boldsymbol{x}_{\natural}) \le d_{\mathcal{M}}(\boldsymbol{q}_u, \boldsymbol{x}_{\natural}) - \frac{1}{80}d_{\mathcal{M}}(\boldsymbol{q}_u, \boldsymbol{q}_{u^+}). \tag{D.37}$$

**Proof.** Let $\gamma : [0, 1] \to \mathcal{M}$ be a minimum length geodesic joining $\boldsymbol{q}_u$ and $\boldsymbol{q}_{u^+}$ with constant speed $d_{\mathcal{M}}(\boldsymbol{q}_u, \boldsymbol{q}_{u^+})$, where $\gamma(0) = \boldsymbol{q}_u, \gamma(1) = \boldsymbol{q}_{u^+}$. Then we have

$$\begin{aligned}
d_{\mathcal{M}}(\boldsymbol{x}_{\natural}, \boldsymbol{q}_{u^+}) - d_{\mathcal{M}}(\boldsymbol{x}_{\natural}, \boldsymbol{q}_u) &= \int_{t=0}^{1}\frac{d}{dt}d_{\mathcal{M}}(\boldsymbol{x}_{\natural}, \gamma(t))dt \\
&= \int_{t=0}^{1}\left\langle\frac{d}{d\boldsymbol{y}}d_{\mathcal{M}}(\boldsymbol{x}_{\natural}, \boldsymbol{y})\Big|_{\boldsymbol{y}=\gamma(t)}, \dot{\gamma}(t)\right\rangle dt \\
&= \int_{t=0}^{1}\left\langle\frac{-\log_{\gamma(t)}\boldsymbol{x}_{\natural}}{\|\log_{\gamma(t)}\boldsymbol{x}_{\natural}\|_2}, \dot{\gamma}(t)\right\rangle dt.
\end{aligned} \tag{D.38}$$

We can further decompose the integrand as follows:

$$\left\langle \frac{-\log_{\gamma(t)} x_\natural}{\|\log_{\gamma(t)} x_\natural\|_2}, \dot{\gamma}(t) \right\rangle$$

$$= \frac{1}{\|\log_{\gamma(t)} x_\natural\|_2} \left\langle -\log_{\gamma(t)} x_\natural + \mathrm{grad}[\varphi_x](\gamma(t)) - \mathrm{grad}[\varphi_x](\gamma(t)), \dot{\gamma}(t) + \xi_{uu^+} - \xi_{uu^+} \right\rangle$$

$$= \frac{1}{d_{\mathcal{M}}(\gamma(t), x_\natural)} \left( \langle \mathrm{grad}[\varphi_x](\gamma(t)), \xi_{uu^+} \rangle + \left\langle -\log_{\gamma(t)} x_\natural - \mathrm{grad}[\varphi_x](\gamma(t)), \dot{\gamma}(t) \right\rangle + \langle \mathrm{grad}[\varphi_x](\gamma(t)), \dot{\gamma}(t) - \xi_{uu^+} \rangle \right).$$
(D.39)

We will proceed to use Lemma D.6 to bound the first term, Lemma D.2 to bound the second term, and Lemmas D.1 and D.3 to bound the last term. In order to apply these lemmas, we observe that $\forall t \in [0,1], d_{\mathcal{M}}(\gamma(t), x_\natural) \leq d_{\mathcal{M}}(\gamma(0), x_\natural) + d_{\mathcal{M}}(\gamma(0), \gamma(t)) \leq d_{\mathcal{M}}(\gamma(0), x_\natural) + d_{\mathcal{M}}(\gamma(0), \gamma(1)) \leq \frac{c}{\kappa} + \frac{1}{100\kappa} \leq \frac{1}{8\kappa}$.

Thus from Lemma D.6 we have

$$\langle \mathrm{grad}[\varphi_x](\gamma(t)), \xi_{uu^+} \rangle \leq \frac{-0.55}{2} \left( \frac{1}{2} d_{\mathcal{M}}(x_\natural, \gamma(0)) - T_{\max} \right) d_{\mathcal{M}}(q_u, q_{u^+})$$

$$+ \left( 3\sqrt{2}\kappa t d_{\mathcal{M}}(q_u, q_{u^+}) d_{\mathcal{M}}(\gamma(t), x_\natural) + 2T_{\max} + t d_{\mathcal{M}}(q_u, q_{u^+}) + \frac{1}{2}\kappa t^2 d_{\mathcal{M}}^2(q_u, q_{u^+}) \right) d_{\mathcal{M}}(q_u, q_{u^+})$$

$$\leq \left( -\frac{1}{8} d_{\mathcal{M}}(x_\natural, \gamma(t)) + \frac{1}{8} d_{\mathcal{M}}(q_u, q_{u^+}) + \frac{1}{2} T_{\max} \right) d_{\mathcal{M}}(q_u, q_{u^+})$$

$$+ \left( \frac{3\sqrt{2}}{8} d_{\mathcal{M}}(q_u, q_{u^+}) + 2T_{\max} + d_{\mathcal{M}}(q_u, q_{u^+}) + \frac{1}{200} d_{\mathcal{M}}(q_u, q_{u^+}) \right) d_{\mathcal{M}}(q_u, q_{u^+})$$

$$\leq \left( -\frac{1}{8} d_{\mathcal{M}}(x_\natural, \gamma(t)) + \frac{5}{2} T_{\max} + 2 d_{\mathcal{M}}(q_u, q_{u^+}) \right) d_{\mathcal{M}}(q_u, q_{u^+}).$$
(D.40)

Similarly, from Lemma D.2 we have

$$\left\langle -\log_{\gamma(t)} x_\natural - \mathrm{grad}[\varphi_x](\gamma(t)), \dot{\gamma}(t) \right\rangle \leq \left\| -\log_{\gamma(t)} x_\natural - \mathrm{grad}[\varphi_x](\gamma(t)) \right\| \cdot \|\dot{\gamma}(t)\|$$

$$\leq \left( \frac{1}{2}\kappa d_{\mathcal{M}}^2(\gamma(t), x_\natural) + T_{\max} \right) \cdot d_{\mathcal{M}}(q_u, q_{u^+})$$
(D.41)

$$\leq \left( \frac{1}{16} d_{\mathcal{M}}(\gamma(t), x_\natural) + T_{\max} \right) \cdot d_{\mathcal{M}}(q_u, q_{u^+}),$$

and Lemmas D.1 and D.3 also give

$$\langle \mathrm{grad}[\varphi_x](\gamma(t)), \dot{\gamma}(t) - \xi_{uu^+} \rangle \leq \|\mathrm{grad}[\varphi_x](\gamma(t))\| \cdot \|\dot{\gamma}(t) - \xi_{uu^+}\|$$

$$\leq (d_{\mathcal{M}}(\gamma(t), x_\natural) + T_{\max}) \cdot \left( \frac{3\kappa}{2} d_{\mathcal{M}}^2(q_u, q_{u^+}) \right)$$
(D.42)

$$\leq \left( \frac{3}{200} d_{\mathcal{M}}(\gamma(t), x_\natural) + \frac{3}{200} T_{\max} \right) d_{\mathcal{M}}(q_u, q_{u^+}).$$

Lastly, we combine the terms to get

$$\langle \mathrm{grad}[\varphi_x](\gamma(t)), \xi_{uu^+} \rangle + \left\langle -\log_{\gamma(t)} x_\natural - \mathrm{grad}[\varphi_x](\gamma(t)), \dot{\gamma}(t) \right\rangle + \langle \mathrm{grad}[\varphi_x](\gamma(t)), \dot{\gamma}(t) - \xi_{uu^+} \rangle$$

$$\leq \left( -\frac{1}{40} d_{\mathcal{M}}(\gamma(t), x_\natural) + \frac{15}{4} T_{\max} + 2 d_{\mathcal{M}}(q_u, q_{u^+}) \right) d_{\mathcal{M}}(q_u, q_{u^+})$$

$$\leq \left( -\frac{1}{40} d_{\mathcal{M}}(\gamma(t), x_\natural) + 4T_{\max} \right) d_{\mathcal{M}}(q_u, q_{u^+})$$
(D.43)

$$\leq \left( -\frac{1}{80} d_{\mathcal{M}}(\gamma(t), x_\natural) \right) d_{\mathcal{M}}(q_u, q_{u^+}),$$

where we've used our assumption that $d_{\mathcal{M}}(\boldsymbol{q}_u, \boldsymbol{q}_{u^+}) \leq \frac{1}{8}T_{\max}$ and $d_{\mathcal{M}}(\boldsymbol{\gamma}(t), \boldsymbol{x}_{\natural}) \geq d_{\mathcal{M}}(\boldsymbol{\gamma}(0), \boldsymbol{x}_{\natural}) - d_{\mathcal{M}}(\boldsymbol{q}_u, \boldsymbol{q}_{u^+}) \geq CT_{\max} - \frac{1}{8}T_{\max} \geq 320T_{\max}$. Finally, plugging this result back to Equation D.39, we observe

$$
\begin{aligned}
d_{\mathcal{M}}(\boldsymbol{x}_{\natural}, \boldsymbol{q}_{u^+}) - d_{\mathcal{M}}(\boldsymbol{x}_{\natural}, \boldsymbol{q}_u) &= \int_{t=0}^{1} \left\langle \frac{-\log_{\boldsymbol{\gamma}(t)} \boldsymbol{x}_{\natural}}{\| \log_{\boldsymbol{\gamma}(t)} \boldsymbol{x}_{\natural} \|_2}, \dot{\boldsymbol{\gamma}}(t) \right\rangle dt \\
&\leq \int_{t=0}^{1} \frac{1}{d_{\mathcal{M}}(\boldsymbol{\gamma}(t), \boldsymbol{x}_{\natural})} \left( -\frac{1}{80} d_{\mathcal{M}}(\boldsymbol{\gamma}(t), \boldsymbol{x}_{\natural}) \right) d_{\mathcal{M}}(\boldsymbol{q}_u, \boldsymbol{q}_{u^+}) dt \\
&\leq -\frac{1}{80} d_{\mathcal{M}}(\boldsymbol{q}_u, \boldsymbol{q}_{u^+}).
\end{aligned}
\tag{D.44}
$$

This completes the proof ∎

# A. Supporting Results

**Preliminiaries on the logarithmic map.** The following sequence of lemmas provides an upper bound on the number of landmarks $|Q|$, under the assumption that the landmarks are $\delta$-separated. Our argument will assume that the manifold $\mathcal{M}$ is *connected* and *geodesically complete*. Under these assumptions, the exponential map

$$
\exp_{\boldsymbol{x}_{\natural}}(\cdot) : T_{\boldsymbol{x}_{\natural}}\mathcal{M} \to \mathcal{M}
\tag{A.1}
$$

is surjective, i.e., for every $\boldsymbol{q} \in \mathcal{M}$, there exists $\boldsymbol{v} \in T_{\boldsymbol{x}_{\natural}}\mathcal{M}$ such that

$$
\exp_{\boldsymbol{x}_{\natural}}(\boldsymbol{v}) = \boldsymbol{q}.
\tag{A.2}
$$

Moreover, by the Hopf-Rinow theorem, there exists a length-minimizing geodesic joining $\boldsymbol{x}_{\natural}$ and $\boldsymbol{q}$, and hence there exists $\boldsymbol{v} \in T_{\boldsymbol{x}_{\natural}}\mathcal{M}$ of norm $\|\boldsymbol{v}\| = d_{\mathcal{M}}(\boldsymbol{x}_{\natural}, \boldsymbol{q})$ satisfying (A.2). In particular, for every $\boldsymbol{q} \in \mathcal{M}$, there exists $\boldsymbol{v} \in T_{\boldsymbol{x}_{\natural}}\mathcal{M}$ of norm at most $\|\boldsymbol{v}\| \leq \text{diam}(\mathcal{M})$ satisfying (A.2).

The *logarithmic map*

$$
\widetilde{\log}_{\boldsymbol{x}_{\natural}} : \mathcal{M} \to T_{\boldsymbol{x}_{\natural}}\mathcal{M}
\tag{A.3}
$$

is defined, in the broadest generality, as the inverse of the exponential map. This mapping can be multi-valued, since for a given $\boldsymbol{q}$ there may be multiple tangent vectors $\boldsymbol{v}$ satisfying (A.2). Notice that because $\exp$ is surjective, its inverse, $\widetilde{\log}$ is well defined for all $\boldsymbol{q} \in \mathcal{M}$. When $d_{\mathcal{M}}(\boldsymbol{x}_{\natural}, \boldsymbol{q}) \leq r_{\text{inj}}$ is smaller than the injectivity radius of the exponential map at $\boldsymbol{x}_{\natural}$ [7], there is a unique minimum norm element $\boldsymbol{v}_{\star}$ of the set $\widetilde{\log}_{\boldsymbol{x}_{\natural}}(\boldsymbol{q})$. This is typically denoted

$$
\log_{\boldsymbol{x}_{\natural}}(\boldsymbol{q})
\tag{A.4}
$$

and satisfies $\|\boldsymbol{v}_{\star}\| = d_{\mathcal{M}}(\boldsymbol{x}_{\natural}, \boldsymbol{q})$.[8] We can extend this notation from $\boldsymbol{q} \in B_{\mathcal{M}}(\boldsymbol{x}_{\natural}, r_{\text{inj}})$ to all of $\mathcal{M}$, by letting

$$
\log_{\boldsymbol{x}_{\natural}}(\boldsymbol{q})
\tag{A.5}
$$

denote a minimum norm element of the set $\widetilde{\log}_{\boldsymbol{x}_{\natural}}(\boldsymbol{q})$, chosen arbitrarily in the case that there are multiple minimizers.[9] With this choice, $\log_{\boldsymbol{x}_{\natural}}(\boldsymbol{q})$ is well-defined, single-valued over all of $\mathcal{M}$, and defines a mapping

$$
\log_{\boldsymbol{x}_{\natural}} : \mathcal{M} \to B_{T_{\boldsymbol{x}_{\natural}}\mathcal{M}}\left(\boldsymbol{0}, \text{diam}(\mathcal{M})\right)
\tag{A.6}
$$

Our analysis will assume that the landmarks $Q$ are $\delta$-separated on $\mathcal{M}$, i.e., $d_{\mathcal{M}}(\boldsymbol{q}_i, \boldsymbol{q}_j) \geq \delta$ for all $i \neq j$. We will show under this assumption that $\log_{\boldsymbol{x}_{\natural}}(\boldsymbol{q}_i)$ and $\log_{\boldsymbol{x}_{\natural}}(\boldsymbol{q}_j)$ are $\delta'$-separated, albeit with a radius of separation $\delta'$ which could be significantly smaller than $\delta$.

This argument makes heavy use of properties of geodesic triangles – in particular, Toponogov's theorem, which compares side lengths of geodesic triangles in spaces of *bounded* sectional curvature to side lengths of triangles in spaces of *constant* sectional curvature. Our argument uses the following properties of the mapping $\log$ defined above:

---

[7]$\text{inj}(\boldsymbol{x}_{\natural}) = \sup\{r > 0 : \exp_{\boldsymbol{x}_{\natural}} \text{ is a diffeomorphism on} B(0, r) \subset T_{\boldsymbol{x}_{\natural}}\mathcal{M}\}$

[8]This is often taken as the *definition* of the logarithmic map.

[9]This selection is possible thanks to the axiom of choice.

- *Inverse Property*: for $\boldsymbol{v} = \log_{\boldsymbol{x}_\natural}(\boldsymbol{q})$, $\exp_{\boldsymbol{x}_\natural}(\boldsymbol{v}) = \boldsymbol{q}$

- *Minimum Norm Property*: $\|\log_{\boldsymbol{x}_\natural}(\boldsymbol{q})\| = d_{\mathcal{M}}(\boldsymbol{x}_\natural, \boldsymbol{q}) \leq \operatorname{diam}(\mathcal{M})$.

Our analysis *does not* require analytical properties of the logarithmic map, such as continuity, which do not obtain beyond the injectivity radius of the exponential map.

**Lemma A.1.** *For any $R > 0$, and any $\delta$-separated pair of points $\boldsymbol{q}, \boldsymbol{q}' \in B_{\mathcal{M}}(\boldsymbol{x}_\natural, R)$ (i.e., pair of points satisfying $d_{\mathcal{M}}(\boldsymbol{q}, \boldsymbol{q}') \geq \delta$), we have*

$$\|\log_{\boldsymbol{x}_\natural} \boldsymbol{q} - \log_{\boldsymbol{x}_\natural} \boldsymbol{q}'\| \geq \frac{\sqrt{2}}{4} \exp(-\kappa R)\delta. \tag{A.7}$$

**Proof.** We will prove this claim by applying Toponogov's theorem, a fundamental result in Riemannian geometry. Toponogov's theorem is a comparison theorem for triangles, which allows us to compare side lengths of geodesic triangles in an arbitrary manifold of bounded sectional curvature to the side lengths of geodesic triangles in a model space of *constant* sectional curvature. From Lemma 10 in (Yan* et al., 2023), the sectional curvatures of $\mathcal{M}$ are bounded from below by the extrinsic curvature $\kappa$, i.e.,

$$\kappa_s \geq -\kappa^2. \tag{A.8}$$

Our plan is as follows: form a geodesic triangle $\triangle(\boldsymbol{x}, \boldsymbol{q}_{-\kappa^2}, \boldsymbol{q}'_{-\kappa^2})$ in the model $M_{-\kappa^2}$ with constant section curvature $-\kappa^2$, whose side lengths satisfy

$$d_{\mathcal{M}_{-\kappa^2}}(\boldsymbol{x}, \boldsymbol{q}_{-\kappa^2}) = d_{\mathcal{M}}(\boldsymbol{x}_\natural, \boldsymbol{q}), \qquad d_{\mathcal{M}_{-\kappa^2}}(\boldsymbol{x}, \boldsymbol{q}'_{-\kappa^2}) = d_{\mathcal{M}}(\boldsymbol{x}_\natural, \boldsymbol{q}'), \tag{A.9}$$

and whose angle satisfies

$$\angle(\boldsymbol{q}_{-\kappa^2}, \boldsymbol{x}, \boldsymbol{q}'_{-\kappa^2}) = \angle(\boldsymbol{q}, \boldsymbol{x}_\natural, \boldsymbol{q}') \tag{A.10}$$

Then by Toponogov's theorem, the third sides of these pair of triangles satisfy the inequality

$$d_{\mathcal{M}}(\boldsymbol{q}, \boldsymbol{q}') \leq d_{\mathcal{M}_{-\kappa^2}}(\boldsymbol{q}_{-\kappa^2}, \boldsymbol{q}'_{-\kappa^2}), \tag{A.11}$$

i.e., the third side in the constant curvature model space is larger than that in $\mathcal{M}$.

We construct the triangle $\triangle(\boldsymbol{x}, \boldsymbol{q}_{-\kappa^2}, \boldsymbol{q}'_{-\kappa^2})$ more explicitly as follows: fix a arbitrary base point $\boldsymbol{x} \in M_{-\kappa^2}$. Let $\boldsymbol{v}, \boldsymbol{v}'$ be two distinct tangent vectors in the tangent space $T_{\boldsymbol{x}}\mathcal{M}_{-\kappa^2}$ satisfying

$$\|\boldsymbol{v}\|_2 = \|\log_{\boldsymbol{x}_\natural} \boldsymbol{q}\|_2, \quad \|\boldsymbol{v}'\|_2 = \|\log_{\boldsymbol{x}_\natural} \boldsymbol{q}'\|_2, \tag{A.12}$$

and $\theta = \angle(\log_{\boldsymbol{x}_\natural} \boldsymbol{q}, \log_{\boldsymbol{x}_\natural} \boldsymbol{q}') = \angle(\boldsymbol{v}, \boldsymbol{v}')$. Set $\boldsymbol{q}_{-\kappa^2} = \exp_{\boldsymbol{x}}(\boldsymbol{v}) \in \mathcal{M}_{-\kappa^2}$, $\boldsymbol{q}'_{-\kappa^2} = \exp_{\boldsymbol{x}}(\boldsymbol{v}') \in \mathcal{M}_{-\kappa^2}$.

Notice that $\|\boldsymbol{v} - \boldsymbol{v}'\| = \|\log_{\boldsymbol{x}_\natural} \boldsymbol{q} - \log_{\boldsymbol{x}_\natural} \boldsymbol{q}'\|$. We would like to *lower bound* this quantity. From (A.11) and the fact that $d_{\mathcal{M}}(\boldsymbol{q}, \boldsymbol{q}') \geq \delta$, we have $d_{\mathcal{M}_{-\kappa^2}}(\boldsymbol{q}_{-\kappa^2}, \boldsymbol{q}'_{-\kappa^2}) \geq \delta$, and the task becomes one of lower bounding $\|\boldsymbol{v} - \boldsymbol{v}'\|$ in terms of this quantity. To facilitate this bound, we move $\mathcal{M}_{-\kappa^2}$ to hyperbolic space $\mathcal{M}_{-1}$, where we can apply standard results from hyperbolic trigonometry, by scaling all side lengths by $\kappa$. Namely, form a third geodesic triangle in $\mathcal{M}_{-1}$, by taking an arbitrary $\boldsymbol{x}_{-1} \in \mathcal{M}_{-1}$, choosing $\boldsymbol{v}_{-1}, \boldsymbol{v}'_{-1} \in T_{\boldsymbol{x}_{-1}}\mathcal{M}_{-1}$ with $\angle(\boldsymbol{v}_{-1}, \boldsymbol{v}'_{-1}) = \theta$ and $\|\boldsymbol{v}_{-1}\|_2 = \kappa\|\boldsymbol{v}\|$, $\|\boldsymbol{v}'_1\|_2 = \kappa\|\boldsymbol{v}'\|$. As above, set $\boldsymbol{q}_{-1} = \exp_{\boldsymbol{x}_{-1}}(\boldsymbol{v}_{-1})$, and $\boldsymbol{q}'_{-1} = \exp_{\boldsymbol{x}_{-1}}(\boldsymbol{v}'_{-1})$. Then

$$d_{\mathcal{M}_{-1}}(\boldsymbol{q}_{-1}, \boldsymbol{q}'_{-1}) = \kappa d_{\mathcal{M}_{-\kappa^2}}(\boldsymbol{q}_{-\kappa^2}, \boldsymbol{q}'_{-\kappa^2}). \tag{A.13}$$

Moreover,

$$\|\boldsymbol{v}_{-1} - \boldsymbol{v}'_{-1}\| = \kappa\|\boldsymbol{v} - \boldsymbol{v}'\|. \tag{A.14}$$

For compactness of notation, let $L'$ denote the third sidelength of $\triangle_{-1}$,

$$L' = d_{\mathcal{M}_{-1}}(\boldsymbol{q}_{-1}, \boldsymbol{q}'_{-1}) = \kappa * d_{\mathcal{M}_{-\kappa^2}}(\boldsymbol{q}_{-\kappa^2}, \boldsymbol{q}'_{-\kappa^2}). \tag{A.15}$$

The lengths of the other two side are $a = \kappa * \|\boldsymbol{v}\|_2 \leq \kappa R$, $b = \kappa * \|\boldsymbol{v}'\|_2 \leq \kappa R$, and angle between these two sides is equal to $\theta$. In the corresponding Euclidean triangle on the tangent space, we also have the two sides are of length $a$ and $b$, and the third side has length

$$L = \kappa * \|\boldsymbol{v} - \boldsymbol{v}'\|_2. \tag{A.16}$$

As $\mathcal{M}_{-1}$ is hyperbolic, from hyperbolic law of cosines, we have

$$\cosh L' = \cosh a \cosh b - \sinh a \sinh b \cos \theta. \tag{A.17}$$

From the fact that $\cosh(a-b) = \cosh a \cosh b - \sinh a \sinh b$, we could further get

$$\cosh L' = \cosh(a-b) + \sinh a \sinh b (1 - \cos \theta). \tag{A.18}$$

Since $\sinh t$ is convex over $t \in [0, \infty)$, we have $\sinh t \leq \frac{t}{\kappa R} \sinh(\kappa R)$ for $t \leq \kappa R$, hence $\sinh a \sinh b \leq \frac{ab}{(\kappa R)^2} \sinh^2(\kappa R)$. And $\cosh(a-b) = \cosh|a-b| \leq \cosh L$. Then we have

$$\cosh L' \leq \cosh L + \frac{ab}{(\kappa R)^2} \sinh^2(\kappa R)(1 - \cos \theta). \tag{A.19}$$

From the law of cosines applying on Euclidean triangle with length of two sides $a, b$ and the length of the third side $L$, we know

$$\begin{aligned}
L^2 &= a^2 + b^2 - 2ab \cos \theta \\
&\geq 2ab(1 - \cos \theta) \\
&\geq 2ab \frac{(\cosh L' - \cosh L)(\kappa R)^2}{ab \sinh^2(\kappa R)} \\
&= 2(\kappa R)^2 \frac{(\cosh L' - \cosh L)}{\sinh^2(\kappa R)}.
\end{aligned} \tag{A.20}$$

Since $d_{\mathcal{M}}(\boldsymbol{q}, \boldsymbol{q}') \geq \delta$, then $L' = \kappa * d_{\mathcal{M}_{-\kappa^2}}(\boldsymbol{q}_{-\kappa^2}, \boldsymbol{q}'_{-\kappa^2}) \geq \kappa * d_{\mathcal{M}}(\boldsymbol{q}, \boldsymbol{q}') \geq \kappa \delta$. Then equation (A.20) implies

$$L^2 + \frac{2(\kappa R)^2}{\sinh^2(\kappa R)} \cosh L \geq \frac{2(\kappa R)^2}{\sinh^2(\kappa R)} \cosh \kappa \delta. \tag{A.21}$$

By triangle inequality, we know $L \leq a + b \leq 2\kappa R$. From the mean value form of the Taylor series, we have $\sinh(\kappa R) = \kappa R \cosh(r_1)$ for some $0 \leq r_1 \leq \kappa R$ and

$$\cosh L \leq 1 + \frac{L^2}{2} \cosh(r_2), \tag{A.22}$$

for some $0 \leq r_2 \leq L$. Multiply equation (A.21) both side by $\frac{\sinh^2(\kappa R)}{2(\kappa R)^2}$ and plug in the value above, we get

$$\cosh \kappa \delta \leq \frac{\sinh^2(\kappa R)}{2(\kappa R)^2} L^2 + \cosh L \tag{A.23}$$

$$\leq \frac{\cosh^2(r_1)}{2} L^2 + \left(1 + \frac{L^2}{2} \cosh(r_2)\right). \tag{A.24}$$

Rearrange the terms, we get

$$L^2 \geq 2 \left(\cosh^2(r_1) + \cosh(r_2)\right)^{-1} (\cosh(\kappa \delta) - 1) \tag{A.25}$$

$$\geq 2 \left(\cosh^2(\kappa R) + \cosh(2\kappa R)\right)^{-1} \frac{\kappa^2 \delta^2}{2} \tag{A.26}$$

$$\geq \frac{1}{2} \exp(-2\kappa R) \kappa^2 \delta^2. \tag{A.27}$$

$$\tag{A.28}$$

where from the first to second line we used that $\cosh(t) \geq 1 + \frac{t^2}{2}$. As a result, we have $\|\log_{\boldsymbol{x}_\natural} \boldsymbol{q} - \log_{\boldsymbol{x}_\natural} \boldsymbol{q}'\| = \|\boldsymbol{v} - \boldsymbol{v}'\| = L/\kappa \geq \frac{\sqrt{2}}{2} \exp(-\kappa R)\delta \geq \frac{\sqrt{2}}{4} \exp(-\kappa R)\delta$. ∎

**Lemma A.2.** *Consider $\boldsymbol{x}_\natural \in \mathcal{M}$ and let $\mathcal{Q}_R = \{\boldsymbol{q} : \boldsymbol{q} \in B_\mathcal{M}(\boldsymbol{x}_\natural, R) \cap Q\}$. Then the number of landmarks within $R$ ball centering at $\boldsymbol{x}_\natural$, i.e. $|\mathcal{Q}_R|$, satisfies*

$$|\mathcal{Q}_R| \leq \left(1 + 4\sqrt{2}Re^{\kappa R}/\delta\right)^d, \forall R > 0. \tag{A.29}$$

*In particular, we have*

$$|\mathcal{Q}| \leq \left(1 + 4\sqrt{2}\delta^{-1} \operatorname{diam}(\mathcal{M})e^{\kappa \operatorname{diam}(\mathcal{M})}\right)^d. \tag{A.30}$$

**Proof.** For every $\boldsymbol{q} \in \mathcal{Q}_R$, there is a *unique* tangent vector $\log_{\boldsymbol{x}_\natural} \boldsymbol{q} \in T_{\boldsymbol{x}_\natural}\mathcal{M}$. Now we define the set $\mathcal{V}_R = \{\log_{\boldsymbol{x}_\natural} \boldsymbol{q} \in T_{\boldsymbol{x}_\natural}\mathcal{M} \quad \forall \boldsymbol{q} \in \mathcal{Q}_R\}$. Then the number of landmarks within the intrinsic ball $\boldsymbol{B}_\mathcal{M}(\boldsymbol{x}_\natural, R)$ is $|\mathcal{Q}_R| = |\mathcal{V}_R|$.

Let $\delta_R = \frac{\sqrt{2}}{4}\exp(-\kappa R)\delta$. From Lemma A.1, we know that $\mathcal{V}_R$ forms a $\delta_R$-separated subset of $B(0, R)$ on $T_{\boldsymbol{x}_\natural}\mathcal{M}$. And for any $\log_{\boldsymbol{x}_\natural} \boldsymbol{q} \in \mathcal{V}_R$, we have $\left\|\log_{\boldsymbol{x}_\natural} \boldsymbol{q}\right\|_2 = d_\mathcal{M}(\boldsymbol{x}_\natural, \boldsymbol{q}) \leq R$. Then it's natural to notice that $|\mathcal{V}_R| \leq P(B(0, R), \delta_R)$, where $P(B(0, R), \delta_R)$ is the largest cardinality of a $\delta_R$-separated subset of $B(0, R) \in T_{\boldsymbol{x}_\natural}\mathcal{M}$.

Since $P(B(0, R), \delta_R)$ is the largest number of closed disjoint balls with centers in $B(0, R)$ and radii $\delta_R/2$, then by volume comparison, we have

$$P(B(0, R), \delta_R) * \operatorname{vol}(B_{\delta_R/2}) \leq \operatorname{vol}(B_{\delta_R/2+R}). \tag{A.31}$$

Then we will have $|\mathcal{V}_R| \leq P(B(0, R), \delta_R) \leq \left(1 + \frac{2R}{\delta_R}\right)^d$ which gives the bound we need.

To bound $\mathcal{Q}$, we can simply take $R = \operatorname{diam}(\mathcal{M})$ and notice $|\mathcal{Q}| = \left|\mathcal{Q}_{\operatorname{diam}(\mathcal{M})}\right|$. $\blacksquare$

**Lemma A.3.** *Let $\mathcal{M} \in \mathbb{R}^D$ be a complete d-dimensional manifold. Suppose the set of landmarks $Q = \{\boldsymbol{q}\} \subset \mathcal{M}$ forms a $\delta$-net for $\mathcal{M}$, and first -order edges $E^1$ satisfies that $u \xrightarrow{1} v \in E^1$ if $\|\boldsymbol{q}_u - \boldsymbol{q}_v\|_2 \leq R_{nbrs}$, where $R_{nbrs} \leq \tau_\mathcal{M}$, and $\tau_\mathcal{M}$ is the reach of the manifold. Assume $\delta \leq \operatorname{diam} \mathcal{M}$, and $\kappa \operatorname{diam}(\mathcal{M}) \geq 1$. Then the number of first-order edges $\left|E^1\right|$ satisfies*

$$\log\left|E^1\right| \leq (\kappa \operatorname{diam}(\mathcal{M}) + \log(a) - \log(\delta) + \log(\operatorname{diam}(\mathcal{M})) + \log(100))\, d. \tag{A.32}$$

**Proof.** From construction, we have

$$\left|E^1\right| \leq |Q| \max_{\boldsymbol{q}_u \in \mathcal{Q}} \left|E_u^1\right|, \tag{A.33}$$

where $E_u^1$ denotes the first-order edges at landmark $\boldsymbol{q}_u$. As we have $R_{\mathrm{nbrs}} \leq \tau_\mathcal{M} \leq 1/\kappa$, following Lemma A.2 and Equation (A.30), we get

$$\left|E^1\right| \leq \left(1 + 4\sqrt{2}\delta^{-1} R_{\mathrm{nbrs}}e^{\kappa R_{\mathrm{nbrs}}}\right)^d \left(1 + 4\sqrt{2}\delta^{-1} \operatorname{diam}(\mathcal{M})e^{\kappa \operatorname{diam}(\mathcal{M})}\right)^d \tag{A.34}$$

$$= \left(1 + 4\sqrt{2}ae^{\kappa R_{\mathrm{nbrs}}}\right)^d \left(1 + 4\sqrt{2}\delta^{-1} \operatorname{diam}(\mathcal{M})e^{\kappa \operatorname{diam}(\mathcal{M})}\right)^d. \tag{A.35}$$

We recall that $a = R_{\mathrm{nbrs}}/\delta \geq 40$. Since $e^{\kappa R_{\mathrm{nbrs}}} \geq 1$, we have $\frac{1}{40}ae^{\kappa R_{\mathrm{nbrs}}} \geq 1$, which means $1 + 4\sqrt{2}ae^{\kappa R_{\mathrm{nbrs}}} \leq (\frac{1}{40} + 4\sqrt{2})ae^{\kappa R_{\mathrm{nbrs}}} \leq (\frac{1}{40} + 4\sqrt{2})a \cdot e$, since $\kappa R_{\mathrm{nbrs}} \leq 1$. Similarly, since $\delta \leq \operatorname{diam} \mathcal{M}$, and $\kappa \operatorname{diam}(\mathcal{M}) \geq 1$, $1 + 4\sqrt{2}\delta^{-1} \operatorname{diam}(\mathcal{M})e^{\kappa \operatorname{diam}(\mathcal{M})} \leq (\frac{1}{e} + 4\sqrt{2})\delta^{-1} \operatorname{diam}(\mathcal{M})e^{\kappa \operatorname{diam}(\mathcal{M})}$, which gives

$$\left|E^1\right| \leq \left((\frac{e}{40} + 4\sqrt{2}e)a\right)^d \left((\frac{1}{e} + 4\sqrt{2})\delta^{-1} \operatorname{diam}(\mathcal{M})e^{\kappa \operatorname{diam}(\mathcal{M})}\right)^d \tag{A.36}$$

$$\leq \left((\frac{e}{40} + 4\sqrt{2}e)(\frac{1}{e} + 4\sqrt{2})\right)^d \left(a\delta^{-1} \operatorname{diam}(\mathcal{M})e^{\kappa \operatorname{diam}(\mathcal{M})}\right)^d \tag{A.37}$$

$$\leq (100)^d \left(a\delta^{-1} \operatorname{diam}(\mathcal{M})e^{\kappa \operatorname{diam}(\mathcal{M})}\right)^d. \tag{A.38}$$

Taking the log, we get

$$\log \left| E^1 \right| \leq \left( \kappa \operatorname{diam}(\mathcal{M}) + \log(a) - \log(\delta) + \log(\operatorname{diam}(\mathcal{M})) + \log(100) \right) d. \tag{A.39}$$

■

**Lemma A.4.** *There exists a numerical constant $C$ such that with probabilty at least $1 - e^{-9d/2}$,*

$$T_{\max} = \sup \left\{ \langle v, z \rangle \mid \begin{array}{l} d_{\mathcal{M}}(y, x_\natural) \leq 1/\kappa, \\ v \in T_y \mathcal{M}, \|v\|_2 = 1 \end{array} \right\}. \tag{A.40}$$

*satisfies*

$$T_{\max} \leq C \max\{\kappa, 1\} \times \sigma \sqrt{d}. \tag{A.41}$$

**Proof.** Let $\bar{\kappa} = \max\{\kappa, 1\}$. From Theorem 10 of (Yan* et al., 2023), we have that with probability at least $1 - e^{-\frac{x^2}{2\sigma^2}}$,

$$T_{\max} \leq 12\sigma \left( \bar{\kappa} \sqrt{2\pi(d+1)} + \sqrt{\log 12\bar{\kappa}} \right) + x. \tag{A.42}$$

We note that there exists numerical constants $C_1, C_2$, such that $\sqrt{\log 12\bar{\kappa}} \leq C_1 \bar{\kappa}$, and $\sqrt{d+1} \leq C_2 \sqrt{d}$. Setting $x = 3\sigma\sqrt{d}$, then we have

$$\begin{aligned} 12\sigma(\bar{\kappa}\sqrt{2\pi(d+1)} + \sqrt{\log 12\bar{\kappa}}) + x &\leq 12\sigma\bar{\kappa}\sqrt{2\pi}C_2\sqrt{d} + \sigma C_1 \bar{\kappa} + 3\sigma\sqrt{d} \\ &= \left( 12\bar{\kappa}\sqrt{2\pi}C_2 + C_1\bar{\kappa} + 3 \right)\sigma\sqrt{d} \end{aligned}, \tag{A.43}$$

yielding the claimed bound.

■

**Lemma A.5.** *Let*

$$T_{\max} = \sup \left\{ \langle v, z \rangle \mid \begin{array}{l} d_{\mathcal{M}}(y, x_\natural) \leq 1/\kappa, \\ v \in T_y \mathcal{M}, \|v\|_2 = 1 \end{array} \right\}. \tag{A.44}$$

*Then with probability at least $1 - e^{-\frac{t^2}{2\sigma^2}}$, we have*

$$T_{\max} \geq \sigma\sqrt{d/2} - t. \tag{A.45}$$

**Proof.** Since $T_{\max}$ is the supremum of a 1-Lipschitz function and is therefore 1-Lipschitz in $z$, it follows that

$$\mathbb{P}\left[ T_{\max} \leq \mathbb{E}[T_{\max}] - t \right] \leq e^{-t^2/2\sigma^2}. \tag{A.46}$$

By setting $y = x_\natural$ and $v = \mathcal{P}_{T_{x_\natural}\mathcal{M}} z$ we obtain $T_{\max} \geq \left\| \mathcal{P}_{T_{x_\natural}\mathcal{M}} z \right\|_2$ and thus $\mathbb{E}[T_{\max}] \geq \mathbb{E}\left[ \left\| \mathcal{P}_{T_{x_\natural}\mathcal{M}} z \right\|_2 \right]$.

Since $z$ is i.i.d. Gaussian with variance $\sigma^2$, the rotational invariance of Gaussian distributions implies that $\mathbb{E}[\|\mathcal{P}_{T_{x_\natural}\mathcal{M}} z\|] = \mathbb{E}[\|\sigma g_d\|]$, where $g_d \sim \mathcal{N}(0, I_d)$ is a $d$ dimensional standard Gaussian vector. As $\|g_d\|$ is 1-Lipschitz, from Lemma A.6 we have

$$1 \geq \operatorname{Var}[\|g_d\|] = \mathbb{E}[\|g_d\|^2] - \mathbb{E}[\|g_d\|]^2 = d - \mathbb{E}[\|g_d\|]^2, \tag{A.47}$$

and thus $\mathbb{E}[\|g_d\|] \geq \sqrt{d-1}$. Therefore,

$$\mathbb{E}[\|\mathcal{P}_{T_{x_\natural}\mathcal{M}} z\|] = \mathbb{E}[\sigma\|g_d\|] \geq \sigma\sqrt{d-1} \geq \sigma\sqrt{d/2}, \quad \text{for } d \geq 2. \tag{A.48}$$

For the case where $d = 1$, we compute directly:

$$\mathbb{E}[\|\mathcal{P}_{T_{x_\natural}\mathcal{M}}\|] = \sqrt{2}\sigma \frac{\Gamma((d+1)/2)}{\Gamma(d/2)} = \sqrt{2/\pi}\sigma \geq \sqrt{d/2}\sigma. \tag{A.49}$$

Combining the cases for $d \geq 2$ and $d = 1$, and substituting into Equation (A.46), we conclude that

$$T_{\max} \geq \sigma\sqrt{d/2} - t \quad \text{with probability at least } 1 - e^{-t^2/2\sigma^2}. \tag{A.50}$$

■

**Lemma A.6** (Bounded Variance of 1-Lipschitz Function). *Let $g \sim \mathcal{N}(0, I)$ be a standard Gaussian, $f$ is a 1-Lipschitz function, then we have*

$$\text{Var}[f(g)] \leq 1. \tag{A.51}$$

**Proof.** In the prove, we utilize the Gaussian Poincaré inequality (Boucheron et al., 2003)[Theorem 3.20] which says that

$$\text{Var}[h(g)] \leq \mathbb{E}[\|\nabla h(g)\|^2] \tag{A.52}$$

for any $C^1$ function $h$. Let $\rho_\varepsilon$ be the standard Gaussian mollifier $\rho_\varepsilon(z) = \frac{1}{(2\pi\varepsilon)^d} e^{-\frac{\|z\|^2}{2\varepsilon}}$ and let

$$f_\varepsilon = f * \rho_\varepsilon = \int_z f(x - x) \rho_\varepsilon(z) dz,$$

then $f_\varepsilon$ is smooth. As

$$|f_\varepsilon(x) - f_\varepsilon(y)| = \left| \int_z \left( f_\varepsilon(x - z) - f_\varepsilon(y - z) \right) \rho_\varepsilon(z) dz \right| \tag{A.53}$$

$$\leq \int_z |(f_\varepsilon(x - z) - f_\varepsilon(y - z))| \rho_\varepsilon(z) dz \tag{A.54}$$

$$\leq \|x - y\| \int_z \rho_\varepsilon(z) dz = \|x - y\|, \tag{A.55}$$

$f_\varepsilon$ is also 1-Lipschitz. Following the Gaussian Poincaré inequality we have

$$\text{Var}[f_\varepsilon(g)] \leq \mathbb{E}[\|\nabla f_\varepsilon(g)\|^2] \leq 1. \tag{A.56}$$

To conclude the result, we need to show the interchangeability of the interation and the limit. As $f_\varepsilon$ is 1-Lipschitz, we have

$$|f_\varepsilon(x)|^2 \leq \|f_\varepsilon(0)\| + \|x\| \tag{A.57}$$

$$\leq \left| \int_z f(-z) \rho_\varepsilon(z) dz \right| + \|x\| \tag{A.58}$$

$$\leq \int_z \|z\| \rho_\varepsilon(z) dz + \|x\| \tag{A.59}$$

$$= \sqrt{\varepsilon} \mathbb{E}[\|g\|] + \|x\|. \tag{A.60}$$

As the moments of a standard Gaussian are upper bounded, $|f_\varepsilon(g) - \mathbb{E}[f_\varepsilon(g)]|^2$ can be uniformly upper bounded by some integrable function for all $\varepsilon \leq 1$. And thus

$$\text{Var}[f[g]] = \int_{\varepsilon \to 0} \text{Var}[f_\varepsilon(g)] \leq 1. \tag{A.61}$$

∎

# B. Experimental Details

## B.1. Gravitational Waves Data Generation

We generate synthetic gravitational waveforms with the PyCBC package (Nitz et al., 2023) with masses drawn from a Gaussian distribution with mean 35 and variance 15. We use rejection sampling to limit masses to the range [20, 50]. Each waveform is sampled at 2048Hz, padded or truncated to 1 second, and normalized to have unit $\ell^2$ norm. We simulate noise as i.i.d. Gaussian with standard deviation $\sigma = 0.01$ (Yan* et al., 2023). The training set consists of $100,000$ noisy waveforms, the test set contains $20,000$ noisy waveforms.

## B.2. Noisy vs. Denoised Gravitational Wave Signal Visualization

The following example (Figure 10) presents a visualization of a noisy gravitational wave signal and its corresponding denoised version, obtained using Denoiser 1 described in Table 1.

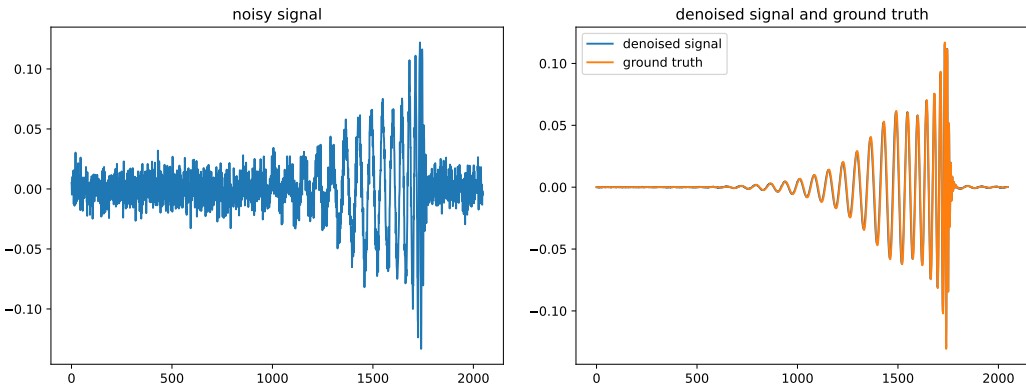

Figure 10: **Left:** Noisy gravitational wave signal. **Right:** Corresponding denoised output obtained using Denoiser 1 described in 1 and the ground truth signal.

## C. Traversal Networks on Synthetic Manifolds

In this section, we present traversal networks created by Algorithm 2 on the following synthetic manifolds: sphere (Figure 12), torus (Figure 11).

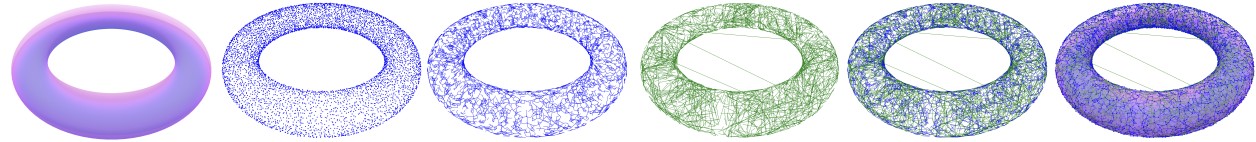

Figure 11: Traversal network on the torus with noise level $\sigma = 0.01$. Left-to-Right, Top-to-Bottom: Clean manifold, landmarks (blue dots), first-order edges (blue lines), zero-order edges (green lines), final traversal network, and final traversal network overlayed with clean manifold.

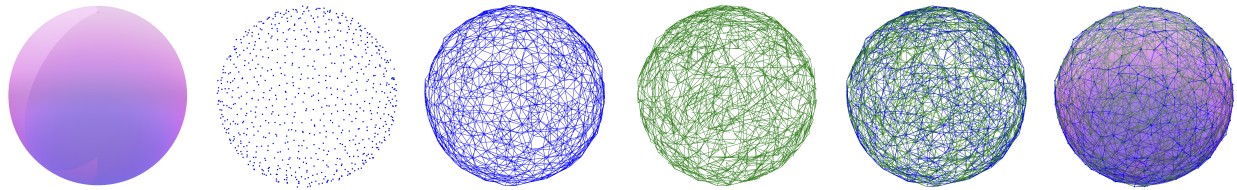

Figure 12: Traversal network on the sphere created based on 100,000 noisy points with noise level $\sigma = 0.01$. Left-to-Right, Top-to-Bottom: Clean manifold, landmarks (blue dots), first-order edges (blue lines), zero-order edges (green lines), final traversal network, and final traversal network overlayed with clean manifold.

## D. Incremental PCA for Efficient Tangent Space Approximation

In this section, we detail the tangent space approximation implementation mentioned in Algorithm 2 and detailed in Algorithm 4 and Algorithm 5. We use Incremental Principal Component Analysis (PCA) to efficiently process streaming high-dimensional data. Below we present the mathematics and algorithmic details of our implementation.

**Initializing Local Model Parameters at New Landmarks:** If a newly created landmark $\boldsymbol{q}_M$ in Algorithm 2 has no other landmarks within $R_{\text{nbrs}}$ distance, then its tangent space $T_{\boldsymbol{q}_M}$ is initialized randomly. Otherwise, we identify all existing landmarks within radius $R_{\text{nbrs}}$ – referred to as the first-order neighbors – and establish connections to them. The local parameters $T_{\boldsymbol{q}_M}$ and $\Xi_{\boldsymbol{q}_M}$ are then initialized based on the information from these first-order neighbors, as detailed below.

Let $\{\boldsymbol{q}_i\}_{i=1}^k$ denote the set of first-order neighbors of landmark $\boldsymbol{q}_M$, excluding $\boldsymbol{q}_M$ itself. We compute the normalized

---

**Algorithm 4** `IncrPCAonMatrix`$(\boldsymbol{X}, d)$

---

1: **Input:** centered data matrix $\boldsymbol{X} = [\boldsymbol{x}_1 - \boldsymbol{q}_M, \ldots, \boldsymbol{x}_n - \boldsymbol{q}_M] \in \mathbb{R}^{D \times n}$ containing points within radius $R(M)$ of $\boldsymbol{q}_M$, intrinsic dimension $d$.
2: $\boldsymbol{U}_1 \leftarrow [\boldsymbol{x}_1 - \boldsymbol{q}_M / \|\boldsymbol{x}_1 - \boldsymbol{q}_M\|]$
3: $\boldsymbol{\Lambda}_1 \leftarrow [\|\boldsymbol{x}_1 - \boldsymbol{q}_M\|_2^2]$
4: **for** i = 1, \ldots, n - 1 **do**
5: $\quad \boldsymbol{U}_{i+1}, \boldsymbol{\Lambda}_{i+1} \leftarrow$ `IncrPCA`$(\boldsymbol{x}_{i+1} - \boldsymbol{q}_M, \boldsymbol{U}_i, \boldsymbol{\Lambda}_i, i + 1, d)$
6: **end for**
7: **Output:** $\boldsymbol{U}_n, \boldsymbol{\Lambda}_n$

---

difference vectors:

$$\boldsymbol{h}_i = \frac{\boldsymbol{q}_i - \boldsymbol{q}_M}{\|\boldsymbol{q}_i - \boldsymbol{q}_M\|}, \tag{D.1}$$

and assemble them into a matrix $\boldsymbol{H} = \begin{bmatrix} \boldsymbol{h}_1 & \boldsymbol{h}_2 & \ldots & \boldsymbol{h}_k \end{bmatrix}$. The tangent space $T_{\boldsymbol{q}_M}$ is spanned by the orthonormal matrix $\boldsymbol{U}_{\boldsymbol{q}_M}$ obtained through truncated singular value decomposition of $\boldsymbol{H}$, which ensures $\boldsymbol{U}_{\boldsymbol{q}_M} \in \mathbb{R}^{D \times d}$, where $D$ and $d$ represent ambient and intrinsic dimensions, respectively. Edge embeddings $\Xi_{\boldsymbol{q}_M}$ are then created via projecting vectors $\boldsymbol{q}_i - \boldsymbol{q}_M$ onto $T_{\boldsymbol{q}_M}$.

**Updating Tangent Space Approximations Efficiently:** Once a new landmark $\boldsymbol{q}_M$ is created, along with its tangent space $T_{\boldsymbol{q}_M}$ and edge embeddings $\Xi_{\boldsymbol{q}_M}$, we will update local models at $\boldsymbol{q}_M$ using incoming noisy samples that fall within radius $R(M)$. A straightforward way to perform this update is to re-estimate quantities such as the landmark and associated tangent space for each sample as the online learning process progresses. However, repeatedly estimating these quantities from the full dataset is computationally inefficient. To address this, we employ incremental PCA for tangent space updates, following the methods proposed in (Brand, 2006; Arora et al., 2012), as detailed below.

Let vertex $M$ have local parameters $\boldsymbol{q}_M, T_{\boldsymbol{q}_M}, \Xi_{\boldsymbol{q}_M}$. Define centered data matrix $\boldsymbol{X}_n = [\boldsymbol{x}_1 - \boldsymbol{q}_M, \boldsymbol{x}_2 - \boldsymbol{q}_M, \ldots, \boldsymbol{x}_n - \boldsymbol{q}_M] \in \mathbb{R}^{D \times n}$. When a new sample $\boldsymbol{x}_{n+1}$ arrives, it is appended to form the updated matrix $\boldsymbol{X}_{n+1} = [\boldsymbol{X}_n, \boldsymbol{x}_{n+1} - \boldsymbol{q}_M]$. Let $\boldsymbol{U}_n \boldsymbol{\Lambda}_n \boldsymbol{U}_n^T$ denote the rank-$l$ ($l \leq d$, where $d$ is the intrinsic dimension of the manifold.) approximation of the $\boldsymbol{X}_n \boldsymbol{X}_n^T$, where $\boldsymbol{U}_n \in \mathbb{R}^{D \times l}$ is an orthonormal matrix that approximates the basis of the tangent space at $\boldsymbol{q}_M$ prior the arrival of new sample $\boldsymbol{x}_{n+1}$, and $\boldsymbol{\Lambda}_n \in \mathbb{R}^{l \times l}$ is a diagonal matrix of corresponding eigenvalues. Our goal is to *efficiently* update the tangent space at $\boldsymbol{q}_M$ upon receiving the new sample $\boldsymbol{x}_{n+1}$.

We note that

$$\begin{aligned} \boldsymbol{X}_{n+1} \boldsymbol{X}_{n+1}^T &= \begin{bmatrix} \boldsymbol{X}_n & \boldsymbol{x}_{n+1} - \boldsymbol{q}_M \end{bmatrix} \begin{bmatrix} \boldsymbol{X}_n & \boldsymbol{x}_{n+1} - \boldsymbol{q}_M \end{bmatrix}^T \tag{D.2} \\ &= \boldsymbol{X}_n \boldsymbol{X}_n^T + (\boldsymbol{x}_{n+1} - \boldsymbol{q}_M)(\boldsymbol{x}_{n+1} - \boldsymbol{q}_M)^T. \tag{D.3} \end{aligned}$$

Let $\boldsymbol{S} = \boldsymbol{U}_n \boldsymbol{\Lambda}_n \boldsymbol{U}_n^T + (\boldsymbol{x}_{n+1} - \boldsymbol{q}_M)(\boldsymbol{x}_{n+1} - \boldsymbol{q}_M)^T$. Then $\boldsymbol{S}$ is an approximation of $\boldsymbol{X}_{n+1} \boldsymbol{X}_{n+1}^T$. A rank-$l$ approximation of $\boldsymbol{S}$ therefore provides a rank-$l$ approximation of the $\boldsymbol{X}_{n+1} \boldsymbol{X}_{n+1}^T$. The following outlines the procedure for deriving the rank-$l$ approximation of $\boldsymbol{S}$.

The matrix $\boldsymbol{S}$ can be written as follows.

$$\begin{aligned} \boldsymbol{S} &= \boldsymbol{U}_n \boldsymbol{\Lambda}_n \boldsymbol{U}_n^T + (\boldsymbol{x}_{n+1} - \boldsymbol{q}_M)(\boldsymbol{x}_{n+1} - \boldsymbol{q}_M)^T \tag{D.4} \\ &= \begin{bmatrix} \boldsymbol{U}_n & \boldsymbol{x}_{n+1} - \boldsymbol{q}_M \end{bmatrix} \begin{bmatrix} \boldsymbol{\Lambda}_n & \boldsymbol{0} \\ \boldsymbol{0}^T & 1 \end{bmatrix} \begin{bmatrix} \boldsymbol{U}_n & \boldsymbol{x}_{n+1} - \boldsymbol{q}_M \end{bmatrix}^T. \tag{D.5} \end{aligned}$$

We define the component of $\boldsymbol{x}_{n+1} - \boldsymbol{q}_M$ that is orthogonal to the subspace spanned $\boldsymbol{U}_n$ as $(\boldsymbol{x}_{n+1} - \boldsymbol{q}_M)^\perp = (\boldsymbol{I} - \boldsymbol{U}_n \boldsymbol{U}_n^T)(\boldsymbol{x}_{n+1} - \boldsymbol{q}_M)$. Then we have

---

**Algorithm 5** IncrPCA($\boldsymbol{x}_{i+1} - \boldsymbol{q}_M, \boldsymbol{U}_i, \boldsymbol{\Lambda}_i, i+1, d$)

---

1: **Inputs:** $\boldsymbol{x}_{i+1} - \boldsymbol{q}_M, \boldsymbol{U}_i, \boldsymbol{\Lambda}_i, i+1, d$

2: $\boldsymbol{\Lambda}_{\exp} \leftarrow \begin{bmatrix} \boldsymbol{\Lambda}_i & \boldsymbol{0} \\ \boldsymbol{0}^T & 0 \end{bmatrix}$

3: $(\boldsymbol{x}_{i+1} - \boldsymbol{q}_M)^{\perp} = (\boldsymbol{I} - \boldsymbol{U}_i\boldsymbol{U}_i)^T (\boldsymbol{x}_{i+1} - \boldsymbol{q}_M)$

4: $\boldsymbol{K}_{i+1} = \boldsymbol{\Lambda}_{exp} + \begin{bmatrix} \boldsymbol{U}_i^T (\boldsymbol{x}_{i+1} - \boldsymbol{q}_M) \\ \left\| (\boldsymbol{x}_{i+1} - \boldsymbol{q}_M)^{\perp} \right\| \end{bmatrix} \begin{bmatrix} \boldsymbol{U}_i^T (\boldsymbol{x}_{i+1} - \boldsymbol{q}_M) \\ \left\| (\boldsymbol{x}_{i+1} - \boldsymbol{q}_M)^{\perp} \right\| \end{bmatrix}^T$

5: $\boldsymbol{U}_{K_{i+1}}, \boldsymbol{\Lambda}_{K_{i+1}} \leftarrow \text{eigendecomposition}(\boldsymbol{K}_{i+1})$

6: $\boldsymbol{U}_{i+1} \leftarrow \begin{bmatrix} \boldsymbol{U}_i & \frac{(\boldsymbol{x}_{i+1}-\boldsymbol{q}_M)^{\perp}}{\left\|(\boldsymbol{x}_{i+1}-\boldsymbol{q}_M)^{\perp}\right\|} \end{bmatrix} \boldsymbol{U}_{K_{i+1}}$

7: $\boldsymbol{\Lambda}_{i+1} \leftarrow \boldsymbol{\Lambda}_{K_{i+1}}$

8: **if** $i \geq d$ **then**

9:     $\boldsymbol{U}_{i+1} \leftarrow \boldsymbol{U}_{i+1}[:, :d]$

10:    $\boldsymbol{\Lambda}_{i+1} \leftarrow \boldsymbol{\Lambda}_{i+1}[:d, :d]$

11: **end if**

12: **Output:** $\boldsymbol{U}_{i+1}, \boldsymbol{\Lambda}_{i+1}$

---

$$\begin{bmatrix} \boldsymbol{U}_n & \boldsymbol{x}_{n+1} - \boldsymbol{q}_M \end{bmatrix} = \begin{bmatrix} \boldsymbol{U}_n & \boldsymbol{U}_n\boldsymbol{U}_n^T (\boldsymbol{x}_{n+1} - \boldsymbol{q}_M) + (\boldsymbol{I} - \boldsymbol{U}_n\boldsymbol{U}_n^T)(\boldsymbol{x}_{n+1} - \boldsymbol{q}_M) \end{bmatrix} \tag{D.6}$$

$$= \begin{bmatrix} \boldsymbol{U}_n & \boldsymbol{U}_n\boldsymbol{U}_n^T (\boldsymbol{x}_{n+1} - \boldsymbol{q}_M) + (\boldsymbol{x}_{n+1} - \boldsymbol{q}_M)^{\perp} \end{bmatrix} \tag{D.7}$$

$$= \begin{bmatrix} \boldsymbol{U}_n & \frac{(\boldsymbol{x}_{n+1}-\boldsymbol{q}_M)^{\perp}}{\left\|(\boldsymbol{x}_{n+1}-\boldsymbol{q}_M)^{\perp}\right\|} \end{bmatrix} \begin{bmatrix} \boldsymbol{I} & \boldsymbol{U}_n^T (\boldsymbol{x}_{n+1} - \boldsymbol{q}_M) \\ \boldsymbol{0}^T & \left\|(\boldsymbol{x}_{n+1} - \boldsymbol{q}_M)^{\perp}\right\| \end{bmatrix}. \tag{D.8}$$

Thus we have

$$\boldsymbol{S} = \begin{bmatrix} \boldsymbol{U}_n & \frac{(\boldsymbol{x}_{n+1}-\boldsymbol{q}_M)^{\perp}}{\left\|(\boldsymbol{x}_{n+1}-\boldsymbol{q}_M)^{\perp}\right\|} \end{bmatrix} \boldsymbol{K} \begin{bmatrix} \boldsymbol{U}_n & \frac{(\boldsymbol{x}_{n+1}-\boldsymbol{q}_M)^{\perp}}{\left\|(\boldsymbol{x}_{n+1}-\boldsymbol{q}_M)^{\perp}\right\|} \end{bmatrix}^T, \tag{D.9}$$

where

$$\boldsymbol{K} = \begin{bmatrix} \boldsymbol{I} & \boldsymbol{U}_n^T (\boldsymbol{x}_{n+1} - \boldsymbol{q}_M) \\ \boldsymbol{0}^T & \left\|(\boldsymbol{x}_{n+1} - \boldsymbol{q}_M)^{\perp}\right\| \end{bmatrix} \begin{bmatrix} \boldsymbol{\Lambda}_n & \boldsymbol{0} \\ \boldsymbol{0}^T & 1 \end{bmatrix} \begin{bmatrix} \boldsymbol{I} & \boldsymbol{0} \\ (\boldsymbol{x}_{n+1} - \boldsymbol{q}_M)^T\boldsymbol{U}_n & \left\|(\boldsymbol{x}_{n+1} - \boldsymbol{q}_M)^{\perp}\right\| \end{bmatrix}. \tag{D.10}$$

We can further simplify $\boldsymbol{K}$ as follows.

$$\boldsymbol{K} = \begin{bmatrix} \boldsymbol{\Lambda}_n & \boldsymbol{U}_n^T (\boldsymbol{x}_{n+1} - \boldsymbol{q}_M) \\ \boldsymbol{0}^T & \left\|(\boldsymbol{x}_{n+1} - \boldsymbol{q}_M)^{\perp}\right\| \end{bmatrix} \begin{bmatrix} \boldsymbol{I} & \boldsymbol{0} \\ (\boldsymbol{x}_{n+1} - \boldsymbol{q}_M)^T\boldsymbol{U}_n & \left\|(\boldsymbol{x}_{n+1} - \boldsymbol{q}_M)^{\perp}\right\| \end{bmatrix} \tag{D.11}$$

$$= \begin{bmatrix} \boldsymbol{\Lambda}_n + \boldsymbol{U}_n^T (\boldsymbol{x}_{n+1} - \boldsymbol{q}_M)(\boldsymbol{x}_{n+1} - \boldsymbol{q}_M)^T\boldsymbol{U}_n & \boldsymbol{U}_n^T (\boldsymbol{x}_{n+1} - \boldsymbol{q}_M)\left\|(\boldsymbol{x}_{n+1} - \boldsymbol{q}_M)^{\perp}\right\| \\ \left\|(\boldsymbol{x}_{n+1} - \boldsymbol{q}_M)^{\perp}\right\|(\boldsymbol{x}_{n+1} - \boldsymbol{q}_M)^T\boldsymbol{U}_n & \left\|(\boldsymbol{x}_{n+1} - \boldsymbol{q}_M)^{\perp}\right\|^2 \end{bmatrix} \tag{D.12}$$

$$= \begin{bmatrix} \boldsymbol{\Lambda}_n & \boldsymbol{0} \\ \boldsymbol{0}^T & 0 \end{bmatrix} + \begin{bmatrix} \boldsymbol{U}_n^T (\boldsymbol{x}_{n+1} - \boldsymbol{q}_M)(\boldsymbol{x}_{n+1} - \boldsymbol{q}_M)^T\boldsymbol{U}_n & \boldsymbol{U}_n^T (\boldsymbol{x}_{n+1} - \boldsymbol{q}_M)\left\|(\boldsymbol{x}_{n+1} - \boldsymbol{q}_M)^{\perp}\right\| \\ \left\|(\boldsymbol{x}_{n+1} - \boldsymbol{q}_M)^{\perp}\right\|(\boldsymbol{x}_{n+1} - \boldsymbol{q}_M)^T\boldsymbol{U}_n & \left\|(\boldsymbol{x}_{n+1} - \boldsymbol{q}_M)^{\perp}\right\|^2 \end{bmatrix} \tag{D.13}$$

$$= \begin{bmatrix} \boldsymbol{\Lambda}_n & \boldsymbol{0} \\ \boldsymbol{0}^T & 0 \end{bmatrix} + \begin{bmatrix} \boldsymbol{U}_n^T (\boldsymbol{x}_{n+1} - \boldsymbol{q}_M) \\ \left\|(\boldsymbol{x}_{n+1} - \boldsymbol{q}_M)^{\perp}\right\| \end{bmatrix} \begin{bmatrix} (\boldsymbol{x}_{n+1} - \boldsymbol{q}_M)^T\boldsymbol{U}_n & \left\|(\boldsymbol{x}_{n+1} - \boldsymbol{q}_M)^{\perp}\right\| \end{bmatrix} \tag{D.14}$$

$$= \begin{bmatrix} \boldsymbol{\Lambda}_n & \boldsymbol{0} \\ \boldsymbol{0}^T & 0 \end{bmatrix} + \begin{bmatrix} \boldsymbol{U}_n^T (\boldsymbol{x}_{n+1} - \boldsymbol{q}_M) \\ \left\|(\boldsymbol{x}_{n+1} - \boldsymbol{q}_M)^{\perp}\right\| \end{bmatrix} \begin{bmatrix} \boldsymbol{U}_n^T (\boldsymbol{x}_{n+1} - \boldsymbol{q}_M) \\ \left\|(\boldsymbol{x}_{n+1} - \boldsymbol{q}_M)^{\perp}\right\| \end{bmatrix}^T. \tag{D.15}$$

We note that $\boldsymbol{K}$ is of size $l+1$ by $l+1$, the $\boldsymbol{K} = \boldsymbol{U}_K \boldsymbol{\Lambda}_K \boldsymbol{U}_K^T$ will cost $\mathcal{O}(d^3)$. From Equation (D.9) and (D.11), we know that

$$\boldsymbol{S} = \left[ \begin{array}{cc} \boldsymbol{U}_n & \frac{(\boldsymbol{x}_{n+1}-\boldsymbol{q}_M)^\perp}{\|(\boldsymbol{x}_{n+1}-\boldsymbol{q}_M)^\perp\|} \end{array} \right] \boldsymbol{U}_K \boldsymbol{\Lambda}_K \boldsymbol{U}_K^T \left[ \begin{array}{cc} \boldsymbol{U}_n & \frac{(\boldsymbol{x}_{n+1}-\boldsymbol{q}_M)^\perp}{\|(\boldsymbol{x}_{n+1}-\boldsymbol{q}_M)^\perp\|} \end{array} \right]^T. \tag{D.16}$$

We define

$$\boldsymbol{U}_{n+1} = \left[ \begin{array}{cc} \boldsymbol{U}_n & \frac{(\boldsymbol{x}_{n+1}-\boldsymbol{q}_M)^\perp}{\|(\boldsymbol{x}_{n+1}-\boldsymbol{q}_M)^\perp\|} \end{array} \right] \boldsymbol{U}_K, \tag{D.17}$$

$$\boldsymbol{\Lambda}_{n+1} = \boldsymbol{\Lambda}_K. \tag{D.18}$$

Then

$$\boldsymbol{S} = \boldsymbol{U}_{n+1} \boldsymbol{\Lambda}_{n+1} \boldsymbol{U}_{n+1}^T. \tag{D.19}$$

If the sample index $i$ greater than $d$, we truncate $\boldsymbol{U}_{n+1}$ and $\boldsymbol{\Lambda}_{n+1}$ to retain only the top $d$ eigenvalues and corresponding eigenvectors, thereby obtaining a rank $\leq d$ approximation of $\boldsymbol{X}\boldsymbol{X}^T$. The computational cost of the update ((D.17) and (D.18)) is $\mathcal{O}(nDd^2)$, with storage of $\mathcal{O}(Dd)$.

## E. Additional Explanations on Algorithm 2

**Landmark Update** Let $\{\boldsymbol{x}_j\}_{j=1}^n$ denote the previously observed noisy samples within the neighborhood $R(i)$ of the landmark $\boldsymbol{q}_i$, and let $\boldsymbol{x}_{n+1}$ be the newly arrived sample with that also lies within this neighborhood. At $(n+1)$-th iteration, the landmark is updated by computing the average over all $n+1$ noisy samples:

$$\boldsymbol{q}_i^{n+1} = \frac{1}{n+1} \sum_{k=1}^{n+1} \boldsymbol{x}_k. \tag{E.1}$$

However, this update rule requires reprocessing all past samples at each iteration, leading to computational inefficiency. To address this, our algorithm employs a streaming (or online) averaging method that incrementally updates the landmark using the previous average:

$$\boldsymbol{q}_i^{n+1} = \frac{n\boldsymbol{q}_i^n}{n+1} + \frac{\boldsymbol{x}_{n+1}}{n+1}. \tag{E.2}$$

**Projection Definition** Let $\boldsymbol{U}_i$ be the approximated basis for the landmark $\boldsymbol{q}_i$. Then the projection of a point $\boldsymbol{x}$ onto the affine space $\boldsymbol{q}_i + T_i$ is defined as

$$\mathcal{P}_{\boldsymbol{q}_i+T_i}\boldsymbol{x} \doteq \boldsymbol{q}_i + \boldsymbol{U}_i\boldsymbol{U}_i^T(\boldsymbol{x}-\boldsymbol{q}_i). \tag{E.3}$$

**Create First-Order Edges for New Landmark** When a new landmark $\boldsymbol{q}_i$ is created, first-order edges are established according to the following rule:

$$\text{create} \quad i \overset{1}{\leftrightarrow} j \quad \text{if} \quad \|\boldsymbol{q}_i - \boldsymbol{q}_j\|_2 \leq R_{\text{nbrs}}. \tag{E.4}$$

**Edge Embedding Definition** The edge embedding $\boldsymbol{\Xi}_i$ associated with the landmark $\boldsymbol{q}_i$, whose approximate tangent space basis is $U_i$, is defined as

$$\left\{ \boldsymbol{U}_i^T(\boldsymbol{q}_j - \boldsymbol{q}_i) \mid i \overset{1}{\to} j \in E^1 \right\}, \tag{E.5}$$

where $E^1$ denotes the set of first-order edges.

## F. Choosing Denoising Radius

The parameter called denoising radius $R(i)$ in Algorithm 2 controls complexity by determining the number of landmarks created. Conceptually, as the online algorithm learns, the error in landmarks decreases, which means that $R(i)$ needs to be decreased as the landmark gets learned. This is why we define a general formula for $R(i)$ as follows:

$$R(i)^2 = c_1 \left( \sigma^2 D + \frac{\sigma^2 D}{N_i^k} + c_2\sigma^2 d \right), \tag{F.1}$$

where $N_i$ denotes the number of points assigned to landmark $\boldsymbol{q}_i$. The power parameter $k$ helps us control the speed of decay of $R(i)$, making it adaptable to different datasets. Table 1 show the specific constants used to create the $R(i)$ parameter to produce Figure 9.

Table 1: The choice of hyperparameters yielding each denoiser. $N_i$ corresponds to the number of points assigned to a landmark $q_i$. For all experiments, $\sigma = 0.01$, $d = 2$, and $D = 2048$.

| DENOISER # | $R^2_{\text{DENOISING}}$ | $R^2_{\text{NBRS}}$ |
|---|---|---|
| 1 | $1.2(\sigma^2 D + \frac{\sigma^2 D}{N_i^{1/2}} + 20\sigma^2 d)$ | $2.39\sigma^2 D$ |
| 2 | $2.06\sigma^2 D$ | $2.39\sigma^2 D$ |
| 3 | $1.2(\sigma^2 D + \frac{\sigma^2 D}{N_i^{1/2}} + 8\sigma^2 d)$ | $2.39\sigma^2 D$ |
| 4 | $2.75\sigma^2 D$ | $3.13\sigma^2 D$ |
| 5 | $1.3(\sigma^2 D + \frac{\sigma^2 D}{N_i^{1/3}} + 20\sigma^2 d)$ | $2.39\sigma^2 D$ |
| 6 | $1.15(\sigma^2 D + \frac{\sigma^2 D}{N_i^{1/2}} + 4\sigma^2 d)$ | $2.39\sigma^2 D$ |
| 7 | $2.39\sigma^2 D$ | $2.75\sigma^2 D$ |
| 8 | $1.5(\sigma^2 D + \frac{\sigma^2 D}{N_i^{1/2}} + 30\sigma^2 d)$ | $2.39\sigma^2 D$ |
| 9 | $2\sigma^2 D$ | $2.39\sigma^2 D$ |
| 10 | $2.19\sigma^2 D$ | $2.39\sigma^2 D$ |
| 11 | $3.13\sigma^2 D$ | $3.53\sigma^2 D$ |
| 12 | $1.94\sigma^2 D$ | $2.39\sigma^2 D$ |

## G. Additional Experiments and Details

### G.1. Autoencoders

We include nonlinear autoencoders as an additional baseline for denoising gravitational wave signals, as they are generic learning architectures specifically designed to leverage low-dimensional structure in data. To this end, we create 15 different networks (see Table 2) with various depths and widths and symmetric encoder/decoders. The widest layers match the input dimensionality, while the narrowest bottleneck layer is set to dimension 2. All autoencoders are trained using the Adam optimizer with a learning rate of $1 \times 10^{-3}$. As we can see in the Figure 13, high-complexity autoencoders can reach high accuracy. We also observe that while autoencoders we tested can achieve higher accuracy, our method exhibits significantly better efficiency-accuracy trade-offs. In our work, we notice that varying the $R(i)$ parameter in our mixed-order method improves accuracy of manifold traversal method. As a direction for future work, we plan to further investigate how adjusting $R(i)$ can enable our method to access higher-accuracy regimes.

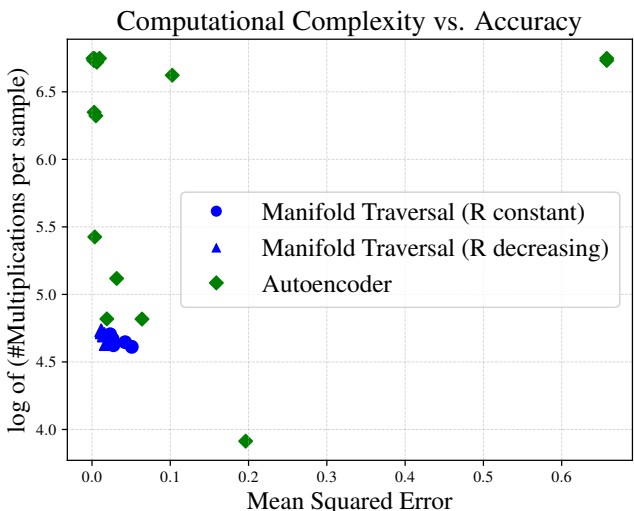

Figure 13: **Test-Time Complexity-Accuracy Tradeoff of Manifold Traversal versus Autoencoders.**

Table 2: Autoencoder Architectures. Decoder is a mirror image of the encoder.

| NUMBER | # OF LAYERS IN ENCODER | DIMENSIONALITY OF LAYERS |
|:---:|:---:|:---:|
| 1 | 1 | $2048 \rightarrow 2$ |
| 2 | 1 | $2048 \rightarrow 1024$ |
| 3 | 2 | $2048 \rightarrow 512 \rightarrow 2$ |
| 4 | 2 | $2048 \rightarrow 1024 \rightarrow 512$ |
| 5 | 4 | $2048 \rightarrow 512 \rightarrow 128 \rightarrow 16 \rightarrow 2$ |
| 6 | 4 | $2048 \rightarrow 1024 \rightarrow 512 \rightarrow 256 \rightarrow 128$ |
| 7 | 6 | $2048 \rightarrow 1024 \rightarrow 512 \rightarrow 128 \rightarrow 64 \rightarrow 16 \rightarrow 2$ |
| 8 | 6 | $2048 \rightarrow 1024 \rightarrow 512 \rightarrow 256 \rightarrow 128 \rightarrow 64 \rightarrow 32$ |
| 9 | 8 | $2048 \rightarrow 1024 \rightarrow 512 \rightarrow 128 \rightarrow 64 \rightarrow 32 \rightarrow 16 \rightarrow 8 \rightarrow 2$ |
| 10 | 8 | $2048 \rightarrow 1024 \rightarrow 512 \rightarrow 256 \rightarrow 128 \rightarrow 64 \rightarrow 32 \rightarrow 16 \rightarrow 8$ |
| 11 | 10 | $2048 \rightarrow 1024 \rightarrow 512 \rightarrow 256 \rightarrow 128 \rightarrow 64 \rightarrow 32 \rightarrow 16 \rightarrow 8 \rightarrow 4 \rightarrow 2$ |
| 12 | 2 | $2048 \rightarrow 32 \rightarrow 2$ |
| 13 | 3 | $2048 \rightarrow 64 \rightarrow 32 \rightarrow 4$ |
| 14 | 3 | $2048 \rightarrow 16 \rightarrow 8 \rightarrow 4$ |
| 15 | 3 | $2048 \rightarrow 16 \rightarrow 4 \rightarrow 2$ |

### G.2. Large-Scale Experiment

We evaluate our method on large-scale real-world image data by performing patch-level denoising. Specifically, we randomly select 300 RGB images from ImageNet, each corrupted with zero-mean Gaussian noise with a standard deviation of $\sigma = 0.1$. From these, we extract $894, 262$ noisy patches of size $8 \times 8 \times 3$ with the stride 8. After random shuffling, we use the first $890, 000$ patches to train our traversal network. As shown in Figure 14, the training error steadily decreases, indicating effective learning for denoising. One visual examples of clean, noisy, and denoised patches and images are shown in Figure 15. While integrating image processing techniques to address pixelation and color artifacts may further stabilize and enhance denoising performance and visual quality, such exploration is beyond the scope of this paper.

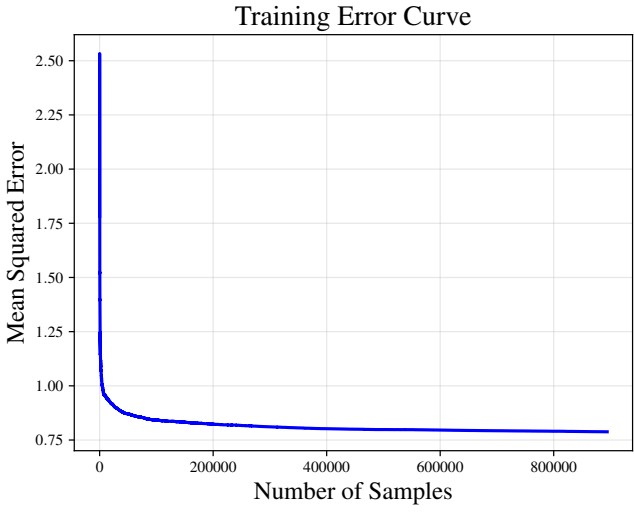

Figure 14: Training Mean Squared Error (MSE) Curve for Large-Scale Image Denoising.

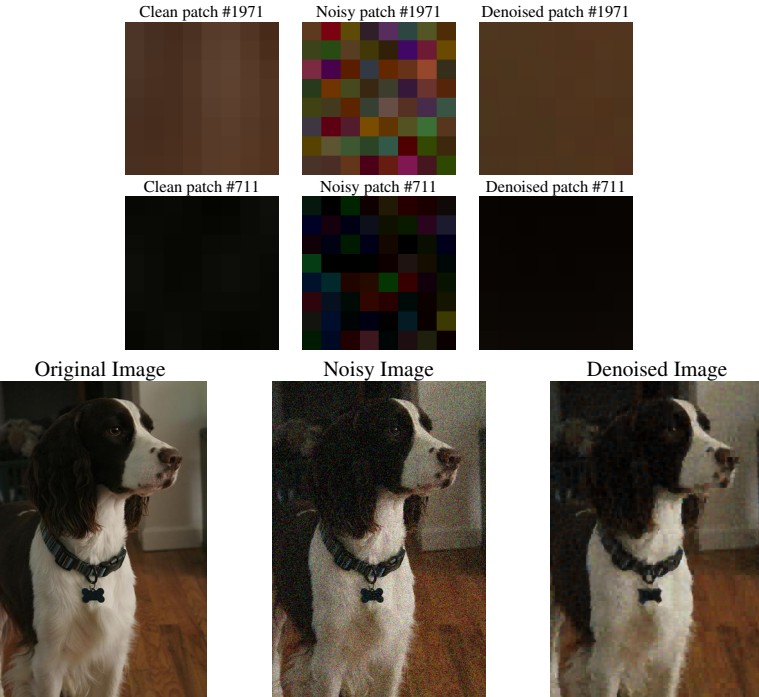

Figure 15: **Large-Scale Image Denoising Top and Middle:** Two different randomly selected patches from the random select image. Each row shows, left-to-right: the clean patch, the noisy patch, and the denoised patch. **Bottom:** Left-to-right: clean high-resolution image, noisy image, denoised image.

### G.3. Single-Image Denoising

We test our method in extreme scenarios when only a single noisy sample is available – an interesting and challenging setting in many important applications. To investigate our method's performance in such a challenging setting, we conduct an additional experiment to denoise a single natural image from the DIV2K dataset(Agustsson & Timofte, 2017) – a high-resolution dataset of natural RGB images with diverse contents. We corrupt the image with Gaussian noise at a noise level of 0.1 and apply our method for patch-level denoising, leveraging the common assumption that image patches lie on a low-dimensional manifold. Specifically, we extract patches of size $8 \times 8 \times 3$ with a stride $4$, yielding $172,042$ patches from a randomly selected image. After shuffling, we use the first $170,000$ patches for training. Figure 16 shows the mean squared error (MSE) decreasing as the number of training patches increases, indicating that the proposed method effectively learns to denoise. We also provide visual comparisons of clean, noisy, and denoised patches and full images in Figure 17.

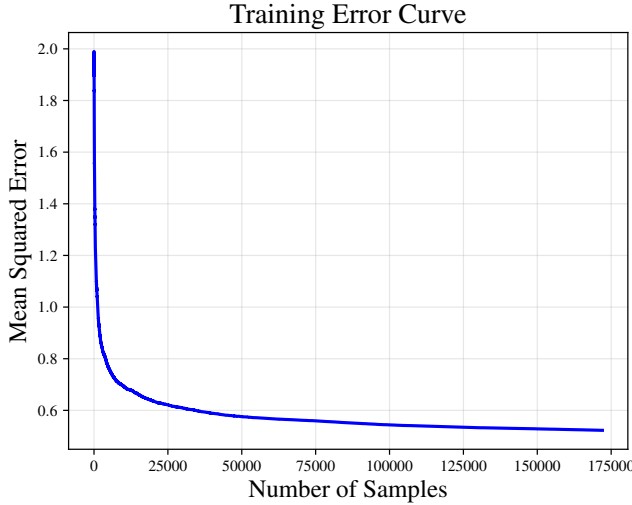

Figure 16: Training Mean Squared Error (MSE) Curve for Single-Image Denoising.

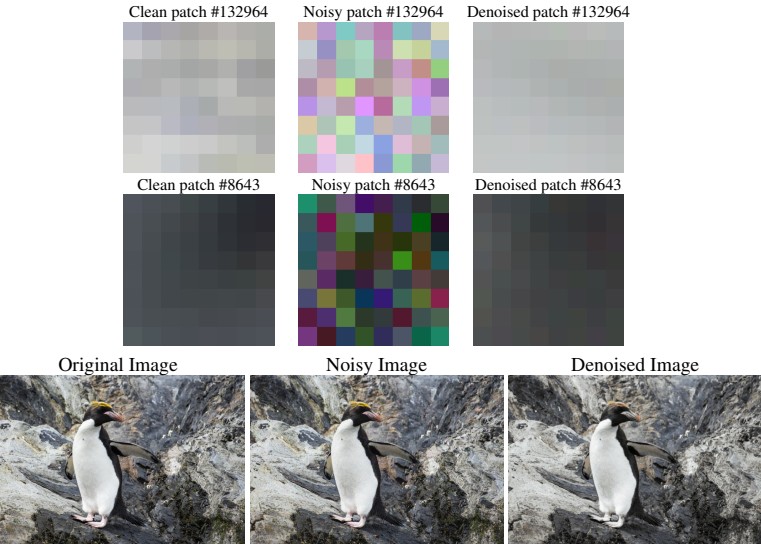

Figure 17: **Single Image Denoising** In the extreme case where only one image is available, our manifold traversal method successfully performs denoising. **Top and Middle:** Two different randomly selected patches from the high-resolution image. Each row shows, left-to-right: the clean patch, the noisy patch, and the denoised patch. **Bottom:** Left-to-right: clean high-resolution image, noisy image, denoised image.

### G.4. Ablation Study on Mixed-Order Method

This section presents an ablation study of the mixed-order method to justify the combination of first-order steps for efficiency and zero-order steps for achieving global optimality. Specifically, we compare with (i) a first-order optimization method exclusively on first-order edges generated by the online learning algorithm, and (ii) a zero-order optimization method applied to all edges, where the original first-order edges are treated as zero-order. Both methods use the same step rules as their respective components in the proposed mixed-order method. The experiments are conducted on the noisy version of the gravitational wave test set described in Section B.1.

As shown in Figure 18, the efficient gradient-based first-order method achieves the lowest computational cost but suffers from the highest error. On the other hand, our mixed-order method achieves a significantly better trade-off between complexity

and accuracy in the high-accuracy regime compared to the zero-order method. These results confirm the effectiveness of our mixed-order method, especially in achieving *efficient, high-accuracy* denoising.

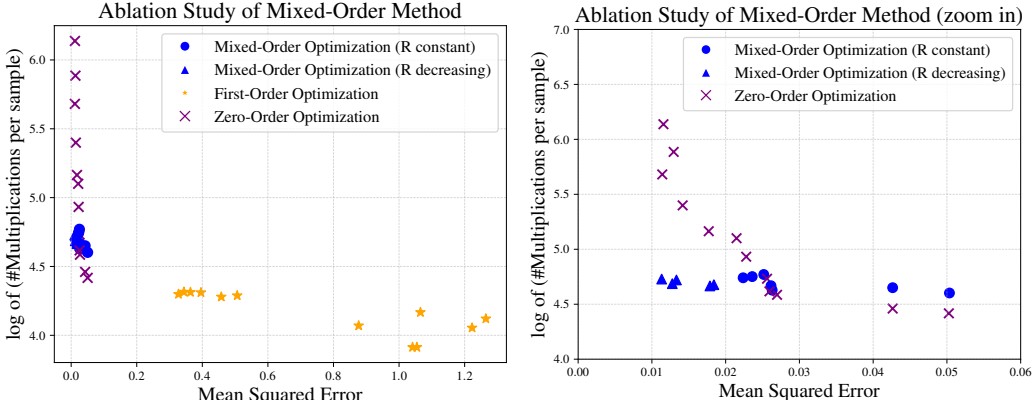

Figure 18: **Left:** Complexity-accuracy trade-off among the first-order optimization, zero-order optimization, and our proposed mixed-order optimization. **Right:** A zoomed-in view of the left portion of the left plot.

