# OpenReview forum: "Fast, Accurate Manifold Denoising by Tunneling Riemannian Optimization"
_ICML.cc/2025/Conference — ICML 2025 poster_

### Official Review · Reviewer_eTvP · 2025-03-08

**Overall Recommendation:** 3

**Summary:**

The paper considers the problem of constructing efficient denoisers that map noisy samples from a manifold to the manifold. An online learning approach is used to construct a graph for optimization, and a mixed-order method is used to aid optimization in order to achieve good performances. Theoretical analyses are provided, and experiments are performed to demonstrate the effectiveness of the proposed approach.

## update after rebuttal

The authors provided more evidence that further strengthens the paper. I keep my original score, weak accept, and did not increase it as I am not that familiar with the task in general.

**Claims And Evidence:**

The claims are supported by mathematical proofs and empirical experiments.

**Essential References Not Discussed:**

N/A

**Experimental Designs Or Analyses:**

The presented experiments are generally well-motivated, and I find the experimental settings sound and valid.

**Methods And Evaluation Criteria:**

The method is shown to outperform the nearest neighbor approach, which is a natural baseline, and I find the proposed method satisfying. My minor complaint is in the abstract the paper is motived using diffusion models, to some degree drawing expectation that a neural approach will be employed, yet the proposed method is largely graph based.

**Other Comments Or Suggestions:**

Page 2, second column, line 97: "Literature on Reimannian Optimization": should be "Literature on Riemannian optimization"

Page 3, first column, line 147-153 and also later: "The projection problem (4)", "The optimization problem (4)", but the equation is not given a label

Page 5, caption of Figure 4: "point q is a local minimizer of q over M", I think the second q should be replaced by the objective

**Other Strengths And Weaknesses:**

In terms of strengths, much theoretical analysis is provided to justify the proposed method. It is interesting that the proposed method outperforms the nearest neighbor approach.

**Questions For Authors:**

The paper proposes using both first order and zeroth order Riemannian optimization. It can be interesting to provide an ablation study on the effect of removing one of them.

**Relation To Broader Scientific Literature:**

The paper considers combining graph-based traversal with Riemannian optimization, which are both established methods.

**Theoretical Claims:**

The claims on Riemannian optimization are generally standard. I did not check the claims in theoretical analysis section.

---

> ### Author Rebuttal · Authors · 2025-04-01
>
> We deeply appreciate the reviewer’s positive feedback and insightful suggestions! We are delighted that the reviewer found our proposed method satisfactory and valued both the supporting experimental results (particularly the strong performance compared to the natural baseline of nearest neighbor search) and the accompanying theoretical analysis.
>
> Below, we address the reviewer’s questions regarding the connection between our method and diffusion models, and we present an ablation study of the mixed-order method, as suggested. Additionally, we conducted further experiments demonstrating that our method also outperforms AutoEncoders, a widely
> used generic learning baseline. All updated experimental results are available at [Figures.pdf](https://anonymous.4open.science/r/Figures_Manifold_Traversal-8D25/Figures.pdf) (https://anonymous.4open.science/r/Figures_Manifold_Traversal-8D25/Figures.pdf).
>
> **Ablation Study:** We thank the reviewer for this excellent idea! We conducted the suggested ablation study on the mixed-order method by selectively removing the first-order or zero-order optimization components one at a time. The results in [Figure 3](https://anonymous.4open.science/r/Figures_Manifold_Traversal-8D25/Figures.pdf)  confirm that mixed-order optimization has significant advantages over both first- and zeroth-order optimization for efficient, high-accuracy denoising.
>
> In [Figure 3](https://anonymous.4open.science/r/Figures_Manifold_Traversal-8D25/Figures.pdf), we observe that the first-order optimization incurs the lowest computational cost. As expected, it also yields the highest error, due to getting stuck at local minima. In the high-accuracy regime, our mixed-order optimization achieves a more favorable complexity–accuracy tradeoff compared to zero-order optimization. This advantage arises because zero-order methods are less efficient when operating over dense graphs. Conversely, in the low-accuracy regime where the underlying graph is sparse, zero-order optimization slightly outperforms the mixed-order method. This is expected, as in sparse graphs the mixed-order method essentially relies only on its zero-order steps, behaving similarly to pure zero-order optimization.
>
> **Connection to Diffusion Models:** As reviewer pointed out, diffusion models use generic learning architectures. To further validate our approach, we conducted additional experiments comparing our method with AutoEncoders -- a natural generic learning baseline designed to exploit low-dimensional structure in data. As illustrated in [Figure 4](https://anonymous.4open.science/r/Figures_Manifold_Traversal-8D25/Figures.pdf), our mixed-order optimization method -- designed to exploit both the optimization structure of the denoising problem and the manifold structure of the data -- achieves a substantially better accuracy-complexity trade-off compared to the autoencoder. This further highlights the efficiency and effectiveness of our approach. We will add this comparison to the final version of the manuscript. Below we describe the details of this experiment.
>
> To perform an extensive comparison to Autoencoders as baseline, we create eleven different networks with various depths and widths and symmetric encoder/decoders. The widest autoencoder layer has width equal to the data dimensionality, and the smallest bottleneck equals 2. The shallowest autoencoder (which appears as the point on the low-right corner of [Figure 4](https://anonymous.4open.science/r/Figures_Manifold_Traversal-8D25/Figures.pdf)), has one layer in the encoder, and one in the decoder. The deepest autoencoder has 10 layers in the encoder, and 10 in the decoder. As we can see in [Figure 4](https://anonymous.4open.science/r/Figures_Manifold_Traversal-8D25/Figures.pdf), while autoencoders can achieve slightly higher accuracy, our method exhibits significantly better efficiency-accuracy tradeoffs -- by orders of magnitude. In our work, we notice that varying the $R(i)$ parameter in our mixed-order method improves accuracy of manifold traversal, and we plan to explore further in this direction in future work.
>
> Since the denoiser is a critical building block of diffusion models and most test-time compute is being spent on denoising, improving its test-time efficiency can significantly accelerate the overall process. As part of our immediate next work, we plan to integrate the proposed method into diffusion models to further explore this direction.
>
> We also appreciate the reviewer’s careful attention to detail, which has helped us improve the clarity and presentation of the paper. We will fix the typos in the final manuscript.
>
> **Closing Comment:** We sincerely thank the reviewer for their valuable feedback and thoughtful suggestions. We hope our rebuttal has effectively addressed the comments and questions. We would be glad to further discuss or clarify any additional questions.

---

### Official Review · Reviewer_PJMj · 2025-03-14

**Overall Recommendation:** 3

**Summary:**

This paper addresses the problem of efficiently denoising new noisy data sampled from an unknown manifold M, relying only on noisy samples. To this end, a framework for test-time efficient manifold denoising was proposed. In the theoretical analyses, the optimality of the proposed methods was elucidated. In the experimental analyses, complexity-performance tradeoffs compared to baseline methods were examined on scientific and imagery data.

Update after rebuttal
I checked all responses and comments from authors and reviewers. Some of my questions were addressed in the rebuttal. Therefore, I keep my score.

**Claims And Evidence:**

The main problems addressed in the paper are identified by “Learned denoisers play a fundamental role in various signal generation (e.g., diffusion models) and reconstruction (e.g., compressed sensing) architectures, whose success derives from their ability to leverage low-dimensional structure in data.”

In the theoretical analyses, convergence properties of the proposed methods, such as convergence regions and rates, were analyzed. However, these properties were not associated with those of the methods (e.g. diffusion and CS models) considered in the main problem definition.

In the experimental analyses, the proposed methods were examined on synthetic manifolds and High-Dimensional Scientific Data. However, further analyses on several additional tasks in comparison with the aforementioned models should be given.

**Essential References Not Discussed:**

Most of the major prior work were considered in the literature review.

**Experimental Designs Or Analyses:**

As mentioned in the previous comments, the proposed methods are evaluated in limited settings. The experimental analyses should be improved by comparing the proposed methods with additional baselines and related work using larger models on larger scale datasets.

**Methods And Evaluation Criteria:**

The proposed methods and evaluation criteria make sense for the problem or application at hand, but limited.

**Other Comments Or Suggestions:**

Please see the above comments.

**Other Strengths And Weaknesses:**

The paper is well written and mathematical results are clear. However, explanation of experimental setups and results should be improved. Moreover, authors should describe how they define and compute projections and solve the optimization problems given in the algorithms in the experimental analyses. Since these details are omitted, implementation of the algorithms is not trivial and reproduction of the results can be challenging.

**Questions For Authors:**

Please see the above comments.

**Relation To Broader Scientific Literature:**

As mentioned above, the paper is well written in general. However, theoretical and experimental analyses and results are limited, and should be improved in comparison with the other denoising methods.

**Theoretical Claims:**

Theoretical analyses are clear, and I enjoyed the proofs. However, further comparison with the other baselines and related models would improve the novelty and merit of the results.

---

> ### Author Rebuttal · Authors · 2025-04-01
>
> We thank the reviewer for their appreciation of our writing and theoretical analysis. At the core of our work is a novel, efficient and accurate algorithm for manifold denoising, which reframes learning-to-denoise as learning-to-optimize. Our key technical innovations includes an accurate and efficient mixed-order method, and a scalable online learning algorithm that learns to optimize from only noisy data. This design enables our method to scale effectively to large, high-dimensional datasets, and to practical scenarios where the manifold is unknown and only noisy samples are available.
>
> Below, we present additional experiments and explanations. We also conducted an ablation study ([Figure 3](https://anonymous.4open.science/r/Figures_Manifold_Traversal-8D25/Figures.pdf)), which shows the superiority of mixed-order over both first- and zeroth-order methods. All new experimental results are available at https://anonymous.4open.science/r/Figures_Manifold_Traversal-8D25/Figures.pdf.
>
> **Connection to Larger Architectures:** Thank you for your insightful comment. Denoising serves as a critical building block for diffusion-based signal generation and signal recovery (compressed sensing (CS)). Our theory demonstrates that mixed-order optimization solves the manifold denoising problem efficiently and accurately — with a number of operations that scales well in the ambient dimension, and a mean-squared error which is close to statistically optimal. These properties have implications for the efficiency and performance of architectures that use denoising as a building block: the overall number of operations is bounded by the number of diffusion/CS outer loop step times the number of operations used by the denoiser. Moreover, in denoising-based CS reconstruction, there is a direct relationship between the MSE of the denoiser and the MSE of CS reconstruction [Metzler et al, 2016]. By reframing the denoising problem (inner loop) as the optimization problem, which is within the outer loop of tasks such as signal reconstruction, we can consolidate these two loops into a single optimization framework, which could yield significant improvements in computation. We leave this direction to future work.
>
> **Additional Experimental Analyses:** We sincerely thank the reviewer for suggestions. We have conducted additional experiments on real-world RGB images to further demonstrate the denoising capability of our method. In the scenario where only a single noisy image is available, the training error curve shows a clear decreasing trend [Figures 5, 6, and 7](https://anonymous.4open.science/r/Figures_Manifold_Traversal-8D25/Figures.pdf). In the case with large-scale data, we see that the method is able to learn meaningful structure of real-world image patches ([Figure 8](https://anonymous.4open.science/r/Figures_Manifold_Traversal-8D25/Figures.pdf)). In our next work, we will integrate the proposed method into larger architectures, such as CS and diffusion models.
>
> **Additional Baseline:** We thank the reviewer for the helpful and constructive feedback. As an additional baseline, we have added nonlinear autoencoders, which are generic learning architectures that capture low-dimensionality. Our mixed-order method — designed to exploit optimization structure of denoising problems and manifold structure of data — achieves a substantially better accuracy-complexity trade-off compared to autoencoders ([Figure 4](https://anonymous.4open.science/r/Figures_Manifold_Traversal-8D25/Figures.pdf)), further highlighting the efficiency of our approach. We will add this comparison to the final manuscript. For more details, we kindly refer the reviewer to our response ( paragraph [Connection to Diffusion Models]) to Reviewer eTvP due to space limitation.
>
> **Large-Scale Data Denoising:** We appreciate the reviewer’s feedback regarding the exploration of larger-scale data. We have applied our method to real-world large-scale image data and performed  denoising at the patch level. We use images from ImageNet with additive Gaussian noise on 1,310,000 patches. **The results on large-scale data further demonstrate our method's denoising ability([Figure 8](https://anonymous.4open.science/r/Figures_Manifold_Traversal-8D25/Figures.pdf)).**
>
> **Improved Documentation of Experimental Setup:** We sincerely thank the reviewer for highlighting the importance of clarifying experimental setup and algorithmic details. Reproducibility is instrumental to progressing science. We will give more details in the supplementary material in the revised manuscript, e.g., computation of incremental PCA for efficient tangent space estimation, definitions of projections, and other details. We will also release all code and data for reproducibility our work.
>
> **Closing Comment:**
> We sincerely thank the reviewer for their time and thoughtful feedback. We hope our responses effectively addressed and clarified the thoughtful questions posed by the reviewer.

---

### Official Review · Reviewer_9XFD · 2025-03-18

**Overall Recommendation:** 4

**Summary:**

This paper proposes a new framework for efficient denoising of noisy data sampled from an unknown manifold, which treats "learning to denoise" as "learning to optimize." The key innovations include: 1) online learning that learns to optimize clean signals using only noisy data and improve the optimizer on the fly; 2) mixed-order methods ensure that the learned optimizers reach global optimality, balancing efficiency and denoising performance. The authors provide theoretical analysis as well as conducting experiments on synthetic and real data.

**Claims And Evidence:**

Yes, I list the details in the [Other Strengths And Weaknesses] section.

**Essential References Not Discussed:**

The authors have discussed the most important and related works.

**Experimental Designs Or Analyses:**

Yes, I checked the experimental designs and results analysis.

**Methods And Evaluation Criteria:**

Yes, I list the details in the [Other Strengths And Weaknesses] section.

**Other Comments Or Suggestions:**

It would be very nice to see some visualization results showing the noisy data and the denoised results.

**Other Strengths And Weaknesses:**

Strengths

1) The paper introduces a novel method that translates the denoising process into an optimization problem over a manifold, leveraging Riemannian optimization techniques. It provides a new perspective on denoising.

2) The proposed method is efficiency. The approach is computationally efficient, particularly in high-dimensional settings, and scales well with the size of the dataset and dimensionality.

3) The introduction of mixed-order traversal which combines first-order and zero-order steps is interesting. It ensures robust convergence towards global minima, enhancing the reliability of the optimization process.

4) The idea of online learning is also novel. Its ability to build necessary structures on-the-fly as data points are encountered is a pioneering aspect, allowing for real-time adaptability and efficiency.

5) In addition, the authors not only provided the theoretical analyses with near-optimal denoising guarantees, but also conducted experiments on both synthetic manifolds as well as scientific and imagery data.



Weaknesses
1) I am wondering if due to some privacy policy, one only collected a very few noisy samples (e.g., in a very extreme case, a single noisy data), will this method still work? Since the noisy samples are very small, then the online learning strategy may not work as expected.
2) The other extreme case, for large-scale, complex image data with different kinds of noise, saying the noise from this benchmark (https://github.com/hendrycks/robustness), how would this method work?

**Questions For Authors:**

I list some of my questions in the [Other Strengths And Weaknesses] section.

**Relation To Broader Scientific Literature:**

I list the details in the [Other Strengths And Weaknesses] section.

**Theoretical Claims:**

Yes, I checked part of the theoretical results.

---

> ### Author Rebuttal · Authors · 2025-04-01
>
> We thank the reviewer for the positive feedback and thoughtful questions! We're glad the core idea of learning-to-optimize for denoising, along with our mixed-order method and scalable online learning approach, was well received. We also appreciate the recognition of our experimental and theoretical contributions. Below, we address the reviewer’s questions with additional results. Additionally, we present an ablation study ([Figure 3] (https://anonymous.4open.science/r/Figures_Manifold_Traversal-8D25/Figures.pdf)), which further highlights that mixed-order optimization has significant advantages over both first- and zeroth-order optimization for efficient, high-accuracy denoising. All updated results are available at [Figures.pdf] (https://anonymous.4open.science/r/Figures_Manifold_Traversal-8D25/Figures.pdf).
>
> **Single Noisy Data Denoising:** We thank the reviewer for this excellent question! The scenario where only a single noisy sample is available is indeed both interesting and challenging. To investigate this setting, we conducted an additional experiment focused on denoising a single natural image from the DIV2K dataset [Agustsson and Timofte, 2017]  — a high-resolution dataset of natural RGB images with diverse contents. The result showed that the training error steadily decreased ([Figures 5, 6, 7] (https://anonymous.4open.science/r/Figures_Manifold_Traversal-8D25/Figures.pdf)). Details follow below.
>
> As suggested, we applied our method to a single training image, performing patch-level denoising under the common assumption that patches lie on a low-dimensional manifold. We choose a patch size as $ 8 \times 8 \times 3$, with a stride 8, which results in about 50,000 patches in a single randomly selected image from the DIV2K dataset. [Figures 5, 6, 7] (https://anonymous.4open.science/r/Figures_Manifold_Traversal-8D25/Figures.pdf) also show denoised images and patches.
>
> **Large-Scale Data Denoising:** We appreciate the reviewer’s feedback and suggestions regarding the exploration of large-scale, complex image data. Following the reviewer’s suggestion, we have applied our method to real-world image data and performed  denoising at the patch level. We use images from ImageNet with additive Gaussian noise on 1,310,000 patches. The results demonstrate clear denoising progress, evidenced by steadily decreasing training error curves, as shown in [Figure 8](https://anonymous.4open.science/r/Figures_Manifold_Traversal-8D25/Figures.pdf).
>
> Traditionally, image degradations, such as additive noise, blur, zoom, saturation, and others, are handled via iterative reconstruction methods. These methods alternate between two steps: (1) enforcing consistency with the observed image via a degradation model, and (2) applying a prior on the clean image, typically through a proximal operator. In approaches like Plug-and-Play [Venkatakrishnan et al., 2013], this proximal step is replaced by a learned denoiser, allowing for more flexible and data-driven regularization. Our method can be directly applied as a denoiser for Gaussian noise. However, for more complex degradations  such as motion blur and others — a single-step L2 projection onto the clean-signal manifold is insufficient. Just as our method can be integrated into diffusion models and compressed sensing reconstruction pipelines, it can also be incorporated into iterative algorithms for signal restoration under more complex degradations. When the degradation model is known, frameworks like Plug-and-Play can be used to combine our denoising method with degradation-specific reconstruction steps.
>
> Our method remains applicable to other types of challenging degradations when embedded within an iterative reconstruction framework that explicitly accounts for the degradation process, such as Plug-and-Play.
>
> **Visualizations of Denoised Data:** We thank the reviewer for this helpful suggestion. In response, we have added visualizations that display the noisy inputs, the corresponding denoised outputs, and the ground truth signals. These examples — now presented in [Figure 2](https://anonymous.4open.science/r/Figures_Manifold_Traversal-8D25/Figures.pdf) (noisy gravitational wave (GW), denoised GW, and ground truth GW) and [Figures 5, 6, 7] (https://anonymous.4open.science/r/Figures_Manifold_Traversal-8D25/Figures.pdf) (ground truth image, noisy image, and denoised image) — offer qualitative evidence of the effectiveness of our method and clearly demonstrate its ability to recover clean signals from noisy observations.
>
> **Closing Comment:** We sincerely thank the reviewer for their thoughtful feedback and constructive suggestions. We're glad that the core ideas and contributions of our work resonated with you, and we truly appreciate your comments, which helped us further improve our work.

---

### Official Review · Reviewer_6tJ4 · 2025-03-19

**Overall Recommendation:** 3

**Summary:**

This paper studies the interesting manifold denoising problem (ref. 1) with the focus “learning-to-optimize” and proposes test-time efficient denoising algorithm. Also mixed-order optimization is proposed to help achieve near optimal results, given first-order gradient only optimization is more efficient.

Strengths: the high-level idea of combing both first order (gradient) and zero order (zero-order neighbors search) is interesting, and the construction of manifold traversal networks is good to see. Also, theoretical analysis (section 7 and appendix) results show the proposed algorithm can achieve certain error accuracy (i.e., linked with geodesic curvature, intrinsic dimension) with upper bounded computational complexity.

Experimental results on synthetic data are presented in section 8 and illustrate the trend of training stage denoiser error converging to theoretical lower bound. Test time efficiency and the comparisons with nearest neighbor search are included to show the proposed algorithm can achieve same level of accuracy with lower computational cost.

References:

1. Hein, M. and Maier, M. Manifold denoising. Advances in neural information processing systems, 19, 2006

**Claims And Evidence:**

Yes

**Essential References Not Discussed:**

This work cited a number of papers from traditional denoising methods to more recent deep neural network leaning based paper, and for manifold denoising then ref. 1 below is included, though there are more directly related works for manifold denoising, e.g., more recent denoising works like ref. 2 or others, should be good to include more closely related work discussions.


Ref. 1: Hein, M. and Maier, M. Manifold denoising. Advances in neural information processing systems, 19, 2006

Ref. 2: D. Gong, F. Sha, and G. Medioni. Locally linear denoising on image manifolds. AISTATS, Proc. Mach. Learn. Res., 9:265–272, 2010

**Experimental Designs Or Analyses:**

Section 8 included both (1) training error converge trend, and (2) test time accuracy vs. complexity tradeoff, and figure 5 for visualization of graph construction results for manifold traversal networks.

**Methods And Evaluation Criteria:**

Yes

**Other Comments Or Suggestions:**

1. Landmark, first mentioned in section 4 (page 4), feel should be helpful to add some illustration here, i.e., high level the concept and usage of landmark in this work, consider it is one important part of the proposed denoising framework.

**Other Strengths And Weaknesses:**

Limitations:

1. Experimental results, unless I missed, seems only included synthetic data for the test time accuracy evaluation and comparisons with alternative methods. It will be informative to see results from real world data, and some real-world data indeed with lower intrinsic dimension.

2. Section 8 only included "nearest neighbor search" as the alternative baseline, it should be helpful to include other (more recent) works, to support the proposed algorithm in this paper.

**Questions For Authors:**

NA

**Relation To Broader Scientific Literature:**

NA

**Theoretical Claims:**

I looked Theorem 7.1, and some more from appendix section A, though not very carefully for examination.

---

> ### Author Rebuttal · Authors · 2025-04-01
>
> We sincerely appreciate the reviewer’s valuable feedback to improve our work. We are pleased that the reviewer finds the manifold denoising problem interesting, as well as our idea of rethinking learning-to-denoise as learning-to-optimize, and our accurate and efficient mixed-order method. We are also glad that the reviewer appreciates our theoretical analysis, which demonstrates near-optimal denoising performance and establishes an upper bound on the computational complexity, both of which are tied to the manifold’s geometric properties. Below, we respond to the reviewer’s questions with additional experiments. All updated results are available at [Figures.pdf] (https://anonymous.4open.science/r/Figures_Manifold_Traversal-8D25/Figures.pdf).
>
> **Additional References:** We thank the reviewer for suggesting the inclusion of additional related works. We will incorporate [Hein and Maier, 2006], [Gong et al, 2010], [Hu et al, 2021], [Puchkin and Spokoiny, 2022] in the final manuscript. We discuss the two suggested references below.
>
> [Hein and Maier, 2006] presents a manifold denoising algorithm based on graph-based diffusion process. [Gong et al, 2010] approximates the manifold by linear subspaces, and denoises based on the local linear approximation. Both works and ours denoise based on only noisy data without prior knowledge of the manifold. However, they are inefficient at test time: denoising previously unseen data requires a linear scan to locate its nearest neighbors. At the core, these methods rely on nearest neighbor search, while our approach achieves significantly better accuracy-complexity trade-offs at test-time.
>
> **Real-World Data:** We appreciate the reviewer’s thoughtful suggestion. We would like to emphasize that using synthetic gravitational waves (GWs) is standard practice in GW astrophysics, as it enables systematic evaluation of methods’ performance/limitations in a setting where only a few hundred confirmed detections exist.
>
> Following the suggestion, we have also applied our method to real-world images; patches are assumed to lie near a low-dimensional manifold, and denoising is performed at the patch level. We perform experiments on large-scale real-world data from ImageNet with additive Gaussian noise on 1,310,000 patches. **These experiments on large-scale data further demonstrate our method's denoising ability beyond synthetic settings** ([Figure 8] (https://anonymous.4open.science/r/Figures_Manifold_Traversal-8D25/Figures.pdf)).
>
> We tested our method under extreme scarcity, when only one noisy real-world image is available as training data. The results show clear denoising progress, with steadily decreasing training error curves ([Figures 5, 6, 7] (https://anonymous.4open.science/r/Figures_Manifold_Traversal-8D25/Figures.pdf)). We will include training curves in the main paper and additional visuals in the Appendix of the final manuscript.
>
> **Additional Baseline Comparison:** We thank the reviewer for helpful feedback. We chose nearest-neighbor search as our baseline because it serves as the foundation for state-of-the-art provable denoising methods [Yao et al, 2023], [Sober and Levin, 2020]. Our proposed mixed-order method achieves orders-of-magnitude improvements in computation while maintaining theoretical guarantees.
>
> As suggested, we have added nonlinear Autoencoders as a baseline, which are generic learning architectures designed to leverage low-dimensionality. The result shows that our mixed-order optimization method — designed to exploit both the optimization structure of the denoising problem and the manifold structure of data — achieves a substantially better accuracy-complexity trade-off compared to the Autoencoder ([Figure 4] (https://anonymous.4open.science/r/Figures_Manifold_Traversal-8D25/Figures.pdf)). We will add this comparison to the final manuscript. For more details, we kindly refer the reviewer to the our response (**[Connection to Diffusion Models]** paragraph) to Reviewer eTvP for more details due to space limitation.
>
> **Additional Landmark Elaboration:** We thank the reviewer for emphasizing the need to clarify the landmark concept. We will include
> the following illustrations in the final manuscript. [Figure 1] (https://anonymous.4open.science/r/Figures_Manifold_Traversal-8D25/Figures.pdf) provides an illustration of landmarks. Intuitively, they serve as a discrete approximation of the unknown manifold. To enable optimization over the manifold, we need a structured domain, which is formed by the landmarks and connecting edges, facilitating optimization. Importantly, all these components — including the landmarks, edges, and other related quantities — are learned directly from the noisy data, using the proposed online learning algorithm.
>
> **Closing Comments:** We sincerely thank the reviewer for the valuable feedback, which helped us improve the work. We hope our responses address your concerns and are happy to provide further clarification if needed.

---

### Decision · Program_Chairs · 2025-05-01

**Decision:**

Accept (poster)

**Comment:**

The paper proposes a new mixed-order optimization method for manifold denoising (which, in the case of Gaussian noise, is equivalent to minimizing a quadratic function over the manifold). This is a natural problem whose efficient solution could potentially be useful in many settings. However, first-order methods can get stuck due to non-convexity, and to resolve this the authors propose an "edge-growing" scheme that allows local descent to escape bad optima. Statistical analysis is provided in support of this approach, and the authors show that the sample complexity dependence is linear in the ambient dimension and exponential in the manifold dimension, which isn't too bad for very low dimensional surfaces.

All reviewers concurred that the paper makes solid contributions and recommended an accept. My one big concern -- which prevents me from giving a higher score -- is the lack of compelling experiments; Section 8 is disappointingly brief. Given that the algorithm is natural and not too hard to implement, consider including more experimental evidence for more realistic denoising problems.